# Extreme Metrics from Large Ensembles: Investigating the Effects of Ensemble Size on their Estimates

Claudia Tebaldi[1], Kalyn Dorheim[1], Michael Wehner[2], and Ruby Leung[3]

[1]Joint Global Change Research Institute, Pacific Northwest National Laboratory, College Park, MD, USA
[2]Lawrence Berkeley National Laboratory, Berkeley, CA, USA
[3]Pacific Northwest National Laboratory, Richland, WA, USA

**Correspondence:** Claudia Tebaldi (claudia.tebaldi@pnnl.gov)

**Abstract.** We consider the problem of estimating the ensemble sizes required to characterize the forced component and the internal variability of a number of extreme metrics. While we exploit existing large ensembles, our perspective is that of a modeling center wanting to estimate a-priori such sizes on the basis of an existing small ensemble (we assume the availability of only 5 members here). We therefore ask if such small-size ensemble is sufficient to estimate accurately the population variance (i.e., the ensemble internal variability) and then apply a well-established formula that quantifies the expected error in the estimation of the population mean (i.e., the forced component) as a function of the sample size $n$, here taken to mean the ensemble size. We find that indeed we can anticipate errors in the estimation of the forced component for temperature and precipitation extremes as a function of $n$ by plugging into the formula an estimate of the population variance derived on the basis of 5 members. For a range of spatial and temporal scales, forcing levels (we use simulations under Representative Concentration Pathway 8.5), and two models considered here as our proof of concept, it appears that an ensemble size of 20 or 25 members can provide estimates of the forced component for the extreme metrics considered that remain within small absolute and percentage errors. Additional members beyond 20 or 25 add only marginal precision to the estimate, and this remains true when statistical inference through extreme value analysis is used. We then ask about the ensemble size required to estimate the ensemble variance (a measure of internal variability) along the length of the simulation, and – importantly – about the ensemble size required to detect significant changes in such variance along the simulation with increased external forcings. Using the $F$-test we find that estimates on the basis of only 5 or 10 ensemble members accurately represent the full ensemble variance even when the analysis is conducted at the grid-point scale. The detection of changes in the variance when comparing different times along the simulation, especially for the precipitation-based metrics, requires larger sizes, but not larger than 15 or 20 members. While we recognize that there will always exist applications and metric definitions requiring larger statistical power and therefore ensemble sizes, our results suggest that for a wide range of analysis targets and scales an effective estimate of both forced component and internal variability can be achieved with sizes below 30 members. This invites consideration of the possibility of exploring additional sources of uncertainty, such as physics parameter settings, when designing ensemble simulations.

**Keywords.** Large Ensembles, Extreme Metrics, Internal Variability, Forced Signal Detection, ETCCDI

# 1 Introduction

Recently, much attention and resources have been dedicated to running and analyzing large ensembles of climate model simulations under perturbed initial conditions (e.g., Deser et al., 2012; Pausata et al., 2015; Steinman et al., 2015; Bittner et al., 2016; Li and Ilyina, 2018; Maher et al., 2018; Deser et al., 2020; Lehner et al., 2020; Maher et al., 2021a, b). Both detecting the forced component in externally forced experiments, and quantifying the role of internal variability are being facilitated by the availability of these large ensembles. Many variables and metrics of model output have been analyzed, with large ensembles allowing precise estimates of their current and future statistics. Large ensembles are also being used to answer methodological questions, particularly about the precision these experiments can confer to the estimate of those variables and metrics, and how that varies with increasing ensemble sizes (e.g., Milinski et al., 2020). Recent efforts by multiple modeling centers to coordinate these experiments so that they can be comparable (by being run under the same scenarios of future greenhouse gas emissions) allow answering those questions robustly, accounting for the size and behavior over time of internal variability, which is known to be a model-specific characteristic (Deser et al., 2020).

In this methodological study we adopt the point of view of a modeling center interested in estimating current and future behavior of several metrics of extremes, having to decide on the size of a large ensemble. Such decision, we assume, needs to be reached on the basis of a limited number of initial condition members, which the center would run as a standard experiment. We choose a size of 5, which is a fairly common choice for future projection experiments, and use the statistics we derive on the basis of such small ensemble to estimate the optimal size of a larger ensemble, according to standards of performance that we specify. We test our estimate of the optimal size by using a perfect model setting, defining as 'the truth' what a full large ensemble gives us. We use two large ensembles available through the CLIVAR SMILES initiative (Lehner et al., 2020), the CESM1-CAM5 LENS (of 40 ensemble members, (Kay et al., 2015)) and the CanESM2 ensemble (of 50 members, (Kirchmeier-Young et al., 2017; Kushner et al., 2018)), both run over the historical period and under RCP8.5 according to the CMIP5 protocol (Riahi et al., 2011; Taylor et al., 2012).

Our metrics of interest are indices describing the tail behavior of daily temperature and precipitation. We conduct the analysis in parallel for extremes of temperature and precipitation because we expect our answers to be dependent on the signal-to-noise ratio affecting these two atmospheric quantities, which we know to be different in both space and time (Hawkins and Sutton, 2009, 2011; Lehner et al., 2020).

We consider the goal of identifying the forced change over the course of the $21^{st}$ century in the extremes behavior. We seek an answer in terms of the ensemble size for which we expect the estimate of the forced component to approximate the truth within a given tolerance, or for which our estimate does not change significantly with additional ensemble members. We also consider the complementary problem of identifying the ensemble size that fully characterizes the variability around the forced component. After all, considering future changes in extremes usually has salience for impact risk analysis, and any risk-oriented framework will be better served by characterizing both the expected outcomes (i.e. the central estimates) and the uncertainties surrounding them. Both types of questions can be formulated at a wide range of geographic scales, as the information that climate model experiments provide is used for evaluation of hazards at local scales, for assessment of risk and

adaptation options, all the way to globally aggregated metrics, usually most relevant for mitigation policies. The time horizon of interest may vary as well. Therefore we present results from grid-point scales all the way to global average scales, and for mid-century and late-century projections, specific years or decades along the simulations, or whole century-long trajectories.

The consideration of two models, two atmospheric quantities and several extreme metrics, each analyzed at a range of spatial and temporal scales help our conclusions to be robust and – we hope – applicable beyond the specifics of our study.

## 2 Models, Experiments and Metrics

The CESM1-CAM5 LENS (CESM ensemble from now on) has been the object of significant interest and many published studies, as the more than 1,300 citations of Kay et al. (2015) testify to, and, if in lesser measure, so has been the CanESM2 ensemble (CanESM ensemble from now on). The CESM model has a resolution of about 1 degree in the longitude-latitude dimensions (Hurrell et al., 2013), while CanESM has a coarser resolution of about 2 degrees (Arora et al., 2011). Both have been run by perturbing the atmospheric state at a certain date of the historical simulation (5 different simulations in the case of CanESM) by applying "errors" of the order of magnitude of machine precision. These perturbations have been found to generate alternative system trajectories that spread out losing memory in the atmosphere of the respective initial conditions within a few days of simulation time (Marotzke, 2019). We note that sources of variability from different ocean states, particularly at depth, are not systematically sampled by this type of ensembles, albeit they are partially addressed by the design of the CanESM2 that uses de-facto different ocean states. For CESM, 38 or 40 ensemble members (depending on the variable considered) are available, covering the period between 1920 and 2100, while CanESM only starts from 1950 but has 48 or 50 ensemble members. In the following we will not distinguish precisely between the full size or the full size minus two, as the results are not influenced by this small difference. Both models were run under historical and RCP8.5 external forcing, the latter applied starting at 2006. In our analysis we focus first on results from the CESM ensemble, and use CanESM to confirm the robustness of our results. For consistency, we use the period 1950-2100 for both ensembles.

We use daily output of minimum and maximum temperature at the surface (TASMIN and TASMAX) and average precipitation (PR) and compute a number of extreme metrics, all of them part of the Expert Team on Climate Change Detection and Indices suite (ETCCDI) (Alexander, 2016). All the metrics amount to annual statistics descriptive of daily output. They are:

- **TXx**: highest value over the year of daily maximum temperature (interpretable as the warmest day of the year);

- **TXn**: lowest value over the year of daily maximum temperature (interpretable as the coldest day of the year);

- **TNx**: highest value over the year of daily minimum temperature (interpretable as the warmest night of the year);

- **TNn**: lowest value over the year of daily minimum temperature (interpretable as the coldest night of the year);

- **Rx1Day**: precipitation amount falling on the wettest day of the year;

- **Rx5Day**: average daily amount of precipitation during the wettest 5 consecutive days (i.e., the wettest pentad) of the year.

We choose these indices as they reflect diverse aspects of daily extremes, but also because of a technical matter: their
definitions all result in the identification of what statistical theory of extreme values calls "block maxima" or "block minima"
(here the block is composed of the 365 days of the year). The same theory establishes that quantities so defined lend themselves
to be fitted by the Generalized Extreme Value distribution (GEVs) (Coles, 2001). GEV fitting allows us to apply the power of
inferential statistics, through which we can estimate return levels for any given period (e.g., the 20-, 50- or 100-year events),
and their confidence intervals. We will be looking at how these statistics – i.e., tail inference by a statistical approach that was
intended specifically for data-poor problems - change with the number of data points at our disposal, varying with ensemble
size, and asking if the statistical approach buys us any statistical power with respect to the simple "counting" of events across
the ensemble realizations.

## 3     Methods

### 3.1     Identifying the forced component

Milinski et al. (2020), use the ensemble mean computed on the basis of the full ensemble as a proxy for the true forced signal,
and analyze how its approximation gains in precision by using an increasingly larger ensemble size. By a bootstrap approach,
subsets of the full ensemble of a given size $n$ are sampled (without replacement) multiple times (in our analysis we will use 100
times), their mean (for the metric of interest) is computed, and the multiple replications of this mean are used to compute the
Root Mean Square Error (RMSEs) with respect to the full ensemble mean. Note that this bootstrap approach at estimating errors
is expected to become less and less accurate as $n$ increases, as was also noted in Milinski et al. (2020). For $n$ approaching the
size of the full ensemble, the repeated sampling from a finite population introduces increasingly stronger dependencies among
the samples, which share larger and larger numbers of members, therefore underestimating $\text{RMSE}(n)$. More problematically,
this approach would not be possible if we did not have a full-ensemble to exploit, and if our model was thought of having
different characteristics in variability than the models for which large ensembles are available. As a more realistic approach,
therefore, we assume that only 5 ensemble members are available and we abandon the bootstrap, proposing to use a different
method to infer the expected error as a function of

- An estimate of the ensemble variability that we compute on the basis of the 5 members available;

- The variable size $n$ of the ensemble that we are designing.

We will compare our inferred errors according to our method to the "true" errors that the availability of an actual large ensemble
allows us to compute.

It is a well known result of descriptive statistics that the standard error of the sample mean around the true mean decreases
as a function of $n$, the sample size, as in $\sigma/\sqrt{(n)}$ (see  Wehner (2000) for an application to GCM ensemble size computations
well before the advent of large ensembles). Here $\sigma$ is the true standard deviation of the population. In our case it is the standard
deviation of the ensemble, and we take as its true value what we compute as the full ensemble standard deviation on the basis

of 40 or 50 members for CESM and CanESM respectively, while we estimate it on the basis of only 5 members and show how our inferred errors compare to the true errors. We will also show in the first application of our method to global average trajectories how the error estimated by the bootstrap compares to ours and the true error, confirming that for $n$ approaching the full ensemble size the bootstrap underestimates the RMSE. For the remainder of our analysis we will not use the bootstrap approach further.

Since we are considering extreme metrics that can be modelled by a GEV, we also derive a range of return levels at a set of individual locations.

If a random variable $z$ (say the temperature of the hottest day of the year, TXx) is distributed according to a GEV distribution, its distribution function has the form:

$$G(z) = \exp\left\{-\left[1 - \xi\left(\frac{z-\mu}{\sigma}\right)\right]^{-1/\xi}\right\},$$

where three parameters $\mu, \sigma$, and $\xi$ determine its domain and its behavior. The domain is defined as $\{z : 1 + \xi(z-\mu)/\sigma > 0\}$, and the three parameters satisfy the following conditions: $-\infty < \mu < \infty$, $\sigma > 0$ and $-\infty < \xi < \infty$. $\mu, \sigma$, and $\xi$ represent the location, scale and shape parameter respectively, related to the mean, variability and tail behavior of the random quantity $z$.

If $p$ (say $p = 0.01$) is the tail probability to the right of level $z_p$ under the GEV probability density function, $z_p$ is said to be the return level associated to the $1/p$-year return period (100-yr return period in this example), and is given by:

$$z_p = \begin{cases} \mu - \frac{\sigma}{\xi}[1 - \{-\log(1-p)\}^{-\xi}], & \text{for } \xi \neq 0 \\ \mu - \sigma\log\{-\log(1-p)\}, & \text{for } \xi = 0. \end{cases}$$

Thus $z_p$ in our example represents the temperature in the hottest day of the year expected to occur only once every 100 years (in a stationary climate) or with $0.01$ probability every year (a definition more appropriate in the case of a transient climate).

We estimate the parameters of the GEVs, and therefore the quantities that are function of them, like $z_p$ and their confidence intervals by maximum likelihood, using the R package `extRemes`[1].

Because of the availability of multiple ensemble members we can choose a narrow window along the simulations (we choose 11 years) to satisfy the requirement of stationarity that the standard GEV fit postulates. We perform separate GEV fits centered around several dates along the simulation i.e., 2000, 2050, 2095[2] (the last chosen to allow extracting a symmetric window at the end of the simulations). The GEV parameters are estimated separately for a range of ensemble sizes $n$ up to the full size available (for each $n$ we concatenate 11 years from the first $n$ members of the ensemble, obtaining a sample of $11 * n$ values, and for each value of $n$ the same subset of members is used across all metrics, locations, times and return periods). On the basis of those estimates we compute return levels and their confidence intervals for several return periods $X$, $X = 2, 5, 10, 20, 50, 100$ (expressed in years) and assess when the estimates of the central value converge and what the trade-off is between sample size and width of the confidence interval. Lastly, we can use a simple counting approach, based

---

[1] Available from `https://CRAN.R-project.org/package=extRemes`
[2] Since some of the simulations end at 2099 this becomes 2094 in such cases

on computing the empirical cumulative distribution function from the same sample, to determine those same $X$-year events. I.e., after computing the empirical CDF we choose the value that leaves to its right no more than $p * n * 11$ data points, where $p$ is the tail probability corresponding to the $1/p = X$-year return period as defined above. The comparison will verify if fitting a GEV allows to achieve an accurate estimate using a smaller ensemble size than the empirical approach (where accurate is defined as close to the estimate obtained by the full ensemble).

We perform the analysis for a set of individual locations (i.e., grid-points), as for most extreme quantities there would be little value in characterizing very rare events as means of large geographical regions. Figure C9 shows the 15 locations that we chose with the goal of testing a diverse set of climatic conditions.

## 3.2 Characterizing internal variability

Recognizing the importance of characterizing variability besides the signal of change, we ask how many ensemble members are required to fully characterize the size of internal variability and its possible changes over the course of the simulation due to increasing anthropogenic forcing. Process based studies are suited to tackle the question of how and why changes in internal variability manifest themselves in transient scenarios (Huntingford et al., 2013), while here we simply describe the behavior of a straightforward metric, the within-ensemble standard deviation. We look at this quantity at the grid-point scale and we investigate how many ensemble members are needed to robustly characterize the full ensemble behavior, which here again we assume to be representative of the true variability of the system. This translates into two separate questions. First, for a number of dates along the simulation spanning the $20^{\text{th}}$ and $21^{\text{st}}$ centuries, we ask how many ensemble members are needed to estimate an ensemble variance that is not statistically significantly different from that estimated on the basis of the full ensemble (note that we have implicitly answered this by verifying that, at least for the computation of the expected RMSE in Section 3.1, plugging in an estimate of $\sigma$ based on 5 members appears to be accurate). Second, we first detect changes in variance between all possible pairs of these dates on the basis of the full ensemble, and we then ask how many ensemble members are needed to detect the same changes. We use F-tests to determine significant differences in variance, and since we apply them at each grid point we adopt a method for controlling the False Discovery Rate described for environmental applications in Ventura et al. (2004) and Wilks (2016) as a way to correct for multiple testing fallacies.

## 4 Results

In the following presentation of our main findings we choose two representative metrics, TNx (warmest night of the year) and Rx5Day (average rainfall amount during the 5 wettest days of the year) using the 40-member CESM ensemble. In the appendix we include the same type of results for the additional metrics considered and the 50-member CanESM ensemble. We will discuss if and when the results presented in this section differ from those shown in the appendix.

## 4.1 Identifying the forced component

We start from time series of annual values of globally averaged TNx and Rx5Day (Figure 1, top panels). We compute them for each ensemble member separately, and average them over $n$ ensemble members as the ensemble size $n$ increases, applying the bootstrap approach and computing $\mathrm{RMSE}(n)$ (see Section 3.1) at every year along the simulation.

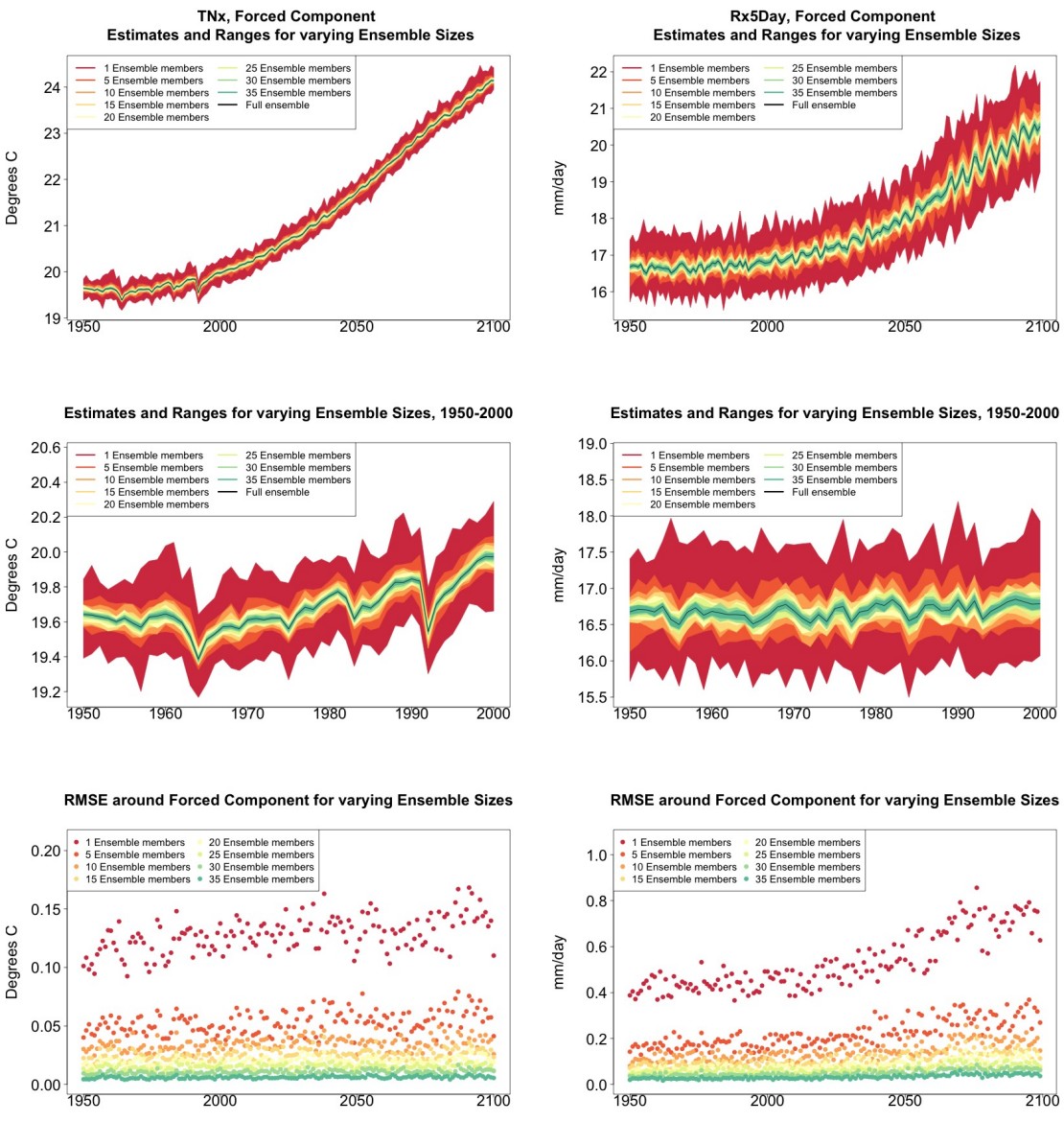

**Figure 1.** Time series for TNx (warmest night of the year, left) and Rx5Day (average daily amount during the 5 consecutive wettest days of the year, right) showing how the estimate of the forced component of their global mean trajectories over the period 1950-2100 changes when averaging an ensemble of increasing size. Top row shows the entire time series. Middle row zooms into the relatively flatter period of 1950-2000, so that the y-axis range allows a clearer assessment of the relative size of the uncertainty ranges for different sizes of the ensembles. The ranges are determined by bootstrapping. Bottom row plots the bootstrapped RMSE for every year and each ensemble size.

As Figure 1 indicates, for both quantities the marginal effect of increasing the ensemble size by 5 members is not constant but rather decreases as the ensemble size increases. This is qualitatively visible in the evolution of the ranges in the panels of

the first two rows, and is measured along the y-axis of the plots along the bottom row, where $\text{RMSE}(n)$ for increasing $n$ is shown (each $n$ corresponding to a different color).

|  | 1953 (F) | 1953 (F-5) | 1953 (B) | 2000 (F) | 2000 (F-5) | 2000 (B) |
|---|---|---|---|---|---|---|
| n=1 | 0.10 (0.08,0.13) | 0.09 (0.07,0.17) | 0.10 | 0.14 (0.11,0.18) | 0.12 (0.09,0.17) | 0.14 |
| n=5 | 0.05 (0.04,0.06) | 0.04 (0.03,0.06) | 0.04 | 0.06 (0.05,0.08) | 0.05 (0.04,0.08) | 0.05 |
| n=10 | 0.03 (0.03,0.04) | 0.03 (0.02,0.04) | 0.03 | 0.04 (0.04,0.06) | 0.04 (0.03,0.05) | 0.04 |
| n=15 | 0.03 (0.02,0.03) | 0.02 (0.02,0.03) | 0.02 | 0.04 (0.03,0.05) | 0.03 (0.02,0.04) | 0.03 |
| n=20 | 0.02 (0.02,0.03) | 0.02 (0.02,0.03) | 0.02 | 0.03 (0.03,0.04) | 0.03 (0.02,0.04) | 0.02 |
| n=25 | 0.02 (0.02,0.03) | 0.02 (0.01,0.02) | 0.01 | 0.03 (0.02,0.04) | 0.02 (0.02,0.03) | 0.02 |
| n=30 | 0.02 (0.02,0.02) | 0.02 (0.01,0.02) | 0.01 | 0.03 (0.02,0.03) | 0.02 (0.02,0.03) | 0.01 |
| n=35 | 0.02 (0.01,0.02) | 0.02 (0.01,0.02) | 0.00 | 0.02 (0.02,0.03) | 0.02 (0.02,0.03) | 0.01 |
|  | 2050 (F) | 2050 (F-5) | 2050 (B) | 2097 (F) | 2097 (F-5) | 2097 (B) |
| n=1 | 0.12 (0.09,0.15) | 0.15 (0.12,0.21) | 0.11 | 0.15 (0.12,0.19) | 0.13 (0.10,0.17) | 0.15 |
| n=5 | 0.05 (0.04,0.07) | 0.07 (0.05,0.09) | 0.05 | 0.07 (0.05,0.09) | 0.06 (0.04,0.08) | 0.06 |
| n=10 | 0.04 (0.03,0.05) | 0.05 (0.04,0.07) | 0.03 | 0.05 (0.04,0.06) | 0.04 (0.03,0.06) | 0.04 |
| n=15 | 0.03 (0.02,0.04) | 0.04 (0.03,0.05) | 0.02 | 0.04 (0.03,0.05) | 0.03 (0.03,0.05) | 0.03 |
| n=20 | 0.03 (0.02,0.03) | 0.03 (0.03,0.05) | 0.02 | 0.03 (0.03,0.04) | 0.03 (0.02,0.04) | 0.03 |
| n=25 | 0.02 (0.02,0.03) | 0.03 (0.02,0.04) | 0.01 | 0.03 (0.02,0.04) | 0.03 (0.02,0.03) | 0.02 |
| n=30 | 0.02 (0.02,0.03) | 0.03 (0.02,0.04) | 0.01 | 0.03 (0.02,0.04) | 0.02 (0.02,0.03) | 0.01 |
| n=35 | 0.02 (0.02,0.03) | 0.03 (0.02,0.04) | 0.01 | 0.03 (0.02,0.03) | 0.02 (0.02,0.03) | 0.01 |

**Table 1.** Error in estimating the global mean of TNx as simulated by the CESM ensemble: Values of the RMSE in approximating the full ensemble mean by the individual runs (first row, $n = 1$), and by ensembles of increasingly larger sizes (from 5 to 35, along the remaining rows). The values along the columns labelled as "(F)" apply the formula by using the "truth" for $\sigma_t$ which we take as the standard deviation of the ensemble computed over all 40 members. We compare these estimates to those derived by plugging into the formula a value of $\sigma_t$ estimated by a subset of 5 ensemble members, and 5 years around the year $t$ considered (columns labelled by "(F-5)"). We also show estimates obtained by the bootstrap approach in the columns labeled by "(B)". These can also be compared to the truth "(F)" and we underline the values that are not consistent with it and its confidence interval, thus pointing out for which ensemble sizes the bootstrap underestimates the RMSE. Results are shown for four individual years ($t$) along the simulation (column-wise), since $\sigma_t$ varies along its length.

This behavior is to be expected, as we know the RMSE of a mean behaves in inverse proportion to the square root of the size of the sample from which the mean is computed, but the actual behavior shown in the plots and Tables 1 and 2 along columns (B) could be misleading, as the variability of the largest means (largest in sample size $n$) could be underestimated by the bootstrap (see Section 3.1). Furthermore, this assessment would not be possible if all we had was a 5-member ensemble for our model. We can therefore compute the formula for the standard error of a mean, $\sigma/\sqrt{n}$ (see Section 3.1), using the full ensemble

to estimate $\sigma$, which we assume to be the true standard deviation of the ensemble. We then repeat the estimation by substituting

a value of $\sigma$ derived using only 5 ensemble members. Table 1 shows RMSEs for the same increasing values of $n$, evaluated at 4 different dates along the simulation, as we expect $\sigma$ to change. The columns labeled (F) apply the formula using the full ensemble size to estimate $\sigma_t$, $t = 1953, 2000, 2050, 2097$. The values along these columns represent the truth against which we compare our estimates based on the first 5 ensemble members (columns labelled (F-5)), and the estimates by the bootstrap (columns labelled as (B)). Importantly for the accuracy of our results, when we use only the first 5 members we increase the sample size by using a window of 5 years around each date $t$. We are aware that this could introduce autocorrelation within the sample values, but the comparison of these results to the truth shows that the estimated values based on the smaller ensemble are an accurate approximation of it, always being consistent with the 95% confidence intervals (shown in parentheses). From the table entries we can assess that the bootstrap estimation is inaccurate once the ensemble size exceeds about $15 - 20$ out of $40$ available (we have colored the cells when this happens grey to underline this behavior). For the larger sizes, the RMSE estimated by the bootstrap falls in all cases to the left of the confidence interval under the (F) column, confirming the tendency to underestimate the RMSE. However, the estimates of RMSE associated with an ensemble size of 10 or 15 already quantifies a high degree of accuracy for the approximation of the ensemble mean of the full 40-member ensemble: those RMSEs for TNx are on the order of $0.02°\mathrm{C} - 0.04°\mathrm{C}$.

Table 2 reports the same analysis results for the precipitation metric, Rx5Day. The same general message can be drawn, with too narrow estimates by the bootstrap approach for ensemble sizes starting at around 20 or 25 members. Even in this case however the estimates for the RMSE is on the order of $0.1 - 0.2\mathrm{mm/day}$ for Rx5Day once the ensemble size exceeds 10.

The lessons learned here are that

1. For both metrics, an accurate estimate of $\sigma_t$, i.e., the instantaneous model internal variability at global scale, is possible using 5 ensemble members (and a window of 5 years around the year $t$ of interest);

2. if the formula for computing the RMSE on the basis of a given sample size is adopted, and that estimate for $\sigma_t$ is plugged in, it is possible, on the basis of an existing 5-member ensemble, to accurately estimate the required ensemble size to identify the forced component within a given tolerance for error. Of course, the size of this tolerance will change depending on the specific application.

We note here that the calculation of the RMSE for increasing ensemble sizes is straightforward, once $\sigma_t$ is estimated. Even more straightforward is the calculation of the expected "gain" in narrowing the RMSE. A simple ratio calculation shows that for $n$ spanning the range 5 to 45 (relevant sizes for our specific examples) the reduction in RMSE follows the sequence $\{100 * 1/\sqrt{n}\}_{n=1,5,...,45}$. Thus, compared to a single model run's RMSE, we expect the RMSE of mean estimates derived by ensemble sizes of $n = 5, 10, 20, 35 \, or \, 45$ to be 45%, 32%, 22%, 17% or 15% of that, respectively.

We assess how the results of the formula compare to the actual error by considering the difference between the smaller size ensemble means and the truth (the full ensemble mean), year by year and comparing that difference to twice the expected RMSE derived by the formula, i.e., $2\sigma/sqrt(n)$, akin to a 95% probability interval for a normally distributed quantity. Here is where our approximation, and the use of possibly autocorrelated samples in the estimates of $\sigma$ could possibly reveal shortcomings. Figure 2, for global averages of the two same quantities, shows the ratios of actual error vs. the 95% probability bound,

|  | 1953 (F) | 1953 (F-5) | 1953 (B) | 2000 (F) | 2000 (F-5) | 2000 (B) |
|---|---|---|---|---|---|---|
| n=1 | 0.40 (0.33,0.51) | 0.33 (0.26,0.46) | 0.40 | 0.50 (0.41,0.64) | 0.45 (0.35,0.63) | 0.49 |
| n=5 | 0.18 (0.15,0.23) | 0.15 (0.12,0.21) | 0.16 | 0.22 (0.18,0.29) | 0.20 (0.16,0.28) | 0.15 |
| n=10 | 0.13 (0.10,0.16) | 0.10 (0.08,0.15) | 0.11 | 0.16 (0.13,0.20) | 0.14 (0.11,0.20) | 0.11 |
| n=15 | 0.10 (0.08,0.13) | 0.09 (0.07,0.12) | 0.08 | 0.13 (0.11,0.17) | 0.12 (0.09,0.16) | 0.09 |
| n=20 | 0.09 (0.07,0.11) | 0.07 (0.06,0.10) | 0.07 | 0.11 (0.09,0.14) | 0.10 (0.08,0.14) | 0.07 |
| n=25 | 0.08 (0.07,0.10) | 0.07 (0.05,0.09) | 0.05 | 0.10 (0.08,0.13) | 0.09 (0.07,0.13) | 0.06 |
| n=30 | 0.07 (0.06,0.09) | 0.06 (0.05,0.08) | 0.03 | 0.09 (0.07,0.12) | 0.08 (0.06,0.11) | 0.05 |
| n=35 | 0.07 (0.06,0.09) | 0.06 (0.04,0.08) | 0.02 | 0.08 (0.07,0.11) | 0.08 (0.06,0.11) | 0.03 |
|  | 2050 (F) | 2050 (F-5) | 2050 (B) | 2097 (F) | 2097 (F-5) | 2097 (B) |
| n=1 | 0.55 (0.45,0.71) | 0.52 (0.40,0.72) | 0.54 | 0.77 (0.63,0.98) | 0.65 (0.51,0.90) | 0.76 |
| n=5 | 0.25 (0.20,0.32) | 0.23 (0.18,0.32) | 0.23 | 0.34 (0.28,0.44) | 0.29 (0.23,0.40) | 0.33 |
| n=10 | 0.17 (0.14,0.22) | 0.16 (0.13,0.23) | 0.17 | 0.24 (0.20,0.31) | 0.21 (0.16,0.29) | 0.19 |
| n=15 | 0.14 (0.12,0.18) | 0.13 (0.10,0.19) | 0.14 | 0.20 (0.16,0.25) | 0.17 (0.13,0.23) | 0.15 |
| n=20 | 0.12 (0.10,0.16) | 0.12 (0.09,0.16) | 0.09 | 0.17 (0.14,0.22) | 0.15 (0.11,0.20) | 0.10 |
| n=25 | 0.11 (0.09,0.14) | 0.10 (0.08,0.14) | 0.08 | 0.15 (0.13,0.20) | 0.13 (0.10,0.18) | 0.10 |
| n=30 | 0.10 (0.08,0.13) | 0.09 (0.07,0.13) | 0.04 | 0.14 (0.11,0.18) | 0.12 (0.09,0.16) | 0.07 |
| n=35 | 0.09 (0.08,0.12) | 0.09 (0.07,0.12) | 0.03 | 0.13 (0.11,0.17) | 0.11 (0.09,0.15) | 0.05 |

**Table 2.** Same as Table 1, for Rx5Day simulated by CESM.

indicating the 100% level by a horizontal line for reference. As can be assessed, the actual error is in most cases much smaller than the 95% bound (as it is not reaching the 100% line in the great majority of cases), and we see that only occasionally the actual error spikes above the 95% bound for individual years, consistent with what would be expected of a normally distributed error. This behavior is consistently true for ensemble sizes larger than $n = 5$.

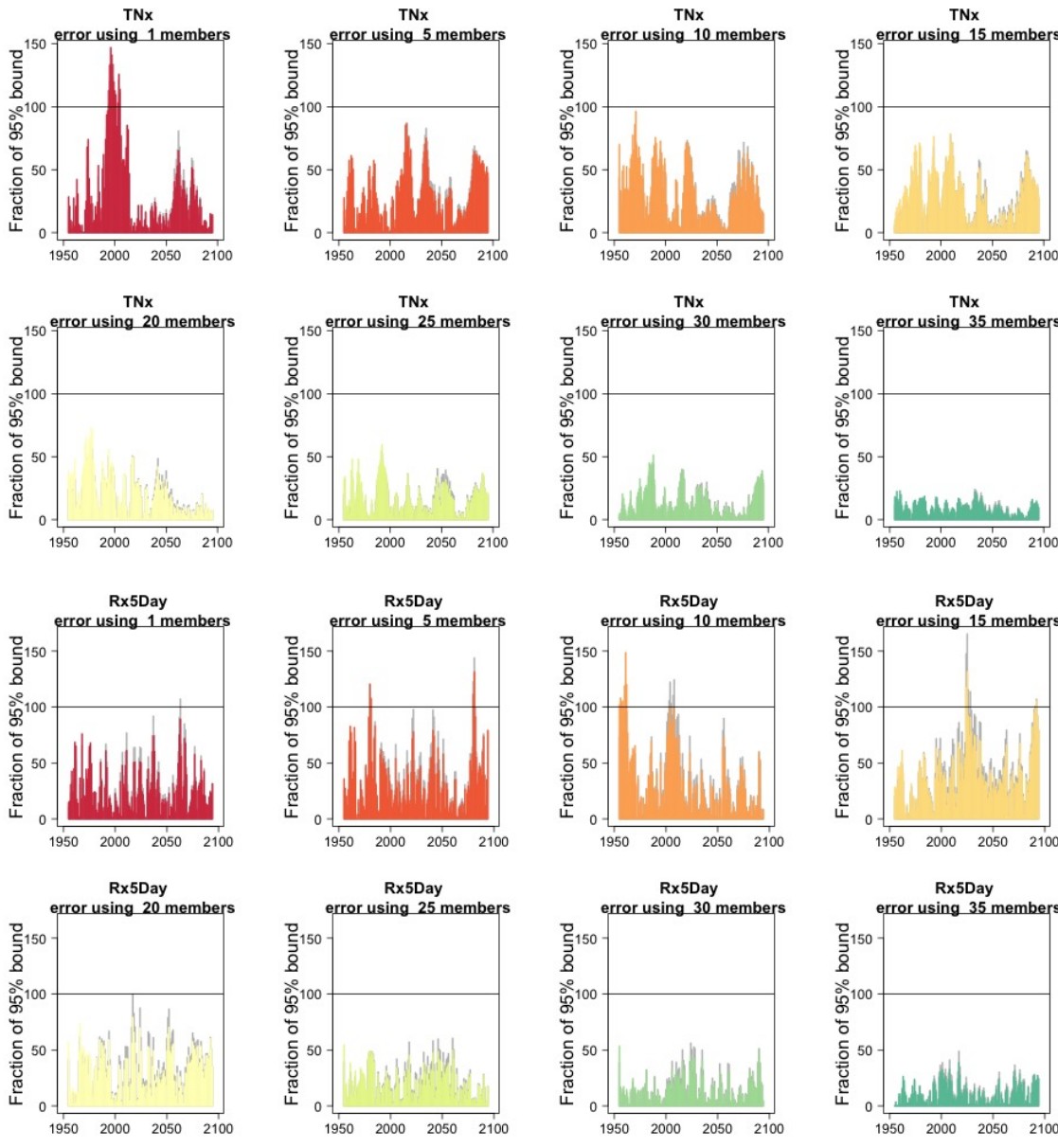

**Figure 2.** In each plot, for each year, the height of the bar gives the error in the estimate of the forced component (defined as the mean of the entire ensemble) as a percentage of the expected 95% probability bound, estimated by the formula $2\sigma_t/\sqrt{n}$ with $n$ the ensemble size. Grey bars if $\sigma_t$ is the truth (i.e., estimated using the whole ensemble), colored bars if $\sigma_t$ is estimated using only 5 ensemble members (but using 5 years around each year). Each plot corresponds to a different and increasing ensemble size: 1,5,10,15,20,25,30,35. The top two rows of plots are for TNx; the bottom two rows are for Rx5Day. All results are for the CESM ensemble.

In the appendix we report the results of applying the same analysis to the rest of the indices. We cannot show all results, but
we tested country averages, zonal averages, land- and ocean-areas averages separately, confirming that the qualitative behavior
we assess here is common to all these other scales of aggregation.

Here we go on to show how the same type of analysis can be applied at the grid-point scale, and still deliver an accurate
bound for the error in approximating the forced component. For the grid scale analysis, we define as the forced component
anomalies by mid- and end-of-century (compared to a baseline) obtained as differences between 5-year averages: 2048-2052
and 2096-2100 vs 2000-2005. We use only 5 members (and as before, a 5-year window for each to increase the sample size)
to estimate the ensemble standard deviation of the two anomalies (separately, as that standard deviation may differ at mid-
and end-of-century) at each grid-point, and compare the actual error when approximating the "true" anomalies (i.e., those
obtained on the basis of the full ensemble) by increasingly larger ensemble sizes to the 95% confidence bound, calculated by
the formula $2\sigma_i^c/\sqrt{n}$ (here $i$ indicates the grid cell, and $c$ indicates the period in the century considered for the anomalies). In
Figures 3 and 4 we show fields of the ratio of actual error to the 95% bound, as the ensemble size increases. Red areas are
ones where the ratio exceeds 100%, i.e., where the bound was exceeded by the actual error, which we would expect to happen
only over 5% of the surface. As can be gauged even by eye, only small and sparse areas appear where the actual error exceeds
the expected error, especially if land regions are considered (incidentally,these indices have been mostly used over land areas,
as input to impact analyses). The prevalence of red areas over the oceans could be due to an underestimation of $\sigma_i^c$ linked
to the use of the 5-year windows and the autocorrelation possibly introduced, consistent with ocean quantities having more
memory than land quantities, but we do not explore that further here. Over the majority of the Earth's surface, particularly
when errors are estimated for ensemble sizes of 20 or more, the bound is a good measure providing an accurate estimate of
the error behavior according to normal distribution theory. Tables B1 through B4 in the appendix confirm this by reporting
percentages of surface areas (distinguishing global, land-only or ocean-only aggregation) where the actual error exceeds the
bound, i.e., where the values of the fields exceed 100%. As can be assessed for all metrics considered in our analysis, 20
ensemble members consistently keep such fraction at or under 5% for the CESM model ensemble, while the coarser resolution
CanESM requires 25 ensemble members for that to be true.

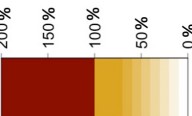

**Figure 3.** Error in the estimation of anomalies in TNx by mid-century (top two rows) and end-of-century (bottom two rows) from the CESM ensemble. In each plot, for increasing ensemble sizes, the color of each grid-point indicates the ratio (as a percentage) between actual error and the 95% confidence bound. Values less than 100% indicate that the actual error in estimating the anomaly at that location is contained within the bound. The color scale highlights in dark red the values above 100%, whose total fraction is reported in Table B1.

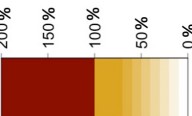

**Figure 4.** Error in the estimation of anomalies in Rx5Day by mid-century (top two rows) and end-of-century (bottom two rows) from the CESM ensemble. In each plot, for increasing ensemble sizes, the color of each grid-point indicates the ratio (as a percentage) between actual error and the 95% confidence bound. Values less than 100% indicate that the actual error in estimating the anomaly at that location is contained within the bound. The color scale highlights in dark red the values above 100%, whose total fraction is reported in Table B3.

Overall, these results attest to the fact that we can use a small ensemble of 5 members to estimate the population standard deviation, and plug it into the formula for the standard error of the sample mean as a function of sample size. Imposing a ceiling for this error allows us then to determine how large an ensemble should be, in order to approximate the forced component to the desired level of accuracy. This holds true across the range of spatial scales afforded by these models, from global means all the way to grid-point values.

## 4.2 GEV results

As explained in Section 3, the extreme metrics we chose can be fit by a Generalized Extreme Value distribution, and return levels for arbitrary return periods derived, with their confidence interval. In this section we ask two questions:

1. How many ensemble members are needed for the estimates to stabilize and the size of the confidence interval not to change in a substantial way?

2. Is there any gain in applying GEV fitting rather than simply "counting" rare events across the ensemble?

Here we show results for our two main metrics, choosing two different locations for each. These results are indicative of what happens across the rest of locations (see Figure C9), for the other metrics and the other model considered (see Appendix for a sampling of those).

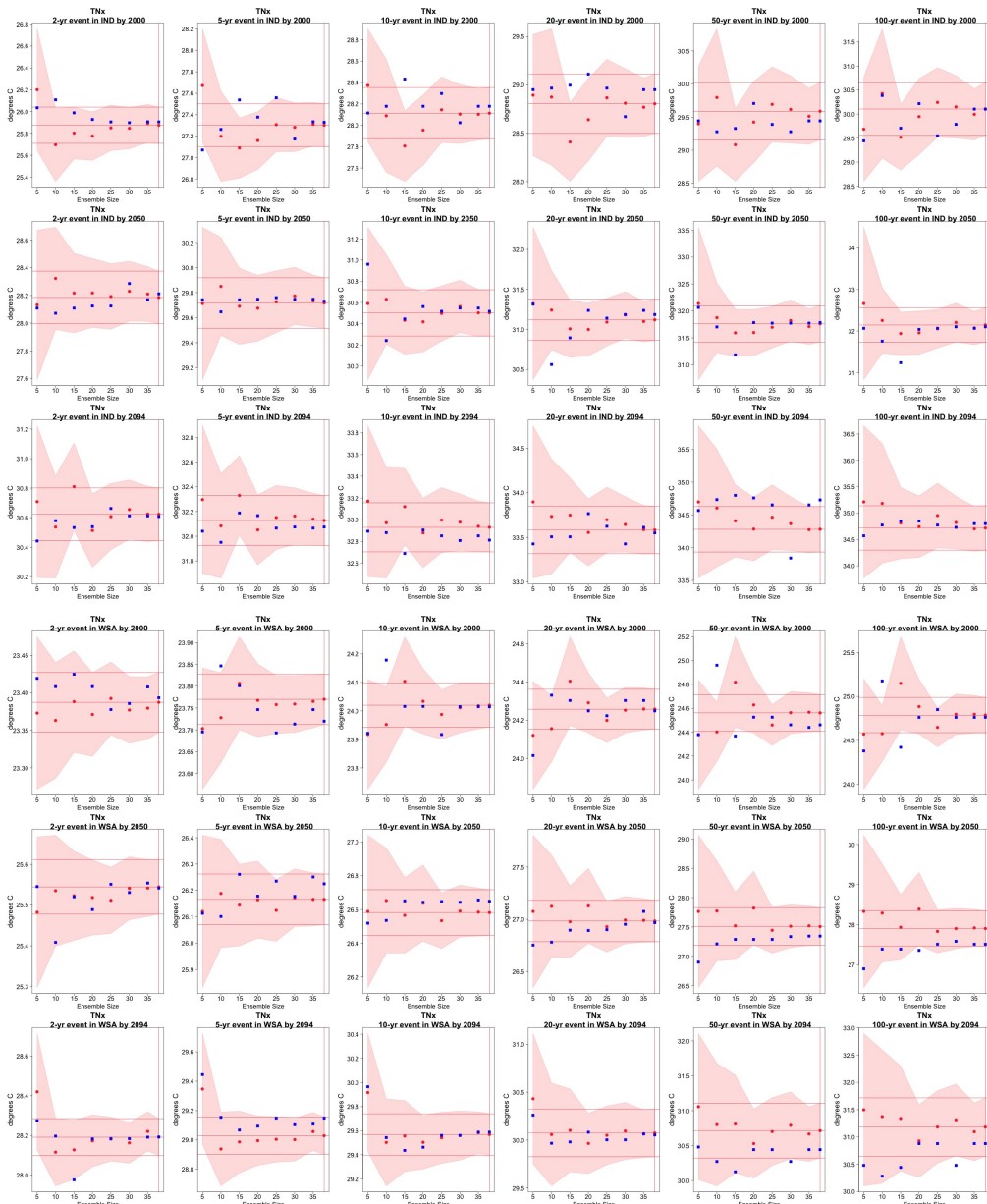

**Figure 5.** Return Levels for TNx at (row-wise) 2000, 2050 and 2100 for (column-wise) 2-, 5-, 10-, 20-, 50-, 100-year return periods, based on estimating a GEV by using 11-yr windows of data around each date. In each plot, for increasing ensemble sizes along the x-axis (from 5 to the full ensemble, 40), the red dots indicate the central estimate, and the pink envelope represents the 95% confidence interval. The estimates based on the full ensemble (central and confidence interval bounds), which we consider the truth, are also drawn across the plot for reference, as horizontal lines. The blue dots in each plot show return levels for the same return periods estimated by counting, i.e., computing the empirical cumulative distribution function of TNx on the basis of the $n \times 11$ years in the sample, where $n$ is the ensemble size. Note that in the 100-yr return level plots the first such dot is obtained by interpolation of the last two values of the CDF, since the sample size is less than 100 (see text). The first three rows show results for a location in India while the following three rows show results for a location in Western South America (see Figure C9).

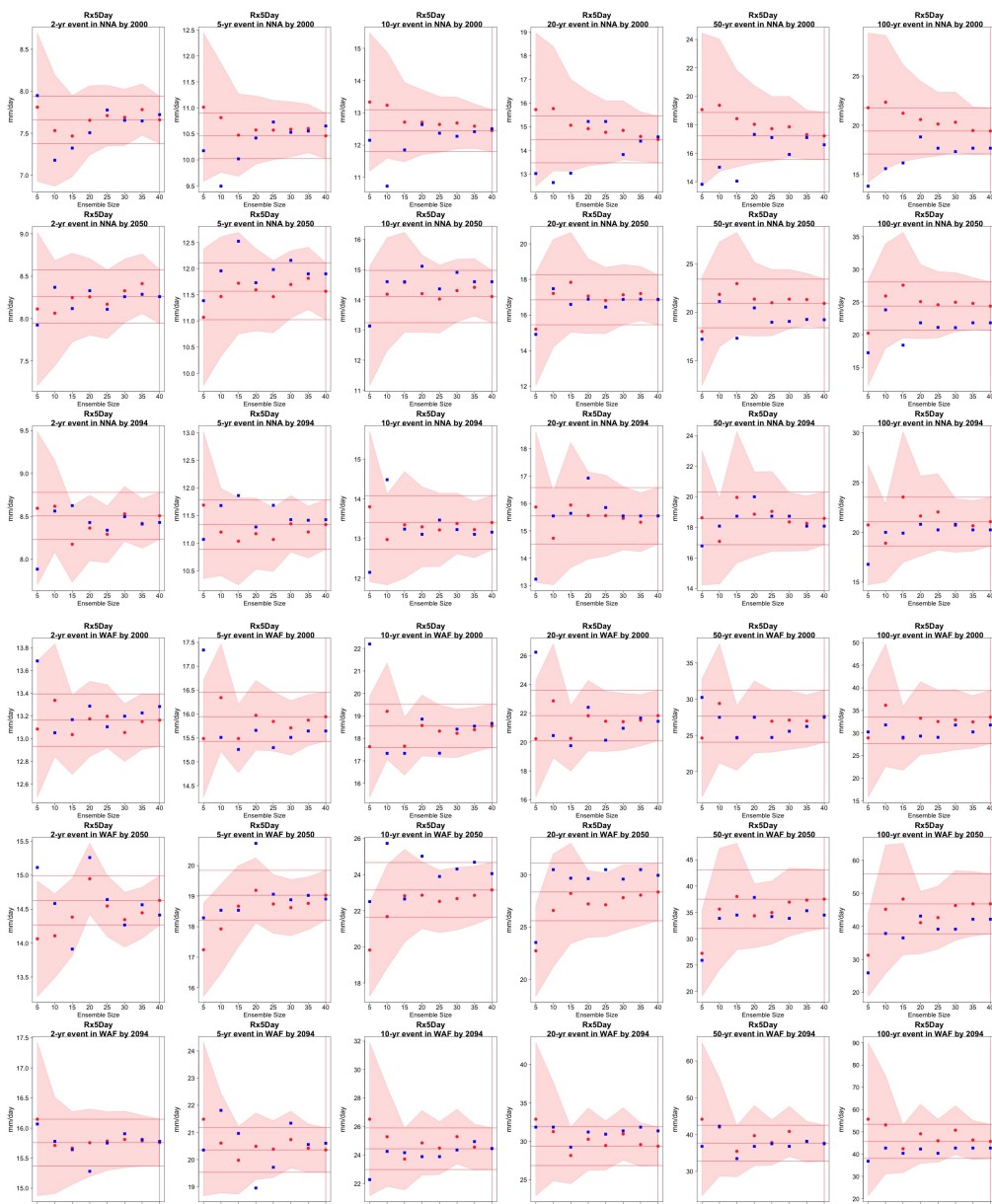

**Figure 6.** Return Levels for Rx5Day at (row-wise) 2000, 2050 and 2100 for (column-wise) 2-, 5-, 10-, 20-, 50-, 100-year return periods, based on estimating a GEV by using 11-yr windows of data around each date. In each plot, for increasing ensemble sizes along the x-axis (from 5 to the full ensemble, 40), the red dots indicate the central estimate, and the pink envelope represents the 95% confidence interval. The estimates based on the full ensemble (central and confidence interval bounds), which we consider the truth, are also drawn across the plot for reference, as horizontal lines. The blue dots in each plot show return levels for the same return periods estimated by counting, i.e., computing the empirical cumulative distribution function of Rx5Day on the basis of the $n \times 11$ years in the sample, where $n$ is the ensemble size. Note that in the 100-yr return level plots the first such dot is obtained by interpolation of the last two values of the CDF, since the sammple size is less than 100 (see text). The first three rows show results for a location in Northern North America while the following three rows show results for a location in West Africa (see Figure C9).

Figures 5 and 6 and several more in the appendix compare for each of the six return levels (along the columns), and across the three projection dates (along the rows), the behavior of the GEV central estimates (red dots) and 95% confidence intervals (pink envelope, calculated according to the maximum likelihood approach) based on an increasing ensemble size (along the x-axis) to the "truth" obtained by the full ensemble, which is drawn as a reference across each plot as horizontal lines. We also compute estimates of the central quantities based on computing the empirical cumulative distribution function (see Section 3.1) from the same data points. These empirical estimates are added to each plot as blue dots for each of the ensemble sizes considered. Note that also for these empirical estimates we use 11-year windows for each ensemble member, so that the sample is exactly the same as that used for fitting the corresponding GEV. Only the left-most blue dot in the 100-yr return level panels is based on interpolating the values of the empirical CDF, which for that sample size is based on only 55 data points. We first observe that in the great majority of cases the central estimate settles within the "true" confidence interval as soon as the ensemble comprises 15 or 20 members. This is true for both model ensembles, i.e., both when the truth is identified through 40 and through 50 ensemble members, as the corresponding plots in the appendix confirm. Therefore, if all that concerns us is the central estimate, an ensemble of 20 members, from which we sample 11-yr windows to enrich the sample size, delivers an estimate of the "truth" within its confidence interval. When an estimate of the uncertainty is concerned, however, the truth remains by definition an unattainable target, as the size of the confidence intervals is always bound to decrease for larger sample sizes. The behavior of the confidence intervals for the return level estimates in the plots, however, suggests that there might be only marginal gains for ensemble sizes beyond 30, for both models. The value of this general result will benefit from an analysis of larger ensembles. In addition, the value of increasing the sample size should always be judged on the basis of the actual size of the 95% confidence intervals in the units of the quantity of interest, and what that size means for managing risks associated with these extremes. This is an aspect that, however, goes beyond the scope of our work. As for the results of the empirical counting approach, i.e., the blue point estimates, we can assess that in the majority of cases, but not across the board when we look closely to all the plots in the appendix, they do not deviate significantly from the central estimates based on fitting the GEV using the same sample size. However, while the latter can provide a measure of uncertainty through confidence intervals, the estimates based on counting events do not come with uncertainty bounds. Another advantage of using the GEV is the ability to extrapolate to even more rare events than the ensemble size would allow to robustly estimate, not underplaying the risk of statistical extrapolations as a general rule.

Further statistical precision could be attained by relaxing the quasi-stationarity assumption and extending the analysis period to contain a longer window of years. Exchanging time for ensemble members however, when beyond a decade's worth, necessitates in most cases the inclusion of temporal covariates: for example, indicators of the phase and magnitude of major modes of variability known to affect the behavior of the atmospheric variables in question over multi-decadal scales. The inclusion of covariates of course adds another source of fitting uncertainty.

### 4.3 Characterizing internal variability

After concerning ourselves with the characterization of the forced component we turn to the complementary problem of characterizing internal variability. Rather than aiming at eliminating the effects of internal variability as we have done so far in the

estimation of a forced signal, we take here the opposite perspective, wanting to fully characterize its size and behavior over space and time. After all, the real world realization will not be akin to the mean of the ensemble, but to one of its members, and we want to be sure to estimate the range of variations such members may display. Thus, we ask how large the ensemble needs to be to fully characterize the variations that the full-size ensemble produces, which once again we take as the truth (as mentioned, the answer to this question can be seen as a systematic confirmation that 5 members are sufficient for the estimation of $\sigma$, one result that we only indirectly affirmed so far). We also ask how large an ensemble is needed to detect changes in the size of internal variability with increasing external forcing. Our definition of internal variability here is simply the size of the ensemble variance. Both these questions we tackle directly at the grid-point scale, as that answer can serve as an upper bound for the characterization of variability at coarser spatial scales. Figures 7 through 10 synthesize our findings for both these questions.

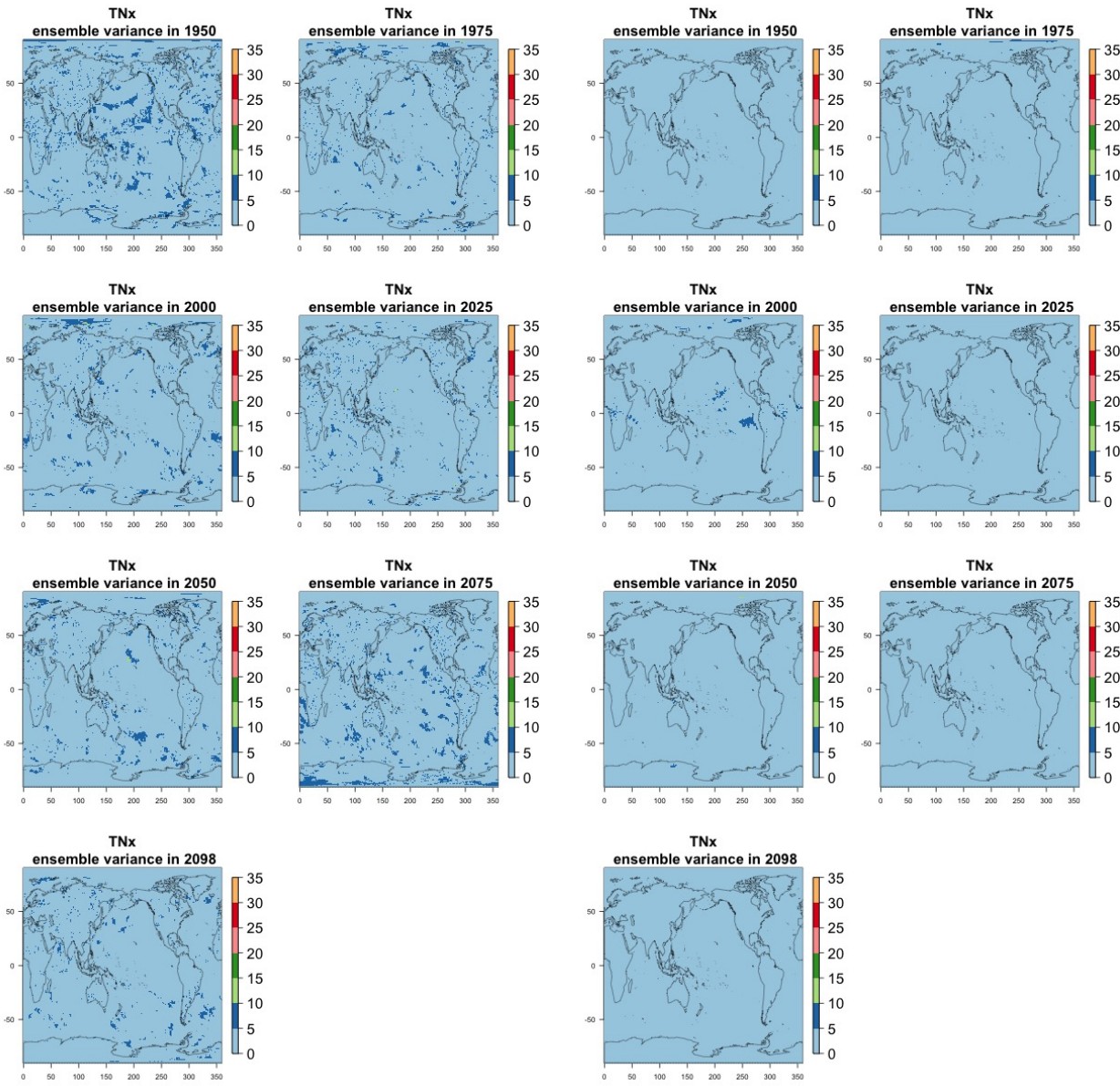

**Figure 7.** Estimating the ensemble variance for TNx: Each plot corresponds to a year along the simulation length (1950, 1975, 2000, 2025, 2050, 2075, 2100). The color indicates the number of ensemble members needed to estimate an ensemble variance at that location that is statistically indistinguishable from that computed on the basis of the full 40-member ensemble, using an F-test to test the null hypothesis of equality in variance. The results of the first two columns use only the specific year for each of the ensemble members. The results of the third and fourth columns enrich the samples by using 5 years around the specific date.

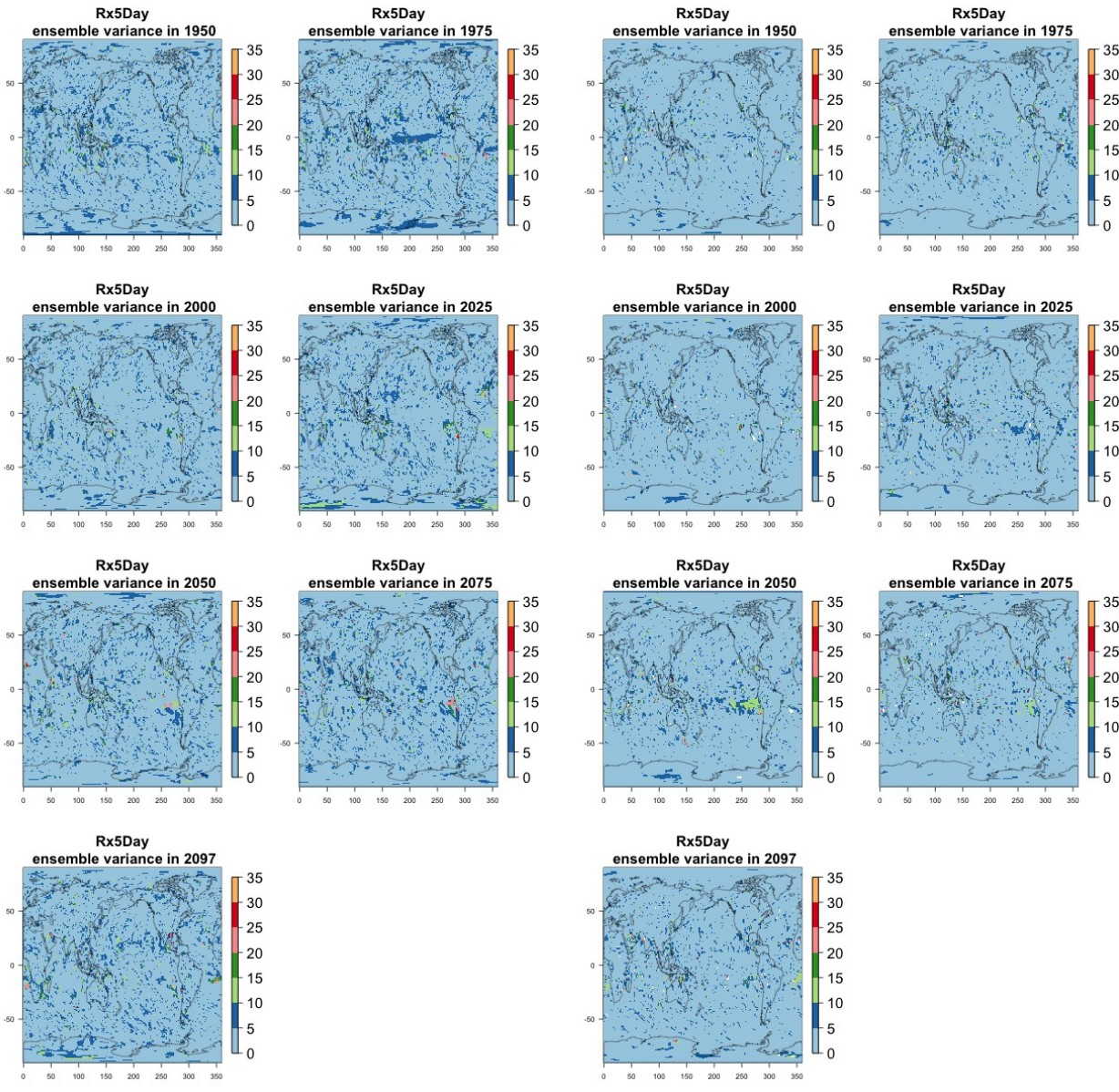

**Figure 8.** Estimating the ensemble variance for Rx5Day: Each plot corresponds to a year along the simulation length (1950, 1975, 2000, 2025, 2050, 2075, 2100). The color indicates the number of ensemble members needed to estimate an ensemble variance at that location that is statistically indistinguishable from that computed on the basis of the full 40-member ensemble, using an F-test to test the null hypothesis of equality in variance. The results of the first two columns use only the specific year for each of the ensemble members. The results of the third and fourth columns enrich the samples by using 5 years around the specific date.

The two columns on the left-hand side of Figure 7 show for several years along the simulation of TNx how many ensemble members are needed (denoted by the colors, see legends) in order to estimate an ensemble variance at each grid-point that is not statistically distinguishable from the same variance estimated by the full 40-member. We use a traditional F-test approach to test the null hypothesis of equality in variance. Note that we do this at various times along the length of the simulation (1950, 1975, 2000, 2025, 2050, 2075, 2100) because we account for the possibility that internal variability might change over its course with increasing external forcing, but for now we remain agnostic on this issue. For all times considered, 5 members are sufficient to estimate an ensemble variance indistinguishable, statistically, from that which would be estimated using the full ensemble at most grid-points over the Earth's surface, as the light blue color indicates. For some sparse locations, however, 10 members are needed to achieve the same type of accuracy. The same type of Figure for the precipitation metric, Figure 8, left two columns, confirms that for this noisier quantity a larger extent of the Earth's surface needs ensemble sizes of 10 or more to accurately estimate the behavior of the full ensemble variance. The two right-hand columns in Figure 7 show corresponding plots where now most of the Earth's surface only requires 5-members. This is the result of "borrowing strength" in the estimation of the ensemble variance by using a 5-year window around the date as we have done for the analysis of $\sigma_t$ in the previous sections. This solution addresses the problem of estimating the variance for both temperature and precipitation metrics, as Figure 8 confirms, reducing also for the latter the number of grid-points that require more statistical power to a noisy speckled pattern. Similar figures in the appendix attest to this remaining true for the other model and the remaining metrics as well. We note here that the patterns shown in some of these figures have indeed the characteristics of noise. To minimize that possibility we have applied a threshold for the significance of the p-values from the F-test obtained through the method that controls the False Discovery Rate (Ventura et al., 2004). The method has been shown to control for the false identification of significant differences "by chance" due to repeating statistical tests hundreds or thousands of times, as in our situation. The same method has been proved effective in particular for multiple testing over spatial fields, despite the presence of spatial correlation (Ventura et al., 2004; Wilks, 2016). We fix the false discovery rate to 5%.

Detecting changes in the size of the variance over time by comparing two dates over the simulation is a problem that we expect to require more statistical power than the problem of characterizing the size of the variance at a given point, as the difference between stochastic quantities is affected by larger uncertainty than the quantities individually considered, unless those are strongly correlated. Figure 9 shows the ensemble size required to detect the same changes in the ensemble variance of TNx that the full ensemble of 40-members detects. Each plot is at the intersection of a column and a row corresponding to two of the dates considered in the previous analysis, indicating that the solution applies to detecting a change in variance between those two dates, as the title of each plot specifies. Here again we use the F-test and the method for controlling the false discovery rate.

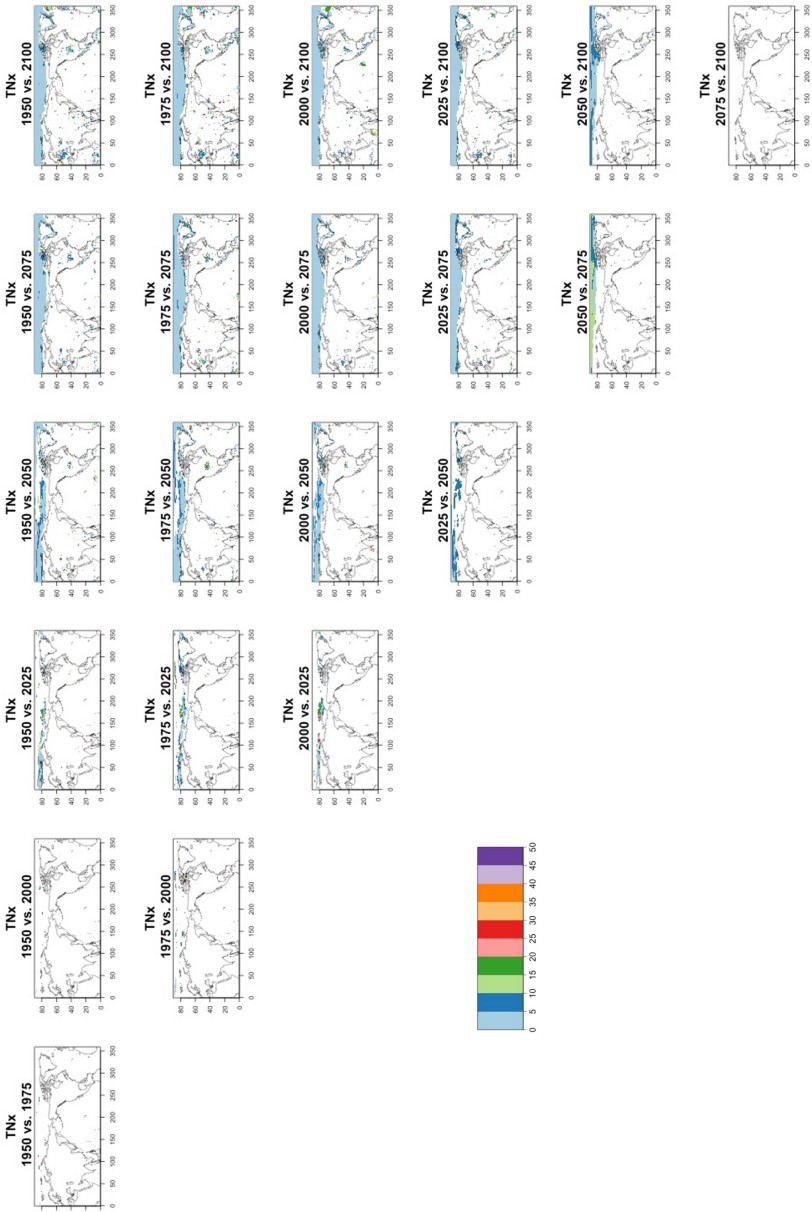

**Figure 9.** Estimating changes in ensemble variance for TNx: each plot corresponds to a pair of years along the simulation (same set of years as depicted in Figures 7 and 8 above). Colored areas are regions where on the basis of the full 40-member ensemble a significant change in variance was detected. The colors indicate the size of the smaller ensemble needed to detect the same change. Here the sampling size is increased by using 5 years around each date. We only show the Northern hemisphere as no region in the southern shows significant changes in variance for this quantity.

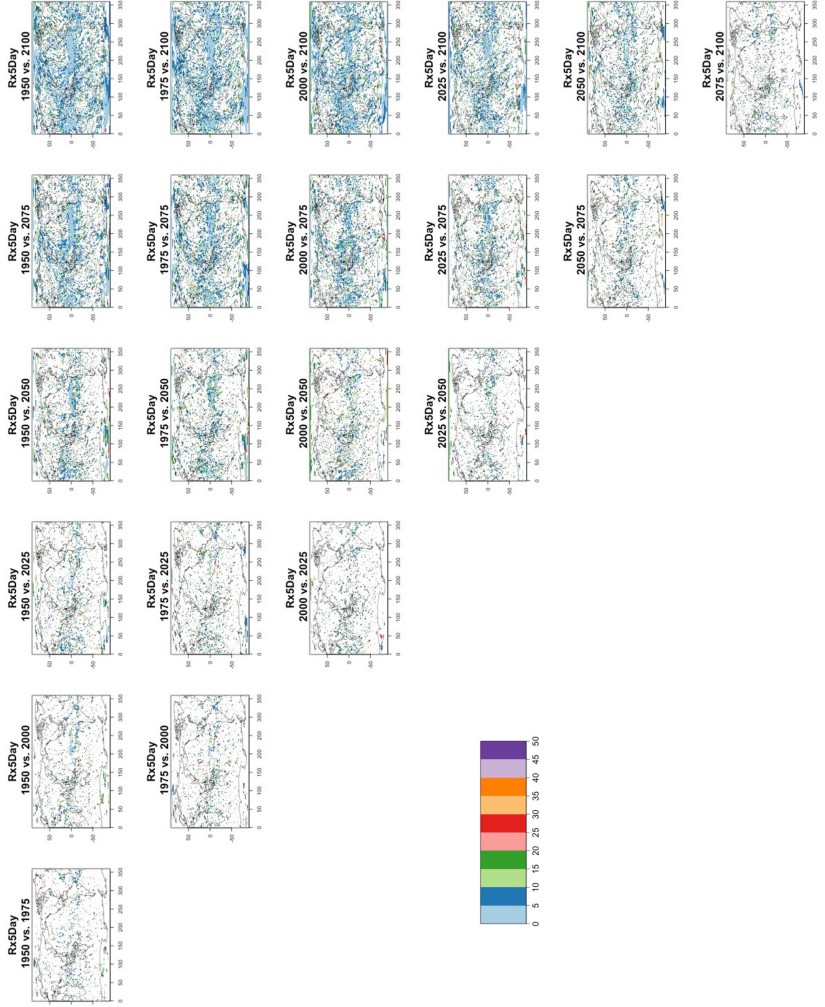

**Figure 10.** Estimating changes in ensemble variance for Rx5Day: each plot corresponds to a pair of years along the simulation (same set of years as depicted in Figures 7 and 8 above). Colored areas are regions where on the basis of the full 40-member ensemble a significant change in variance was detected. The colors indicate the size of the smaller ensemble needed to detect the same change. Here the sampling size is increased by using 5 years around each date.

Blank areas are regions where the full ensemble has not detected any changes in the ensemble variance at that location when comparing the two dates. Colored areas are regions where such change has been detected by the full ensemble, and the color indicates what (smaller) ensemble size is sufficient to detect the same change. Here as in the previous analysis a significant change is detected when the F-test for the ratio of the two variances that are being compared across time has a p-value smaller than the threshold determined by applying the false discovery rate method, and fixing the false discovery rate to 5%. These results are obtained by increasing the sample size using 5 years around the dates, as in the right-hand columns of Figures 7 and 8. In the case of TNx, a metric based on daily minimum temperature, the changes are confined to the Arctic region and in most cases the ensemble size required is again 5, with only one instance where the changes between mid-century and end-of the century require consistently a larger ensemble size over an appreciable extent (as many as 15 members over the region). When we conduct the same analysis on the precipitation metric, shown in Figure 10, we are presented with a spatially noisier picture, with changes in variance scattered throughout the Earth's surface, especially over the oceans. In the case of this precipitation metric the ensemble size required is in many regions as large as 15 or 20 members. These results are made clearer by Figures C30 and C31 in the Appendix, where the grid-boxes where significant changes are present are gathered into histograms (weighted according to the Earth's surface fraction that the grid-boxes represent) that show the ensemble size required along the x-axis. We highlight in those figures the fact that for the temperature-based metric only three histograms, corresponding to three specific time-comparisons, gather grid-boxes covering more than 5% of the Earth's surface, while the coverage is more extensive than 5% for all time comparisons for the precipitation metric. These results are representative of the remaining metrics and the alternative model as Figures C20 through C41 in the Appendix document.

We do not show it explicitly here, as it is not the focus of our analysis, but, for both model ensembles, when the change is significant, the ensemble variance increases over time for both precipitation metrics, indicating that the ensemble spread increases with the strength of external forcing over time under RCP 8.5. This is expected as the variance of precipitation increases in step with its mean. For the temperature based metrics, the changes, when significant, are mostly towards an increase in variance (ensemble spread) with forcings for hot extremes (TNx and TXx, the hottest night and day of the year), for which the significant changes are mostly located in the Arctic region. The ensemble spread decreases instead for cold extremes (TNn and TXn, the coldest night and day of the year), for which the significant changes are mostly located in the Southern ocean.

## 4.4 Signal-to-Noise considerations

Another aspect that is implicitly relevant to the establishment of a required ensemble size, if the estimation is concerned with emergence of the forced component, or, more in general, with 'detection and attribution'-type analysis is the Signal-to-Noise ratio of the quantity of interest. Assuming as we have done in our study that the quantity of interest can be regarded as the mean $\mu$ of a noisy population, the signal to noise ratio is defined as $S_N = \mu/\sigma$ where $\sigma$ is the standard deviation of the population. A critical threshold, say $K$, for $S_N$ is usually set at $K = 1$ or 2, and it is immediate to derive the sample size required for such threshold to be hit, by computing the value of $n$ that makes $\mu/(\sigma/\sqrt{n}) \geq K$, i.e., $n \geq K^2/S_N^2$. Figure 11 shows two maps of the spatially varying ensemble sizes required for the signal to noise ratio to exceed 2, when computing anomalies at

mid-century for the two metrics from the CESM ensemble. In appendix we show maps for the remaining metrics and CanESM. The anomalies are computed as 5-year mean differences, as in Section 4.1 under RCP8.5. If the majority of the Earth's surface requires only 2 to 4 ensemble members to be averaged for the temperature metric to reach the $S_N$ value of 2, the Southern Ocean and the Arctic, together with some limited regions over land need more statistical power, up to 18 ensemble members. The pattern remains similar, but the requirements enhanced for the hottest day of the year (TXx, shown in Figure C42). Cold

extremes evidently are more substantially affected by noise over larger portions of the land regions (TNn and TXn, again in Figure C42). The behavior of the precipitation metrics is qualitatively very different, with the great majority of the globe not reaching that level of $S_N$ even when averaging 40 members, as the white areas in Figure 11, bottom panel, and C42, last panel, signify. This discussion is also model specific. Figure C43 shows the same type of results when using CanESM, a model running at a coarser resolution which we therefore expect to show an emergence of the signal from the noise relatively

more easily. This is confirmed by the homogeneous light blue color for the temperature metrics in Figure C43, indicating that between 2 and 6 ensemble member averages reach an $S_N$ of 2. It remains the case also for CanESM, however, that the noise affects substantially $S_N$ for the precipitation metrics.

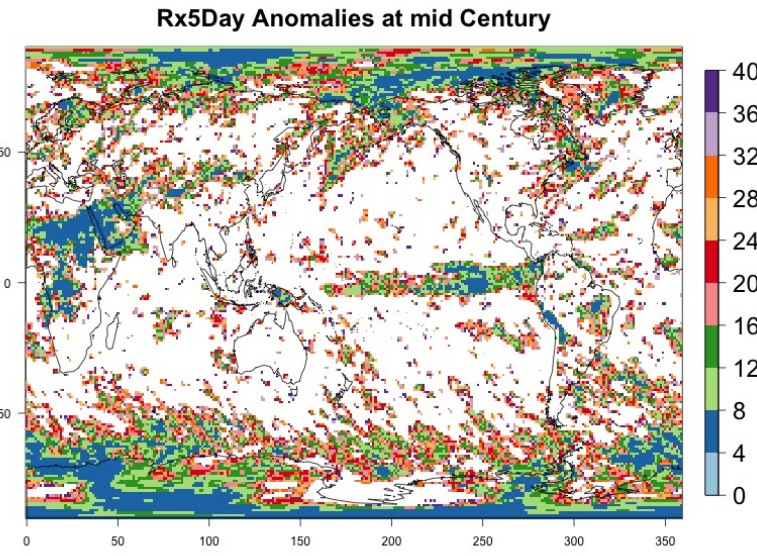

**Figure 11.** Ensemble size $n$ required for the signal to noise ratio of the grid-point scale anomalies to exceed 2 (anomalies defined as the mean of 2048-2052 minus the historical baseline taken as 2000-2005). Results for CESM and hottest night (TXn, top panel) and wettest pentad (Rx5Day, bottom panel) of the year.

## 5 Conclusions

In this study we have addressed the need of deciding a-priori the size of a large ensemble, using an existing 5 member ensemble as our guidance. Aware that the optimal size ultimately depends on the purpose the ensemble is used for, and in order to cover a wide range of possible uses, we chose metrics of temperature and precipitation extremes and we considered output from grid-point scale to global averages. We tackled the problem of characterizing forced changes along the length of a transient scenario simulation, and that of characterizing the system's internal variability and its possible changes. By using a high emission scenario like RCP8.5, but considering behaviors all along the length of the simulations, we are also implicitly addressing a wide range of signal-to-noise magnitudes. Using the availability of existing large ensembles with two different models, CESM1-CAM5 and CanESM2, we could compare our estimates of the expected errors that a given ensemble size would generate with actual errors, obtained using the full ensembles' estimates as our "truth".

First, we find that for the many uses that we explored, it is possible to put a ceiling on the expected error associated with a given ensemble size by exploiting a small ensemble of 5 members. We estimate the ensemble variance at a given simulation date (e.g., 2000, or 2050, or 2095), which is the basis for all our error computations, on the basis of 5 members, "borrowing strength" by using a window of 5 years around that date. The results we assess are consistent with assuming that the quantities of interest are normally distributed with standard deviation $\sigma/\sqrt{n}$, where $\sigma$ can be estimated on the basis of the 5 members available: the error estimates and therefore the optimal sizes computed on the basis of choosing a given tolerance for such errors provide a safe upper bound to the errors that would be committed for a given ensemble size $n$. This is true for all metrics considered, both models, and the full range of scales of aggregation. When we compared such estimates (verified by the availability of the actual large ensembles) there appears to be a sweet spot in the range of ensemble sizes that provides accurate estimates for both forced changes and internal variability, consisting of 20 or 25 members. The larger of these sizes also appears approximately sufficient to conduct an estimation of rare events with as low as $0.01$ probability of occurrence each year, by fitting a GEV and deriving return levels and their confidence intervals. In most cases (locations around the globe, times along the simulation, and metrics considered) enlarging the sample size beyond 25 members provides only marginal improvement in the confidence intervals, while the central estimate does not change significantly from the one established using 25 members, and in most cases accurately approximating that obtained by the full ensemble.

In all cases considered a much smaller ensemble size of 5 to 10 members, if enriched by sampling along the time dimension (that is, using a 5-year window around the date of interest) is sufficient to characterize the ensemble variability, while its changes along the course of the simulations under increasing greenhouse-gases, when found significant using the full ensemble size, can be detected using 15 or 20 ensemble members.

Some caveats are in order. Obviously, the question of how many ensemble members are needed is fundamentally ill-posed, as the answer ultimately and always depends on the most exacting use to which the ensemble is put. One can always find a higher-frequency, smaller-scale metric, and a tighter error bound to satisfy, requiring a larger ensemble size than any previously identified. As tropical cyclone permitting and eventually convection permitting climate model simulations become available, these metrics will be more commonly analyzed. Even for a specific use, the answer depends on the characteristics of internal

variability. The fact that for both the models considered here 5 ensemble members are sufficient to obtain an accurate estimate of it is promising, but not guarantee that 5 are sufficient for all models. In fact, this could also be invalidated by a different experimental exploration of internal variability: new work is adopting different types of initialization, involving ocean states, which could uncover a dimension of internal variability that has so far being under-appreciated(Hawkins et al., 2016; Marotzke, 2019). This would likely change our best estimates of internal variability, and with it possibly the ensemble sizes required to accurately estimate it.

With this work, however we have shown a way to attack the problem "bottom up", starting from a smaller ensemble and building estimates of what would be required for a given problem. One can imagine a more sophisticated set-up where an ensemble can be recursively augmented (rather than assuming a fixed 5-member ensemble as we have done here) in order to approximate the full variability incrementally better. We have also shown that for a large range of questions the size needed is actually well below what we have come to associate with "Large Ensembles". There exist other important sources of uncertainties in climate modeling, one of which is beyond reach of any single modeling center, having to do with structural uncertainty (e.g., Knutti et al. (2010)). Adopting the perspective of an individual model, however, parameter settings have as important a role if not larger as initial conditions. Together with scenario uncertainty, all these dimensions compete over computational resources for their exploration. The same computational resources may be further stretched by the need of downscaling the results of ESM ensembles through regional and impact models (Leduc et al., 2019). Our results may be of guidance in choosing how to allocate those resources among these alternative sources of variation.

*Code and data availability.* The large ensembles output is available through the CLIVAR Large Ensemble Working Group webpage, in the archive maintained through the NCAR CESM community project `cesm.ucar.edu/projects/community-projects/MMLEA/`. R code for these analyses is available from the first author on reasonable request.

# Appendix A: RMSE estimation for more indices and based on the CanESM ensemble

|  | 1953 (F) | 1953 (F-5) | 1953 (B) | 2000 (F) | 2000 (F-5) | 2000 (B) |
|---|---|---|---|---|---|---|
| n=1 | 0.24 (0.19,0.30) | 0.20 (0.16,0.28) | 0.23 | 0.20 (0.17,0.26) | 0.22 (0.17,0.31) | 0.20 |
| n=5 | 0.11 (0.09,0.14) | 0.09 (0.07,0.13) | 0.10 | 0.09 (0.07,0.12) | 0.10 (0.08,0.14) | 0.07 |
| n=10 | 0.07 (0.06,0.10) | 0.06 (0.05,0.09) | 0.07 | 0.06 (0.05,0.08) | 0.07 (0.05,0.10) | 0.05 |
| n=15 | 0.06 (0.05,0.08) | 0.05 (0.04,0.07) | 0.05 | 0.05 (0.04,0.07) | 0.06 (0.04,0.08) | 0.04 |
| n=20 | 0.05 (0.04,0.07) | 0.05 (0.04,0.06) | 0.03 | 0.05 (0.04,0.06) | 0.05 (0.04,0.07) | 0.03 |
| n=25 | 0.05 (0.04,0.06) | 0.04 (0.03,0.06) | 0.02 | 0.04 (0.03,0.05) | 0.04 (0.03,0.06) | 0.02 |
| n=30 | 0.04 (0.04,0.06) | 0.04 (0.03,0.05) | 0.02 | 0.04 (0.03,0.05) | 0.04 (0.03,0.06) | 0.02 |
| n=35 | 0.04 (0.03,0.05) | 0.03 (0.03,0.05) | 0.01 | 0.03 (0.03,0.04) | 0.04 (0.03,0.05) | 0.01 |

|  | 2050 (F) | 2050 (F-5) | 2050 (B) | 2097 (F) | 2097 (F-5) | 2097 (B) |
|---|---|---|---|---|---|---|
| n=1 | 0.20 (0.16,0.26) | 0.21 (0.17,0.30) | 0.20 | 0.24 (0.20,0.32) | 0.18 (0.14,0.25) | 0.24 |
| n=5 | 0.09 (0.07,0.12) | 0.10 (0.08,0.13) | 0.08 | 0.11 (0.09,0.14) | 0.08 (0.06,0.11) | 0.12 |
| n=10 | 0.06 (0.05,0.08) | 0.07 (0.05,0.09) | 0.05 | 0.08 (0.06,0.10) | 0.06 (0.04,0.08) | 0.06 |
| n=15 | 0.05 (0.04,0.07) | 0.06 (0.04,0.08) | 0.04 | 0.06 (0.05,0.08) | 0.05 (0.04,0.07) | 0.06 |
| n=20 | 0.05 (0.04,0.06) | 0.05 (0.04,0.07) | 0.03 | 0.05 (0.04,0.07) | 0.04 (0.03,0.06) | 0.05 |
| n=25 | 0.04 (0.03,0.05) | 0.04 (0.03,0.06) | 0.02 | 0.05 (0.04,0.06) | 0.04 (0.03,0.05) | 0.03 |
| n=30 | 0.04 (0.03,0.05) | 0.04 (0.03,0.05) | 0.02 | 0.04 (0.04,0.06) | 0.03 (0.03,0.05) | 0.02 |
| n=35 | 0.03 (0.03,0.04) | 0.04 (0.03,0.05) | 0.01 | 0.04 (0.03,0.05) | 0.03 (0.02,0.04) | 0.01 |

**Table A1.** Global mean of TNn as simulated by the CESM ensemble: Values of the RMSE in approximating the full ensemble mean by the individual runs (first row, $n = 1$), and by ensembles of increasingly larger sizes (from 5 to 35, along the remaining rows). We compare estimates derived by plugging into the formula a value of $\sigma_t$ estimated by a subset of 5 ensemble members, and 5 years around the year $t$ considered (columns labelled by "(F-5)") to the "truth" defined by the estimates based on the full, 40-member ensemble (columns labelled by "(F)"). We show estimates obtained by the bootstrap approach in the columns labeled by "(B)". These can also be compared to the "truth" and we underline the values that are not consistent with it and its confidence interval, pointing out for which ensemble sizes the bootstrap underestimates the RMSE (usually starting from $n = 20$ or 25). Results are shown for four individual years ($t$) along the simulation (column-wise), since $\sigma_t$ varies along its length.

|  | 1953 (F) | 1953 (F-5) | 1953 (B) | 2000 (F) | 2000 (F-5) | 2000 (B) |
|---|---|---|---|---|---|---|
| n=1 | 0.11 (0.09,0.14) | 0.10 (0.08,0.14) | 0.11 | 0.14 (0.12,0.19) | 0.13 (0.10,0.18) | 0.14 |
| n=5 | 0.05 (0.04,0.06) | 0.04 (0.03,0.06) | 0.05 | 0.06 (0.05,0.08) | 0.06 (0.05,0.08) | 0.05 |
| n=10 | 0.04 (0.03,0.05) | 0.03 (0.02,0.04) | 0.03 | 0.05 (0.04,0.06) | 0.04 (0.03,0.06) | 0.04 |
| n=15 | 0.03 (0.02,0.04) | 0.03 (0.02,0.04) | 0.02 | 0.04 (0.03,0.05) | 0.03 (0.03,0.05) | 0.03 |
| n=20 | 0.02 (0.02,0.03) | 0.02 (0.02,0.03) | 0.02 | 0.03 (0.03,0.04) | 0.03 (0.02,0.04) | 0.02 |
| n=25 | 0.02 (0.02,0.03) | 0.02 (0.02,0.03) | 0.01 | 0.03 (0.02,0.04) | 0.03 (0.02,0.04) | 0.02 |
| n=30 | 0.02 (0.02,0.03) | 0.02 (0.01,0.03) | 0.01 | 0.03 (0.02,0.03) | 0.02 (0.02,0.03) | 0.01 |
| n=35 | 0.02 (0.02,0.02) | 0.02 (0.01,0.02) | 0.01 | 0.02 (0.02,0.03) | 0.02 (0.02,0.03) | 0.01 |

|  | 2050 (F) | 2050 (F-5) | 2050 (B) | 2097 (F) | 2097 (F-5) | 2097 (B) |
|---|---|---|---|---|---|---|
| n=1 | 0.12 (0.10,0.16) | 0.16 (0.12,0.22) | 0.12 | 0.16 (0.13,0.20) | 0.14 (0.11,0.19) | 0.15 |
| n=5 | 0.05 (0.04,0.07) | 0.07 (0.06,0.10) | 0.06 | 0.07 (0.06,0.09) | 0.06 (0.05,0.09) | 0.07 |
| n=10 | 0.04 (0.03,0.05) | 0.05 (0.04,0.07) | 0.03 | 0.05 (0.04,0.06) | 0.04 (0.03,0.06) | 0.04 |
| n=15 | 0.03 (0.03,0.04) | 0.04 (0.03,0.06) | 0.02 | 0.04 (0.03,0.05) | 0.04 (0.03,0.05) | 0.03 |
| n=20 | 0.03 (0.02,0.04) | 0.04 (0.03,0.05) | 0.02 | 0.03 (0.03,0.05) | 0.03 (0.02,0.04) | 0.03 |
| n=25 | 0.02 (0.02,0.03) | 0.03 (0.02,0.04) | 0.01 | 0.03 (0.03,0.04) | 0.03 (0.02,0.04) | 0.02 |
| n=30 | 0.02 (0.02,0.03) | 0.03 (0.02,0.04) | 0.01 | 0.03 (0.02,0.04) | 0.03 (0.02,0.04) | 0.01 |
| n=35 | 0.02 (0.02,0.03) | 0.03 (0.02,0.04) | 0.01 | 0.03 (0.02,0.03) | 0.02 (0.02,0.03) | 0.00 |

**Table A2.** Global mean of TXx as simulated by the CESM ensemble: Values of the RMSE in approximating the full ensemble mean by the individual runs (first row, $n = 1$), and by ensembles of increasingly larger sizes (from 5 to 35, along the remaining rows). We compare estimates derived by plugging into the formula a value of $\sigma_t$ estimated by a subset of 5 ensemble members, and 5 years around the year $t$ considered (columns labelled by "(F-5)") to the "truth" defined by the estimates based on the full, 40-member ensemble (columns labelled by "(F)"). We show estimates obtained by the bootstrap approach in the columns labeled by "(B)". These can also be compared to the "truth" and we underline the values that are not consistent with it and its confidence interval, pointing out for which ensemble sizes the bootstrap underestimates the RMSE (usually starting from $n = 20$ or 25). Results are shown for four individual years ($t$) along the simulation (column-wise), since $\sigma_t$ varies along its length.

|  | 1953 (F) | 1953 (F-5) | 1953 (B) | 2000 (F) | 2000 (F-5) | 2000 (B) |
|---|---|---|---|---|---|---|
| n=1 | 0.21 (0.17,0.27) | 0.19 (0.15,0.27) | 0.21 | 0.20 (0.16,0.25) | 0.21 (0.16,0.29) | 0.19 |
| n=5 | 0.09 (0.08,0.12) | 0.09 (0.07,0.12) | 0.09 | 0.09 (0.07,0.11) | 0.09 (0.07,0.13) | 0.07 |
| n=10 | 0.07 (0.05,0.09) | 0.06 (0.05,0.08) | 0.06 | 0.06 (0.05,0.08) | 0.07 (0.05,0.09) | 0.05 |
| n=15 | 0.05 (0.04,0.07) | 0.05 (0.04,0.07) | 0.04 | 0.05 (0.04,0.07) | 0.05 (0.04,0.08) | 0.04 |
| n=20 | 0.05 (0.04,0.06) | 0.04 (0.03,0.06) | 0.02 | 0.04 (0.04,0.06) | 0.05 (0.04,0.07) | 0.03 |
| n=25 | 0.04 (0.03,0.05) | 0.04 (0.03,0.05) | 0.02 | 0.04 (0.03,0.05) | 0.04 (0.03,0.06) | 0.02 |
| n=30 | 0.04 (0.03,0.05) | 0.04 (0.03,0.05) | 0.02 | 0.04 (0.03,0.05) | 0.04 (0.03,0.05) | 0.02 |
| n=35 | 0.04 (0.03,0.05) | 0.03 (0.03,0.05) | 0.01 | 0.03 (0.03,0.04) | 0.04 (0.03,0.05) | 0.01 |

|  | 2050 (F) | 2050 (F-5) | 2050 (B) | 2097 (F) | 2097 (F-5) | 2097 (B) |
|---|---|---|---|---|---|---|
| n=1 | 0.19 (0.15,0.27) | 0.19 (0.18,0.32) | 0.19 | 0.22 (0.18,0.29) | 0.17 (0.13,0.23) | 0.22 |
| n=5 | 0.09 (0.07,0.12) | 0.09 (0.08,0.14) | 0.08 | 0.10 (0.08,0.13) | 0.08 (0.06,0.10) | 0.11 |
| n=10 | 0.06 (0.05,0.09) | 0.06 (0.06,0.10) | 0.04 | 0.07 (0.06,0.09) | 0.05 (0.04,0.07) | 0.06 |
| n=15 | 0.05 (0.04,0.07) | 0.05 (0.05,0.08) | 0.04 | 0.06 (0.05,0.07) | 0.04 (0.03,0.06) | 0.05 |
| n=20 | 0.04 (0.03,0.06) | 0.04 (0.04,0.07) | 0.03 | 0.05 (0.04,0.06) | 0.04 (0.03,0.05) | 0.04 |
| n=25 | 0.04 (0.03,0.05) | 0.04 (0.04,0.06) | 0.02 | 0.04 (0.04,0.06) | 0.03 (0.03,0.05) | 0.03 |
| n=30 | 0.04 (0.03,0.05) | 0.04 (0.03,0.06) | 0.02 | 0.04 (0.03,0.05) | 0.03 (0.02,0.04) | 0.02 |
| n=35 | 0.03 (0.03,0.05) | 0.03 (0.03,0.05) | 0.01 | 0.04 (0.03,0.05) | 0.03 (0.02,0.04) | 0.01 |

**Table A3.** Global mean of TXn as simulated by the CESM ensemble: Values of the RMSE in approximating the full ensemble mean by the individual runs (first row, $n = 1$), and by ensembles of increasingly larger sizes (from 5 to 35, along the remaining rows). We compare estimates derived by plugging into the formula a value of $\sigma_t$ estimated by a subset of 5 ensemble members, and 5 years around the year $t$ considered (columns labelled by "(F-5)") to the "truth" defined by the estimates based on the full, 40-member ensemble (columns labelled by "(F)"). We show estimates obtained by the bootstrap approach in the columns labeled by "(B)". These can also be compared to the "truth" and we underline the values that are not consistent with it and its confidence interval, pointing out for which ensemble sizes the bootstrap underestimates the RMSE (usually starting from $n = 20$ or 25). Results are shown for four individual years ($t$) along the simulation (column-wise), since $\sigma_t$ varies along its length.

| | 1953 (F) | 1953 (F-5) | 1953 (B) | 2000 (F) | 2000 (F-5) | 2000 (B) |
|---|---|---|---|---|---|---|
| n=1 | 1.06 (0.87,1.36) | 0.90 (0.70,1.25) | 1.05 | 1.25 (1.02,1.60) | 1.15 (0.89,1.59) | 1.23 |
| n=5 | 0.48 (0.39,0.61) | 0.40 (0.31,0.56) | 0.50 | 0.56 (0.46,0.72) | 0.51 (0.40,0.71) | 0.52 |
| n=10 | 0.34 (0.28,0.43) | 0.28 (0.22,0.39) | 0.34 | 0.39 (0.32,0.51) | 0.36 (0.28,0.50) | 0.36 |
| n=15 | 0.27 (0.22,0.35) | 0.23 (0.18,0.32) | 0.20 | 0.32 (0.26,0.41) | 0.30 (0.23,0.41) | 0.29 |
| n=20 | 0.24 (0.19,0.31) | 0.20 (0.16,0.28) | 0.19 | 0.28 (0.23,0.36) | 0.26 (0.20,0.36) | 0.17 |
| n=25 | 0.21 (0.17,0.27) | 0.18 (0.14,0.25) | 0.12 | 0.25 (0.20,0.32) | 0.23 (0.18,0.32) | 0.17 |
| n=30 | 0.19 (0.16,0.25) | 0.16 (0.13,0.23) | 0.08 | 0.23 (0.19,0.29) | 0.21 (0.16,0.29) | 0.11 |
| n=35 | 0.18 (0.15,0.23) | 0.15 (0.12,0.21) | 0.07 | 0.21 (0.17,0.27) | 0.19 (0.15,0.27) | 0.08 |

| | 2050 (F) | 2050 (F-5) | 2050 (B) | 2097 (F) | 2097 (F-5) | 2097 (B) |
|---|---|---|---|---|---|---|
| n=1 | 1.79 (1.47,2.30) | 1.69 (1.32,2.36) | 1.77 | 2.20 (1.81,2.83) | 2.22 (1.73,3.08) | 2.18 |
| n=5 | 0.80 (0.66,1.03) | 0.76 (0.59,1.05) | 0.82 | 0.99 (0.81,1.27) | 0.99 (0.77,1.38) | 0.80 |
| n=10 | 0.57 (0.46,0.73) | 0.54 (0.42,0.75) | 0.42 | 0.70 (0.57,0.89) | 0.70 (0.55,0.98) | 0.58 |
| n=15 | 0.46 (0.38,0.59) | 0.44 (0.34,0.61) | 0.43 | 0.57 (0.47,0.73) | 0.57 (0.45,0.80) | 0.38 |
| n=20 | 0.40 (0.33,0.51) | 0.38 (0.30,0.53) | 0.29 | 0.49 (0.40,0.63) | 0.50 (0.39,0.69) | 0.34 |
| n=25 | 0.36 (0.29,0.46) | 0.34 (0.26,0.47) | 0.18 | 0.44 (0.36,0.57) | 0.44 (0.35,0.62) | 0.25 |
| n=30 | 0.33 (0.27,0.42) | 0.31 (0.24,0.43) | 0.16 | 0.40 (0.33,0.52) | 0.40 (0.32,0.56) | 0.22 |
| n=35 | 0.30 (0.25,0.39) | 0.29 (0.22,0.40) | 0.10 | 0.37 (0.31,0.48) | 0.37 (0.29,0.52) | 0.13 |

**Table A4.** Global mean of Rx1Day as simulated by the CESM ensemble: Values of the RMSE in approximating the full ensemble mean by the individual runs (first row, $n = 1$), and by ensembles of increasingly larger sizes (from 5 to 35, along the remaining rows). We compare estimates derived by plugging into the formula a value of $\sigma_t$ estimated by a subset of 5 ensemble members, and 5 years around the year $t$ considered (columns labelled by "(F-5)") to the "truth" defined by the estimates based on the full, 40-member ensemble (columns labelled by "(F)"). We show estimates obtained by the bootstrap approach in the columns labeled by "(B)". These can also be compared to the "truth" and we underline the values that are not consistent with it and its confidence interval, pointing out for which ensemble sizes the bootstrap underestimates the RMSE (usually starting from $n = 20$ or 25). Results are shown for four individual years ($t$) along the simulation (column-wise), since $\sigma_t$ varies along its length.

|  | 1953 (F) | 1953 (F-5) | 1953 (B) | 2000 (F) | 2000 (F-5) | 2000 (B) |
|---|---|---|---|---|---|---|
| n=1 | 0.14 (0.12,0.17) | 0.12 (0.10,0.17) | 0.14 | 0.16 (0.13,0.19) | 0.12 (0.09,0.17) | 0.15 |
| n=5 | 0.06 (0.05,0.08) | 0.05 (0.04,0.08) | 0.06 | 0.07 (0.06,0.09) | 0.05 (0.04,0.07) | 0.07 |
| n=10 | 0.04 (0.04,0.05) | 0.04 (0.03,0.05) | 0.04 | 0.05 (0.04,0.06) | 0.04 (0.03,0.05) | 0.04 |
| n=15 | 0.04 (0.03,0.04) | 0.03 (0.02,0.04) | 0.04 | 0.04 (0.03,0.05) | 0.03 (0.02,0.04) | 0.03 |
| n=20 | 0.03 (0.03,0.04) | 0.03 (0.02,0.04) | 0.03 | 0.03 (0.03,0.04) | 0.03 (0.02,0.04) | 0.03 |
| n=25 | 0.03 (0.02,0.03) | 0.02 (0.02,0.03) | 0.02 | 0.03 (0.03,0.04) | 0.02 (0.02,0.03) | 0.02 |
| n=30 | 0.03 (0.02,0.03) | 0.02 (0.02,0.03) | 0.01 | 0.03 (0.02,0.04) | 0.02 (0.02,0.03) | 0.02 |
| n=35 | 0.02 (0.02,0.03) | 0.02 (0.02,0.03) | 0.01 | 0.03 (0.02,0.03) | 0.02 (0.02,0.03) | 0.01 |
| n=40 | 0.02 (0.02,0.03) | 0.02 (0.02,0.03) | 0.01 | 0.02 (0.02,0.03) | 0.02 (0.01,0.03) | 0.01 |
|  | 2050 (F) | 2050 (F-5) | 2050 (B) | 2097 (F) | 2097 (F-5) | 2097 (B) |
| n=1 | 0.14 (0.11,0.17) | 0.13 (0.10,0.18) | 0.13 | 0.12 (0.10,0.15) | 0.14 (0.11,0.20) | 0.12 |
| n=5 | 0.06 (0.05,0.08) | 0.06 (0.04,0.08) | 0.06 | 0.05 (0.04,0.07) | 0.06 (0.05,0.09) | 0.06 |
| n=10 | 0.04 (0.04,0.05) | 0.04 (0.03,0.06) | 0.04 | 0.04 (0.03,0.05) | 0.05 (0.04,0.06) | 0.04 |
| n=15 | 0.04 (0.03,0.04) | 0.03 (0.03,0.05) | 0.03 | 0.03 (0.03,0.04) | 0.04 (0.03,0.05) | 0.03 |
| n=20 | 0.03 (0.03,0.04) | 0.03 (0.02,0.04) | 0.03 | 0.03 (0.02,0.03) | 0.03 (0.02,0.04) | 0.02 |
| n=25 | 0.03 (0.02,0.03) | 0.03 (0.02,0.04) | 0.02 | 0.02 (0.02,0.03) | 0.03 (0.02,0.04) | 0.02 |
| n=30 | 0.02 (0.02,0.03) | 0.02 (0.02,0.03) | 0.02 | 0.02 (0.02,0.03) | 0.03 (0.02,0.04) | 0.01 |
| n=35 | 0.02 (0.02,0.03) | 0.02 (0.02,0.03) | 0.01 | 0.02 (0.02,0.03) | 0.02 (0.02,0.03) | 0.01 |
| n=40 | 0.02 (0.02,0.03) | 0.02 (0.02,0.03) | 0.01 | 0.02 (0.02,0.02) | 0.02 (0.02,0.03) | 0.01 |

**Table A5.** Global mean of TNx as simulated by the CanESM ensemble: Values of the RMSE in approximating the full ensemble mean by the individual runs (first row, $n = 1$), and by ensembles of increasingly larger sizes (from 5 to 40, along the remaining rows). We compare estimates derived by plugging into the formula a value of $\sigma_t$ estimated by a subset of 5 ensemble members, and 5 years around the year $t$ considered (columns labelled by "(F-5)") to the "truth" defined by the estimates based on the full 50-member ensemble (columns labelled by "(F)"). We show estimates obtained by the bootstrap approach in the columns labeled by "(B)". These can also be compared to the "truth" and we underline the values that are not consistent with it and its confidence interval, pointing out for which ensemble sizes the bootstrap underestimates the RMSE (usually starting from $n = 20$ or 25). Results are shown for four individual years ($t$) along the simulation (column-wise), since $\sigma_t$ varies along its length.

|      | 1953 (F)           | 1953 (F-5)         | 1953 (B) | 2000 (F)           | 2000 (F-5)         | 2000 (B) |
|------|--------------------|--------------------|----------|--------------------|--------------------|----------|
| n=1  | 0.29 (0.24,0.36)   | 0.35 (0.27,0.49)   | 0.29     | 0.27 (0.22,0.33)   | 0.26 (0.20,0.36)   | 0.26     |
| n=5  | 0.13 (0.11,0.16)   | 0.16 (0.12,0.22)   | 0.10     | 0.12 (0.10,0.15)   | 0.12 (0.09,0.16)   | 0.11     |
| n=10 | 0.09 (0.08,0.12)   | 0.11 (0.09,0.15)   | 0.09     | 0.08 (0.07,0.10)   | 0.08 (0.06,0.11)   | 0.07     |
| n=15 | 0.08 (0.06,0.09)   | 0.09 (0.07,0.13)   | 0.06     | 0.07 (0.06,0.09)   | 0.07 (0.05,0.09)   | 0.05     |
| n=20 | 0.07 (0.05,0.08)   | 0.08 (0.06,0.11)   | 0.05     | 0.06 (0.05,0.07)   | 0.06 (0.05,0.08)   | 0.04     |
| n=25 | 0.06 (0.05,0.07)   | 0.07 (0.05,0.10)   | 0.04     | 0.05 (0.04,0.07)   | 0.05 (0.04,0.07)   | 0.04     |
| n=30 | 0.05 (0.04,0.07)   | 0.06 (0.05,0.09)   | 0.03     | 0.05 (0.04,0.06)   | 0.05 (0.04,0.07)   | 0.03     |
| n=35 | 0.05 (0.04,0.06)   | 0.06 (0.05,0.08)   | 0.03     | 0.05 (0.04,0.06)   | 0.04 (0.03,0.06)   | 0.02     |
| n=40 | 0.05 (0.04,0.06)   | 0.06 (0.04,0.08)   | 0.02     | 0.04 (0.04,0.05)   | 0.04 (0.03,0.06)   | 0.02     |

|      | 2050 (F)           | 2050 (F-5)         | 2050 (B) | 2097 (F)           | 2097 (F-5)         | 2097 (B) |
|------|--------------------|--------------------|----------|--------------------|--------------------|----------|
| n=1  | 0.29 (0.25,0.37)   | 0.30 (0.23,0.41)   | 0.29     | 0.34 (0.28,0.42)   | 0.31 (0.24,0.44)   | 0.33     |
| n=5  | 0.13 (0.11,0.16)   | 0.13 (0.10,0.18)   | 0.13     | 0.15 (0.13,0.19)   | 0.14 (0.11,0.19)   | 0.17     |
| n=10 | 0.09 (0.08,0.12)   | 0.09 (0.07,0.13)   | 0.08     | 0.11 (0.09,0.13)   | 0.10 (0.08,0.14)   | 0.10     |
| n=15 | 0.08 (0.06,0.09)   | 0.08 (0.06,0.11)   | 0.05     | 0.09 (0.07,0.11)   | 0.08 (0.06,0.11)   | 0.08     |
| n=20 | 0.07 (0.05,0.08)   | 0.07 (0.05,0.09)   | 0.05     | 0.08 (0.06,0.09)   | 0.07 (0.05,0.10)   | 0.06     |
| n=25 | 0.06 (0.05,0.07)   | 0.06 (0.05,0.08)   | 0.04     | 0.07 (0.06,0.08)   | 0.06 (0.05,0.09)   | 0.05     |
| n=30 | 0.05 (0.04,0.07)   | 0.05 (0.04,0.07)   | 0.04     | 0.06 (0.05,0.08)   | 0.06 (0.04,0.08)   | 0.04     |
| n=35 | 0.05 (0.04,0.06)   | 0.05 (0.04,0.07)   | 0.03     | 0.06 (0.05,0.07)   | 0.05 (0.04,0.07)   | 0.03     |
| n=40 | 0.05 (0.04,0.06)   | 0.05 (0.04,0.06)   | 0.02     | 0.05 (0.04,0.07)   | 0.05 (0.04,0.07)   | 0.02     |

**Table A6.** Global mean of Rx5Day as simulated by the CanESM ensemble: Values of the RMSE in approximating the full ensemble mean by the individual runs (first row, $n = 1$), and by ensembles of increasingly larger sizes (from 5 to 40, along the remaining rows). We compare estimates derived by plugging into the formula a value of $\sigma_t$ estimated by a subset of 5 ensemble members, and 5 years around the year $t$ considered (columns labelled by "(F-5)") to the "truth" defined by the estimates based on the full, 50-member ensemble (columns labelled by "(F)"). We show estimates obtained by the bootstrap approach in the columns labeled by "(B)". These can also be compared to the "truth" and we underline the values that are not consistent with it and its confidence interval, pointing out for which ensemble sizes the bootstrap underestimates the RMSE (usually starting from $n = 20$ or 25. Results are shown for four individual years ($t$) along the simulation (column-wise), since $\sigma_t$ varies along its length.

|  | 1953 (F) | 1953 (F-5) | 1953 (B) | 2000 (F) | 2000 (F-5) | 2000 (B) |
|---|---|---|---|---|---|---|
| n=1 | 0.20 (0.17,0.25) | 0.20 (0.15,0.28) | 0.20 | 0.21 (0.17,0.26) | 0.22 (0.17,0.30) | 0.20 |
| n=5 | 0.09 (0.07,0.11) | 0.09 (0.07,0.12) | 0.09 | 0.09 (0.08,0.12) | 0.10 (0.08,0.13) | 0.10 |
| n=10 | 0.06 (0.05,0.08) | 0.06 (0.05,0.09) | 0.06 | 0.07 (0.05,0.08) | 0.07 (0.05,0.10) | 0.05 |
| n=15 | 0.05 (0.04,0.06) | 0.05 (0.04,0.07) | 0.05 | 0.05 (0.04,0.07) | 0.06 (0.04,0.08) | 0.04 |
| n=20 | 0.04 (0.04,0.06) | 0.04 (0.03,0.06) | 0.04 | 0.05 (0.04,0.06) | 0.05 (0.04,0.07) | 0.04 |
| n=25 | 0.04 (0.03,0.05) | 0.04 (0.03,0.06) | 0.03 | 0.04 (0.03,0.05) | 0.04 (0.03,0.06) | 0.03 |
| n=30 | 0.04 (0.03,0.05) | 0.04 (0.03,0.05) | 0.02 | 0.04 (0.03,0.05) | 0.04 (0.03,0.06) | 0.02 |
| n=35 | 0.03 (0.03,0.04) | 0.03 (0.03,0.05) | 0.02 | 0.03 (0.03,0.04) | 0.04 (0.03,0.05) | 0.02 |
| n=40 | 0.03 (0.03,0.04) | 0.03 (0.03,0.04) | 0.01 | 0.03 (0.03,0.04) | 0.03 (0.03,0.05) | 0.01 |
|  | 2050 (F) | 2050 (F-5) | 2050 (B) | 2097 (F) | 2097 (F-5) | 2097 (B) |
| n=1 | 0.22 (0.19,0.28) | 0.22 (0.17,0.30) | 0.22 | 0.22 (0.18,0.27) | 0.23 (0.19,0.32) | 0.21 |
| n=5 | 0.10 (0.08,0.12) | 0.10 (0.08,0.13) | 0.10 | 0.10 (0.08,0.12) | 0.10 (0.08,0.15) | 0.08 |
| n=10 | 0.07 (0.06,0.09) | 0.07 (0.05,0.09) | 0.06 | 0.07 (0.06,0.09) | 0.07 (0.06,0.10) | 0.07 |
| n=15 | 0.06 (0.05,0.07) | 0.06 (0.04,0.08) | 0.04 | 0.06 (0.05,0.07) | 0.06 (0.05,0.08) | 0.04 |
| n=20 | 0.05 (0.04,0.06) | 0.05 (0.04,0.07) | 0.04 | 0.05 (0.04,0.06) | 0.05 (0.04,0.07) | 0.03 |
| n=25 | 0.04 (0.04,0.06) | 0.04 (0.03,0.06) | 0.02 | 0.04 (0.04,0.05) | 0.05 (0.04,0.06) | 0.03 |
| n=30 | 0.04 (0.03,0.05) | 0.04 (0.03,0.05) | 0.03 | 0.04 (0.03,0.05) | 0.04 (0.03,0.06) | 0.02 |
| n=35 | 0.04 (0.03,0.05) | 0.04 (0.03,0.05) | 0.02 | 0.04 (0.03,0.05) | 0.04 (0.03,0.05) | 0.02 |
| n=40 | 0.04 (0.03,0.04) | 0.03 (0.03,0.05) | 0.01 | 0.03 (0.03,0.04) | 0.04 (0.03,0.05) | 0.02 |

**Table A7.** Global mean of TNn as simulated by the CanESM ensemble: Values of the RMSE in approximating the full ensemble mean by the individual runs (first row, $n = 1$), and by ensembles of increasingly larger sizes (from 5 to 40, along the remaining rows). We compare estimates derived by plugging into the formula a value of $\sigma_t$ estimated by a subset of 5 ensemble members, and 5 years around the year $t$ considered (columns labelled by "(F-5)") to the "truth" defined by the estimates based on the full, 50-member ensemble (columns labelled by "(F)"). We show estimates obtained by the bootstrap approach in the columns labeled by "(B)". These can also be compared to the "truth" and we underline the values that are not consistent with it and its confidence interval, pointing out for which ensemble sizes the bootstrap underestimates the RMSE (usually starting from $n = 20$ or 25. Results are shown for four individual years ($t$) along the simulation (column-wise), since $\sigma_t$ varies along its length.

|  | 1953 (F) | 1953 (F-5) | 1953 (B) | 2000 (F) | 2000 (F-5) | 2000 (B) |
|---|---|---|---|---|---|---|
| n=1 | 0.16 (0.13,0.20) | 0.14 (0.11,0.19) | 0.16 | 0.17 (0.14,0.22) | 0.14 (0.11,0.19) | 0.17 |
| n=5 | 0.07 (0.06,0.09) | 0.06 (0.05,0.09) | 0.07 | 0.08 (0.06,0.10) | 0.06 (0.05,0.08) | 0.07 |
| n=10 | 0.05 (0.04,0.06) | 0.04 (0.03,0.06) | 0.05 | 0.05 (0.05,0.07) | 0.04 (0.03,0.06) | 0.05 |
| n=15 | 0.04 (0.03,0.05) | 0.04 (0.03,0.05) | 0.03 | 0.04 (0.04,0.06) | 0.03 (0.03,0.05) | 0.04 |
| n=20 | 0.04 (0.03,0.04) | 0.03 (0.02,0.04) | 0.03 | 0.04 (0.03,0.05) | 0.03 (0.02,0.04) | 0.03 |
| n=25 | 0.03 (0.03,0.04) | 0.03 (0.02,0.04) | 0.02 | 0.03 (0.03,0.04) | 0.03 (0.02,0.04) | 0.03 |
| n=30 | 0.03 (0.02,0.04) | 0.03 (0.02,0.04) | 0.02 | 0.03 (0.03,0.04) | 0.02 (0.02,0.03) | 0.02 |
| n=35 | 0.03 (0.02,0.03) | 0.02 (0.02,0.03) | 0.01 | 0.03 (0.02,0.04) | 0.02 (0.02,0.03) | 0.01 |
| n=40 | 0.02 (0.02,0.03) | 0.02 (0.02,0.03) | 0.01 | 0.03 (0.02,0.03) | 0.02 (0.02,0.03) | 0.01 |

|  | 2050 (F) | 2050 (F-5) | 2050 (B) | 2097 (F) | 2097 (F-5) | 2097 (B) |
|---|---|---|---|---|---|---|
| n=1 | 0.14 (0.11,0.17) | 0.14 (0.11,0.20) | 0.13 | 0.13 (0.11,0.16) | 0.14 (0.11,0.20) | 0.13 |
| n=5 | 0.06 (0.05,0.08) | 0.06 (0.05,0.09) | 0.06 | 0.06 (0.05,0.07) | 0.06 (0.05,0.09) | 0.06 |
| n=10 | 0.04 (0.04,0.05) | 0.04 (0.03,0.06) | 0.03 | 0.04 (0.03,0.05) | 0.05 (0.04,0.06) | 0.04 |
| n=15 | 0.03 (0.03,0.04) | 0.04 (0.03,0.05) | 0.03 | 0.03 (0.03,0.04) | 0.04 (0.03,0.05) | 0.03 |
| n=20 | 0.03 (0.03,0.04) | 0.03 (0.02,0.04) | 0.03 | 0.03 (0.02,0.03) | 0.03 (0.03,0.04) | 0.02 |
| n=25 | 0.03 (0.02,0.03) | 0.03 (0.02,0.04) | 0.02 | 0.03 (0.02,0.03) | 0.03 (0.02,0.04) | 0.02 |
| n=30 | 0.02 (0.02,0.03) | 0.03 (0.02,0.04) | 0.01 | 0.02 (0.02,0.03) | 0.03 (0.02,0.04) | 0.01 |
| n=35 | 0.02 (0.02,0.03) | 0.02 (0.02,0.03) | 0.01 | 0.02 (0.02,0.03) | 0.02 (0.02,0.03) | 0.01 |
| n=40 | 0.02 (0.02,0.03) | 0.02 (0.02,0.03) | 0.01 | 0.02 (0.02,0.03) | 0.02 (0.02,0.03) | 0.01 |

**Table A8.** Global mean of TXx as simulated by the CanESM ensemble: Values of the RMSE in approximating the full ensemble mean by the individual runs (first row, $n = 1$), and by ensembles of increasingly larger sizes (from 5 to 40, along the remaining rows). We compare estimates derived by plugging into the formula a value of $\sigma_t$ estimated by a subset of 5 ensemble members, and 5 years around the year $t$ considered (columns labelled by "(F-5)") to the "truth" defined by the estimates based on the full, 50-member ensemble (columns labelled by "(F)"). We show estimates obtained by the bootstrap approach in the columns labeled by "(B)". These can also be compared to the "truth" and we underline the values that are not consistent with it and its confidence interval, pointing out for which ensemble sizes the bootstrap underestimates the RMSE (usually starting from $n = 20$ or 25. Results are shown for four individual years ($t$) along the simulation (column-wise), since $\sigma_t$ varies along its length.

|       | 1953 (F)           | 1953 (F-5)         | 1953 (B) | 2000 (F)           | 2000 (F-5)         | 2000 (B) |
|-------|--------------------|--------------------|----------|--------------------|--------------------|----------|
| n=1   | 0.20 (0.17,0.25)   | 0.19 (0.15,0.27)   | 0.20     | 0.20 (0.17,0.25)   | 0.21 (0.17,0.30)   | 0.20     |
| n=5   | 0.09 (0.07,0.11)   | 0.09 (0.07,0.12)   | 0.09     | 0.09 (0.07,0.11)   | 0.10 (0.07,0.13)   | 0.07     |
| n=10  | 0.06 (0.05,0.08)   | 0.06 (0.05,0.08)   | 0.05     | 0.06 (0.05,0.08)   | 0.07 (0.05,0.09)   | _0.04_   |
| n=15  | 0.05 (0.04,0.06)   | 0.05 (0.04,0.07)   | 0.04     | 0.05 (0.04,0.06)   | 0.06 (0.04,0.08)   | _0.03_   |
| n=20  | 0.04 (0.04,0.06)   | 0.04 (0.03,0.06)   | _0.03_   | 0.04 (0.04,0.06)   | 0.05 (0.04,0.07)   | _0.03_   |
| n=25  | 0.04 (0.03,0.05)   | 0.04 (0.03,0.05)   | 0.03     | 0.04 (0.03,0.05)   | 0.04 (0.03,0.06)   | _0.02_   |
| n=30  | 0.04 (0.03,0.05)   | 0.04 (0.03,0.05)   | _0.02_   | 0.04 (0.03,0.05)   | 0.04 (0.03,0.05)   | _0.02_   |
| n=35  | 0.03 (0.03,0.04)   | 0.03 (0.03,0.05)   | _0.02_   | 0.03 (0.03,0.04)   | 0.04 (0.03,0.05)   | _0.02_   |
| n=40  | 0.03 (0.03,0.04)   | 0.03 (0.03,0.04)   | _0.01_   | 0.03 (0.03,0.04)   | 0.03 (0.03,0.05)   | _0.01_   |

|       | 2050 (F)           | 2050 (F-5)         | 2050 (B) | 2097 (F)           | 2097 (F-5)         | 2097 (B) |
|-------|--------------------|--------------------|----------|--------------------|--------------------|----------|
| n=1   | 0.21 (0.18,0.26)   | 0.21 (0.17,0.30)   | 0.21     | 0.20 (0.17,0.25)   | 0.21 (0.17,0.30)   | 0.20     |
| n=5   | 0.09 (0.08,0.12)   | 0.10 (0.07,0.13)   | 0.08     | 0.09 (0.07,0.11)   | 0.10 (0.07,0.13)   | 0.10     |
| n=10  | 0.07 (0.06,0.08)   | 0.07 (0.05,0.09)   | _0.05_   | 0.06 (0.05,0.08)   | 0.07 (0.05,0.09)   | 0.06     |
| n=15  | 0.05 (0.05,0.07)   | 0.06 (0.04,0.08)   | 0.05     | 0.05 (0.04,0.06)   | 0.05 (0.04,0.08)   | 0.05     |
| n=20  | 0.05 (0.04,0.06)   | 0.05 (0.04,0.07)   | 0.04     | 0.04 (0.04,0.06)   | 0.05 (0.04,0.07)   | _0.03_   |
| n=25  | 0.04 (0.04,0.05)   | 0.04 (0.03,0.06)   | _0.03_   | 0.04 (0.03,0.05)   | 0.04 (0.03,0.06)   | _0.02_   |
| n=30  | 0.04 (0.03,0.05)   | 0.04 (0.03,0.05)   | _0.02_   | 0.04 (0.03,0.05)   | 0.04 (0.03,0.05)   | _0.02_   |
| n=35  | 0.04 (0.03,0.04)   | 0.04 (0.03,0.05)   | _0.02_   | 0.03 (0.03,0.04)   | 0.04 (0.03,0.05)   | _0.02_   |
| n=40  | 0.03 (0.03,0.04)   | 0.03 (0.03,0.05)   | _0.01_   | 0.03 (0.03,0.04)   | 0.03 (0.03,0.05)   | _0.01_   |

**Table A9.** Global mean of TXn as simulated by the CanESM ensemble: Values of the RMSE in approximating the full ensemble mean by the individual runs (first row, $n = 1$), and by ensembles of increasingly larger sizes (from 5 to 40, along the remaining rows). We compare estimates derived by plugging into the formula a value of $\sigma_t$ estimated by a subset of 5 ensemble members, and 5 years around the year $t$ considered (columns labelled by "(F-5)") to the "truth" defined by the estimates based on the full, 50-member ensemble (columns labelled by "(F)"). We show estimates obtained by the bootstrap approach in the columns labeled by "(B)". These can also be compared to the "truth" and we underline the values that are not consistent with it and its confidence interval, pointing out for which ensemble sizes the bootstrap underestimates the RMSE (usually starting from $n = 20$ or 25. Results are shown for four individual years ($t$) along the simulation (column-wise), since $\sigma_t$ varies along its length.

|       | 1953 (F) | 1953 (F-5) | 1953 (B) | 2000 (F) | 2000 (F-5) | 2000 (B) |
|-------|----------|-----------|----------|----------|-----------|----------|
| n=1   | 0.90 (0.75,1.12) | 0.68 (0.53,0.94) | 0.89 | 0.87 (0.73,1.08) | 0.68 (0.53,0.95) | 0.86 |
| n=5   | 0.40 (0.33,0.50) | 0.30 (0.24,0.42) | 0.38 | 0.39 (0.32,0.48) | 0.31 (0.24,0.43) | 0.44 |
| n=10  | 0.28 (0.24,0.35) | 0.21 (0.17,0.30) | 0.25 | 0.28 (0.23,0.34) | 0.22 (0.17,0.30) | 0.26 |
| n=15  | 0.23 (0.19,0.29) | 0.17 (0.14,0.24) | 0.20 | 0.22 (0.19,0.28) | 0.18 (0.14,0.25) | 0.21 |
| n=20  | 0.20 (0.17,0.25) | 0.15 (0.12,0.21) | 0.17 | 0.19 (0.16,0.24) | 0.15 (0.12,0.21) | 0.15 |
| n=25  | 0.18 (0.15,0.22) | 0.14 (0.11,0.19) | 0.12 | 0.17 (0.15,0.22) | 0.14 (0.11,0.19) | 0.11 |
| n=30  | 0.16 (0.14,0.20) | 0.12 (0.10,0.17) | 0.10 | 0.16 (0.13,0.20) | 0.12 (0.10,0.17) | 0.10 |
| n=35  | 0.15 (0.13,0.19) | 0.11 (0.09,0.16) | 0.08 | 0.15 (0.12,0.18) | 0.12 (0.09,0.16) | 0.08 |
| n=40  | 0.14 (0.12,0.18) | 0.11 (0.08,0.15) | 0.07 | 0.14 (0.11,0.17) | 0.11 (0.11,0.17) | 0.05 |

|       | 2050 (F) | 2050 (F-5) | 2050 (B) | 2097 (F) | 2097 (F-5) | 2097 (B) |
|-------|----------|-----------|----------|----------|-----------|----------|
| n=1   | 0.99 (0.82,1.23) | 0.97 (0.76,1.35) | 0.98 | 0.98 (0.82,1.22) | 0.87 (0.68,1.21) | 0.97 |
| n=5   | 0.44 (0.37,0.55) | 0.43 (0.34,0.60) | 0.40 | 0.44 (0.37,0.55) | 0.39 (0.30,0.54) | 0.49 |
| n=10  | 0.31 (0.26,0.39) | 0.31 (0.24,0.43) | 0.28 | 0.31 (0.26,0.39) | 0.27 (0.21,0.38) | 0.27 |
| n=15  | 0.25 (0.21,0.32) | 0.25 (0.20,0.35) | 0.20 | 0.25 (0.21,0.32) | 0.22 (0.17,0.31) | 0.18 |
| n=20  | 0.22 (0.18,0.28) | 0.22 (0.17,0.30) | 0.20 | 0.22 (0.18,0.27) | 0.19 (0.15,0.27) | 0.13 |
| n=25  | 0.20 (0.16,0.25) | 0.19 (0.15,0.27) | 0.15 | 0.20 (0.16,0.24) | 0.17 (0.14,0.24) | 0.13 |
| n=30  | 0.18 (0.15,0.22) | 0.18 (0.14,0.25) | 0.11 | 0.18 (0.15,0.22) | 0.16 (0.12,0.22) | 0.10 |
| n=35  | 0.17 (0.14,0.21) | 0.16 (0.13,0.23) | 0.10 | 0.17 (0.14,0.21) | 0.15 (0.11,0.20) | 0.09 |
| n=40  | 0.16 (0.13,0.19) | 0.15 (0.12,0.21) | 0.07 | 0.15 (0.12,0.21) | 0.14 (0.11,0.19) | 0.07 |

**Table A10.** Global mean of Rx1Day as simulated by the CanESM ensemble: Values of the RMSE in approximating the full ensemble mean by the individual runs (first row, $n = 1$), and by ensembles of increasingly larger sizes (from 5 to 40, along the remaining rows). We compare estimates derived by plugging into the formula a value of $\sigma_t$ estimated by a subset of 5 ensemble members, and 5 years around the year $t$ considered (columns labelled by "(F-5)") to the "truth" defined by the estimates based on the full, 50-member ensemble (columns labelled by "(F)"). We show estimates obtained by the bootstrap approach in the columns labeled by "(B)". These can also be compared to the "truth" and we underline the values that are not consistent with it and its confidence interval, pointing out for which ensemble sizes the bootstrap underestimates the RMSE (usually starting from $n = 20$ or 25). Results are shown for four individual years ($t$) along the simulation (column-wise), since $\sigma_t$ varies along its length.

**Appendix B: Summary of error ratio patterns as shown in Figures 3 and 4 for all metrics and models**

| Ens. size | 1 | 5 | 10 | 15 | 20 | 25 | 30 | 35 |
|---|---|---|---|---|---|---|---|---|
| TNx Global midC | 4.74 | 8.27 | 5.52 | 4.81 | 2.65 | 2.91 | 0.87 | 0.07 |
| TNx Global endC | 3.07 | 6.49 | 6.34 | 3.57 | 3.73 | 2.38 | 1.02 | 0.08 |
| TNx Land midC | 3.8 | 11.05 | 7.14 | 4.3 | 3.11 | 2.95 | 0.96 | 0.14 |
| TNx Land endC | 3.88 | 9.01 | 6.35 | 3.34 | 3.06 | 1.99 | 0.79 | 0.06 |
| TNx Ocean midC | 5.13 | 7.13 | 4.85 | 5.01 | 2.46 | 2.89 | 0.83 | 0.05 |
| TNx Ocean endC | 2.73 | 5.45 | 6.33 | 3.67 | 4 | 2.55 | 1.11 | 0.08 |
| TNn Global midC | 11.98 | 10.18 | 8.66 | 5.18 | 3.68 | 2.67 | 1.21 | 0.2 |
| TNn Global endC | 11.4 | 8.11 | 8.42 | 6.03 | 3.58 | 3.57 | 0.53 | 0.11 |
| TNn Land midC | 13.03 | 10.05 | 8.98 | 5.41 | 2.91 | 4 | 0.73 | 0.18 |
| TNn Land endC | 11.99 | 9.92 | 10.02 | 7.48 | 4.83 | 6.45 | 0.93 | 0.21 |
| TNn Ocean midC | 11.55 | 10.23 | 8.53 | 5.08 | 3.99 | 2.12 | 1.41 | 0.21 |
| TNn Ocean endC | 11.15 | 7.36 | 7.77 | 5.44 | 3.07 | 2.39 | 0.36 | 0.08 |
| TXx Global midC | 14.99 | 7.78 | 8.65 | 4.9 | 5.88 | 3.58 | 1.06 | 0.2 |
| TXx Global endC | 12.28 | 8.98 | 7.92 | 5.3 | 4.98 | 1.72 | 1.11 | 0.29 |
| TXx Land midC | 13.48 | 8.4 | 6.68 | 5.27 | 4.56 | 2.18 | 1.49 | 0.14 |
| TXx Land endC | 14.08 | 11.64 | 7.76 | 4.84 | 4.7 | 2.19 | 1.37 | 0.29 |
| TXx Ocean midC | 15.6 | 7.52 | 9.46 | 4.75 | 6.42 | 4.16 | 0.88 | 0.23 |
| TXx Ocean endC | 11.54 | 7.89 | 7.98 | 5.49 | 5.1 | 1.53 | 1 | 0.3 |
| TXn Global midC | 11.1 | 10.84 | 7.59 | 5.14 | 5.63 | 2.68 | 0.92 | 0.38 |
| TXn Global endC | 10.33 | 11.1 | 7.27 | 4.34 | 3.66 | 2.02 | 0.66 | 0.34 |
| TXn Land midC | 12.16 | 9.22 | 9.3 | 4.84 | 3.52 | 2.39 | 0.85 | 0.26 |
| TXn Land endC | 11.41 | 12.73 | 8.94 | 5.75 | 4.3 | 3.12 | 0.79 | 0.4 |
| TXn Ocean midC | 10.67 | 11.51 | 6.89 | 5.26 | 6.5 | 2.79 | 0.94 | 0.44 |
| TXn Ocean endC | 9.89 | 10.44 | 6.58 | 3.76 | 3.4 | 1.57 | 0.61 | 0.32 |

**Table B1.** Percentage of the global, land or ocean surface where the actual errors exceed the errors estimated on the basis of the formula "a-priori" using 5 ensemble members to estimate $\sigma$. Results for all temperature extreme metrics, derived from the CESM ensemble whose full size is 40 members. Calculations apply cosine-of-latitude weighting. Results for TNx are summaries of the behavior shown in Figure 3, i.e., the fraction of surface represented by locations where the error ratio is larger than 100%. Numbers under small $n$'s are affected by noise, as we randomly choose $n$ members from the full ensemble, only once. As can be gauged, the decreasing behavior of the fractions stabilizes for $n \geq 15$.

| Ens. size | 1 | 5 | 10 | 15 | 20 | 25 | 30 | 35 | 40 | 45 |
|---|---|---|---|---|---|---|---|---|---|---|
| TNx Global midC | 15.89 | 7.38 | 13.35 | 8.32 | 4.77 | 4.59 | 2.77 | 1.76 | 1.48 | 0.64 |
| TNx Global endC | 26.37 | 10.09 | 13.73 | 9.21 | 6.7 | 4.88 | 4.76 | 1.84 | 1.45 | 0.74 |
| TNx Land midC | 12.82 | 7.68 | 10.36 | 9.16 | 4.85 | 5.6 | 2.94 | 1.78 | 0.59 | 0.38 |
| TNx Land endC | 18.52 | 9.9 | 10.65 | 8.94 | 6.61 | 5.33 | 4.91 | 2.3 | 1.07 | 0.49 |
| TNx Ocean midC | 17.23 | 7.25 | 14.65 | 7.96 | 4.73 | 4.15 | 2.7 | 1.76 | 1.87 | 0.76 |
| TNx Ocean endC | 29.79 | 10.17 | 15.06 | 9.32 | 6.74 | 4.68 | 4.69 | 1.64 | 1.62 | 0.85 |
| TNn Global midC | 11.91 | 12.12 | 8.79 | 6.45 | 5.06 | 3.43 | 2.76 | 1.72 | 0.73 | 0.04 |
| TNn Global endC | 9.64 | 14.05 | 11.08 | 11.77 | 7.56 | 6.37 | 5.08 | 3.46 | 1.07 | 0.28 |
| TNn Land midC | 11.85 | 10.39 | 8.38 | 7.17 | 4.62 | 4.24 | 2.96 | 1.78 | 0.59 | 0.02 |
| TNn Land endC | 10.76 | 13.02 | 10.01 | 10.14 | 6.5 | 6.75 | 4.15 | 3.11 | 1.41 | 0.48 |
| TNn Ocean midC | 11.93 | 12.87 | 8.96 | 6.14 | 5.25 | 3.07 | 2.67 | 1.69 | 0.79 | 0.05 |
| TNn Ocean endC | 9.15 | 14.5 | 11.55 | 12.48 | 8.03 | 6.2 | 5.49 | 3.61 | 0.92 | 0.19 |
| TXx Global midC | 9.36 | 10.3 | 7.81 | 6.87 | 5.54 | 4.45 | 3.58 | 1.34 | 0.67 | 0.15 |
| TXx Global endC | 12.06 | 11.61 | 7.99 | 8.57 | 5.72 | 4.14 | 3.54 | 1.6 | 0.4 | 0.05 |
| TXx Land midC | 11.13 | 10.19 | 8.14 | 5.19 | 6.1 | 3.75 | 3.01 | 2.09 | 0.45 | 0.13 |
| TXx Land endC | 15.21 | 10.16 | 9.41 | 9.27 | 6.44 | 4.05 | 3.47 | 1.76 | 0.51 | 0 |
| TXx Ocean midC | 8.59 | 10.35 | 7.67 | 7.6 | 5.3 | 4.76 | 3.82 | 1.02 | 0.77 | 0.16 |
| TXx Ocean endC | 10.7 | 12.24 | 7.37 | 8.26 | 5.4 | 4.18 | 3.57 | 1.53 | 0.35 | 0.07 |
| TXn Global midC | 10.59 | 8.66 | 9.07 | 5.73 | 5.45 | 3.34 | 1.8 | 1.04 | 0.72 | 0.06 |
| TXn Global endC | 10.96 | 10.74 | 9.05 | 7.22 | 4.29 | 3.95 | 2.29 | 1.93 | 0.42 | 0.13 |
| TXn Land midC | 11.76 | 10.32 | 10.95 | 6.62 | 5.45 | 4.15 | 2.06 | 1.19 | 0.69 | 0 |
| TXn Land endC | 12.22 | 11.2 | 8.46 | 6.66 | 3.88 | 2.89 | 2.68 | 1.46 | 0.55 | 0.04 |
| TXn Ocean midC | 10.07 | 7.94 | 8.26 | 5.34 | 5.45 | 2.99 | 1.69 | 0.97 | 0.74 | 0.09 |
| TXn Ocean endC | 10.41 | 10.54 | 9.31 | 7.46 | 4.48 | 4.41 | 2.12 | 2.13 | 0.36 | 0.16 |

**Table B2.** Percentage of the global, land or ocean surface where the actual errors exceed the errors estimated on the basis of the formula "a-priori" using 5 ensemble members to estimate $\sigma$. Results for all temperature extreme metrics, derived from the CanESM ensemble whose full size is 50 members. Calculations apply cosine-of-latitude weighting. Numbers under small $n$'s are affected by noise, as we randomly choose $n$ members from the full ensemble, only once. As can be gauged, the decreasing behavior of the fractions stabilizes for $n \geq 15$.

| Ens. size | 1 | 5 | 10 | 15 | 20 | 25 | 30 | 35 |
|---|---|---|---|---|---|---|---|---|
| Rx5Day Global midC | 4.89 | 9.78 | 8.05 | 6.33 | 5.03 | 3.26 | 1.79 | 0.49 |
| Rx5Day Global endC | 5.83 | 8.84 | 7.72 | 6.46 | 4.58 | 3.22 | 1.61 | 0.39 |
| Rx5Day Land midC | 4.81 | 9.97 | 8.08 | 6.26 | 5.7 | 3.21 | 1.58 | 0.48 |
| Rx5Day Land endC | 5.42 | 8.71 | 7.16 | 6.07 | 4.91 | 2.61 | 1.66 | 0.38 |
| Rx5Day Ocean midC | 4.92 | 9.71 | 8.04 | 6.36 | 4.76 | 3.28 | 1.87 | 0.5 |
| Rx5Day Ocean endC | 5.99 | 8.89 | 7.95 | 6.63 | 4.45 | 3.48 | 1.59 | 0.39 |
| Rx1Day Global midC | 12.83 | 10.71 | 8.58 | 6.77 | 5.33 | 3.36 | 2.03 | 0.49 |
| Rx1Day Global endC | 12.45 | 10.52 | 7.94 | 6.22 | 4.78 | 3.2 | 1.73 | 0.47 |
| Rx1Day Land midC | 11.65 | 11.5 | 7.72 | 6.34 | 5.27 | 3.14 | 1.67 | 0.4 |
| Rx1Day Land endC | 11.39 | 9.82 | 7.83 | 5.24 | 4.6 | 3.04 | 1.72 | 0.29 |
| Rx1Day Ocean midC | 13.32 | 10.39 | 8.93 | 6.95 | 5.35 | 3.46 | 2.18 | 0.53 |
| Rx1Day Ocean endC | 12.89 | 10.8 | 7.99 | 6.63 | 4.86 | 3.26 | 1.73 | 0.54 |

**Table B3.** Percentage of the global, land or ocean surface where the actual errors exceed the errors estimated on the basis of the formula "a-priori" using 5 ensemble members to estimate $\sigma$. Results for the two precipitation extreme metrics, derived from the CESM ensemble whose full size is 40 members. Calculations apply cosine-of-latitude weighting. Results for Rx5Day are summaries of the behavior shown in Figure 4, i.e., the fraction of surface represented by locations where the error ratio is larger than 100%. Numbers under small $n$'s are affected by noise, as we randomly choose $n$ members from the full ensemble, only once. As can be gauged, the decreasing behavior of the fractions stabilizes for $n \geq 15$.

| Ens. size | 1 | 5 | 10 | 15 | 20 | 25 | 30 | 35 | 40 | 45 |
|---|---|---|---|---|---|---|---|---|---|---|
| Rx5Day Global midC | 12.54 | 10.76 | 7.72 | 7.31 | 6.97 | 5.03 | 4.1 | 2.09 | 0.91 | 0.32 |
| Rx5Day Global endC | 12.69 | 11.43 | 9.35 | 8.53 | 6.99 | 5.43 | 3.9 | 2.35 | 1.17 | 0.59 |
| Rx5Day Land midC | 13.03 | 11.12 | 7.6 | 7.53 | 6.63 | 4.84 | 4.68 | 2.23 | 1.13 | 0.4 |
| Rx5Day Land endC | 11.5 | 10.98 | 8.4 | 8.95 | 5.89 | 4.98 | 3.27 | 2.16 | 1.04 | 0.33 |
| Rx5Day Ocean midC | 12.33 | 10.59 | 7.77 | 7.21 | 7.11 | 5.12 | 3.85 | 2.02 | 0.82 | 0.29 |
| Rx5Day Ocean endC | 13.21 | 11.62 | 9.77 | 8.34 | 7.47 | 5.62 | 4.17 | 2.43 | 1.23 | 0.71 |
| Rx1Day Global midC | 12.52 | 11.77 | 8.69 | 9.11 | 6.74 | 5.69 | 4.09 | 2.56 | 1.33 | 0.28 |
| Rx1Day Global endC | 11.56 | 12.17 | 9.29 | 8.64 | 7.12 | 4.84 | 3.65 | 2.78 | 1.31 | 0.23 |
| Rx1Day Land midC | 11.43 | 11.78 | 8.94 | 8.58 | 5.58 | 5.89 | 3.71 | 2.33 | 1.33 | 0.15 |
| Rx1Day Land endC | 11.8 | 12.64 | 10.11 | 7.09 | 6.72 | 4.77 | 3.33 | 2.4 | 1.17 | 0.19 |
| Rx1Day Ocean midC | 13 | 11.77 | 8.58 | 9.35 | 7.24 | 5.6 | 4.25 | 2.67 | 1.33 | 0.34 |
| Rx1Day Ocean endC | 11.45 | 11.97 | 8.93 | 9.31 | 7.3 | 4.87 | 3.8 | 2.94 | 1.37 | 0.25 |

**Table B4.** Percentage of the global, land or ocean surface where the actual errors exceed the errors estimated on the basis of the formula "a-priori" using 5 ensemble members to estimate $\sigma$. Results for the two precipitation extreme metrics, derived from the CanESM ensemble whose full size is 50 members. Calculations apply cosine-of-latitude weighting. Numbers under small $n$'s are affected by noise, as we randomly choose $n$ members from the full ensemble, only once. As can be gauged, the decreasing behavior of the fractions stabilizes for $n \geq 15$.

# Appendix C: Additional Figures

**C1 Forced Component**

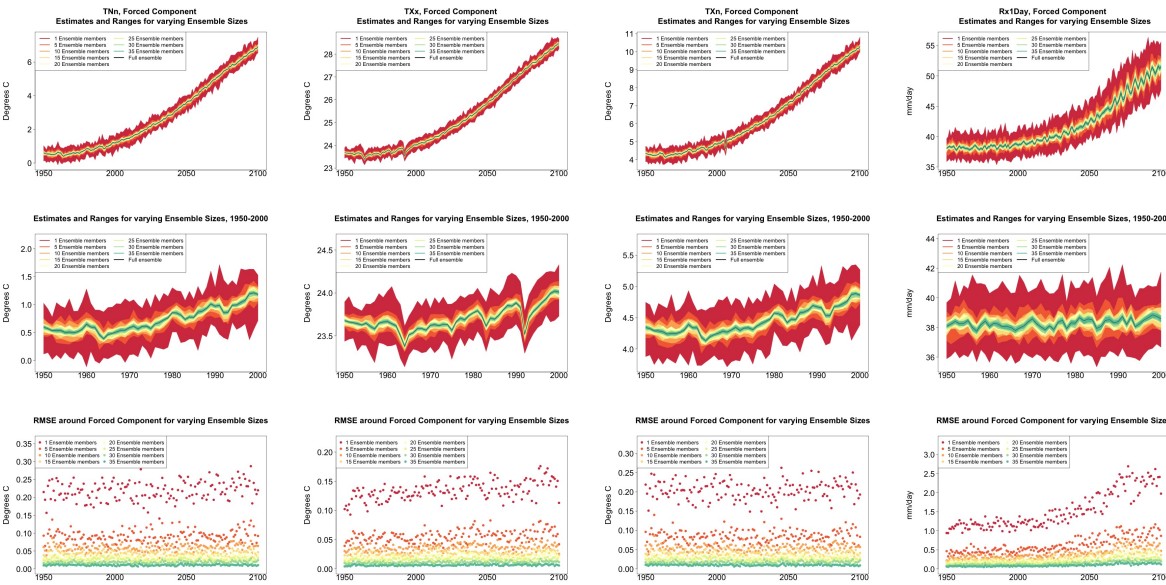

**Figure C1.** Like Figure 1 for the remaining metrics, derived from the CESM ensemble.

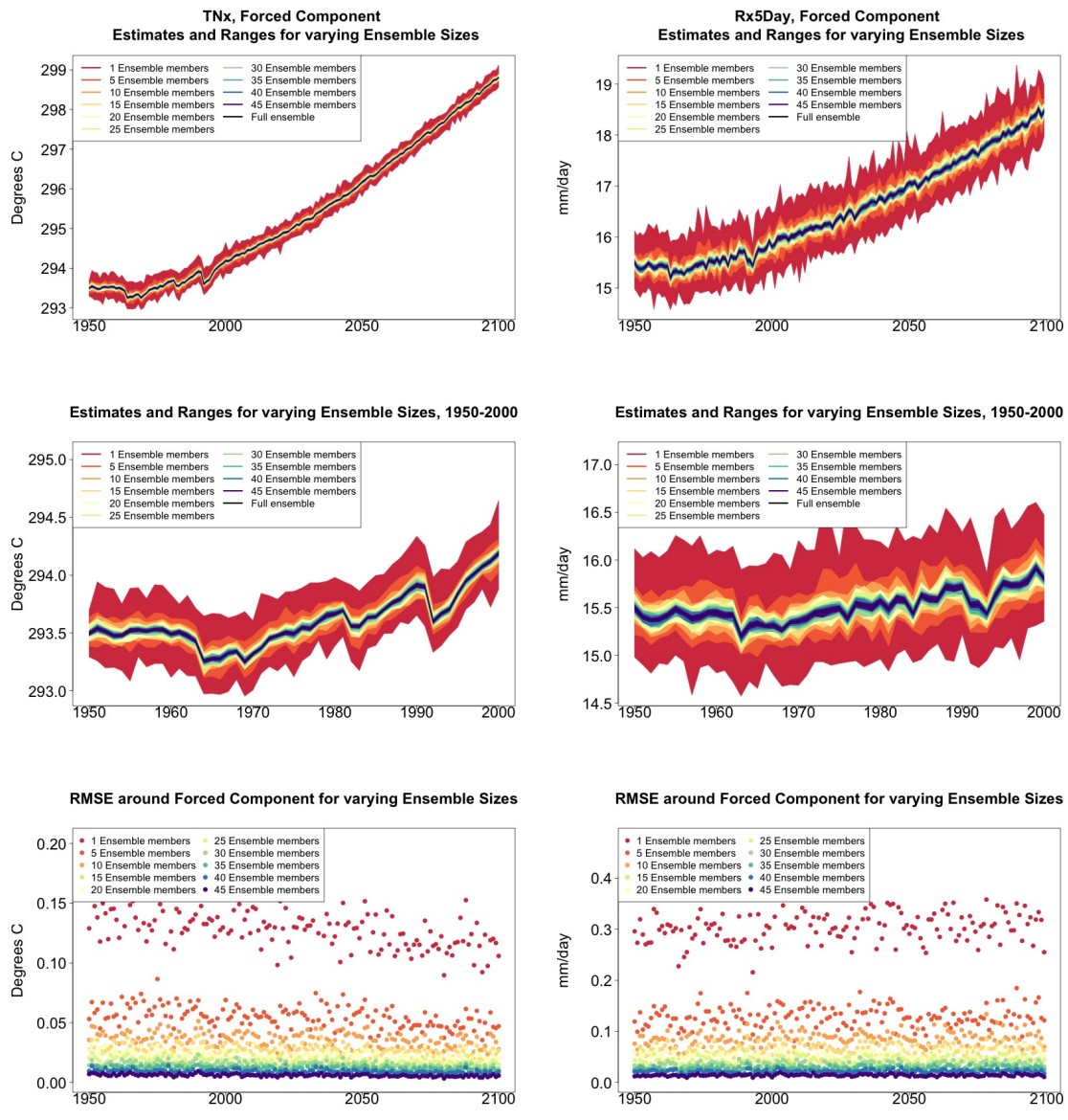

**Figure C2.** Like Figure 1 for metrics derived from the CanESM ensemble.

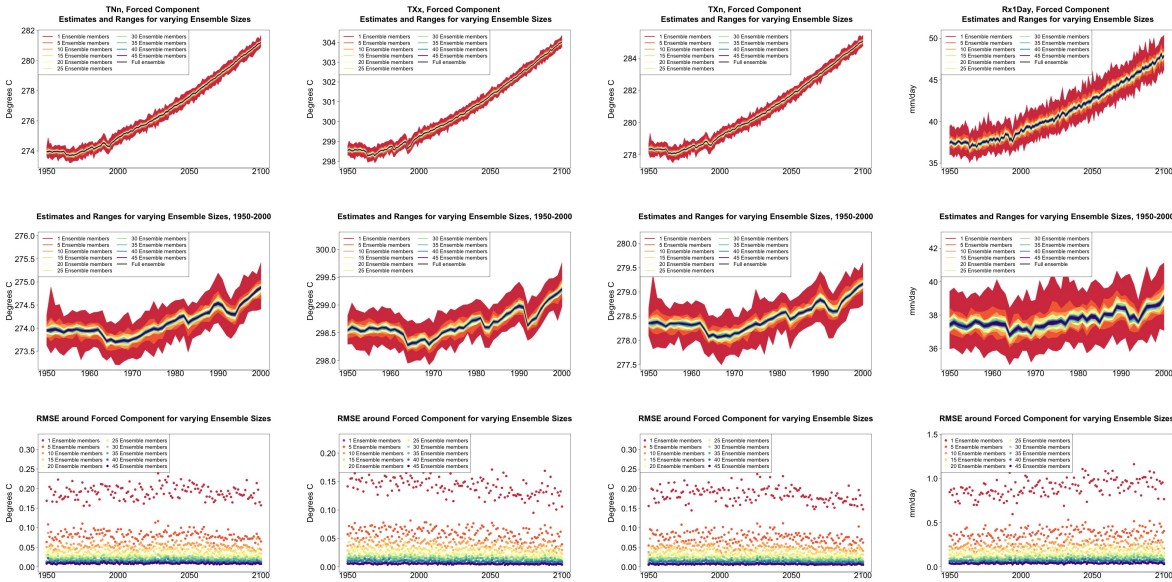

**Figure C3.** Like Figure C1 for metrics derived from the CanESM ensemble.

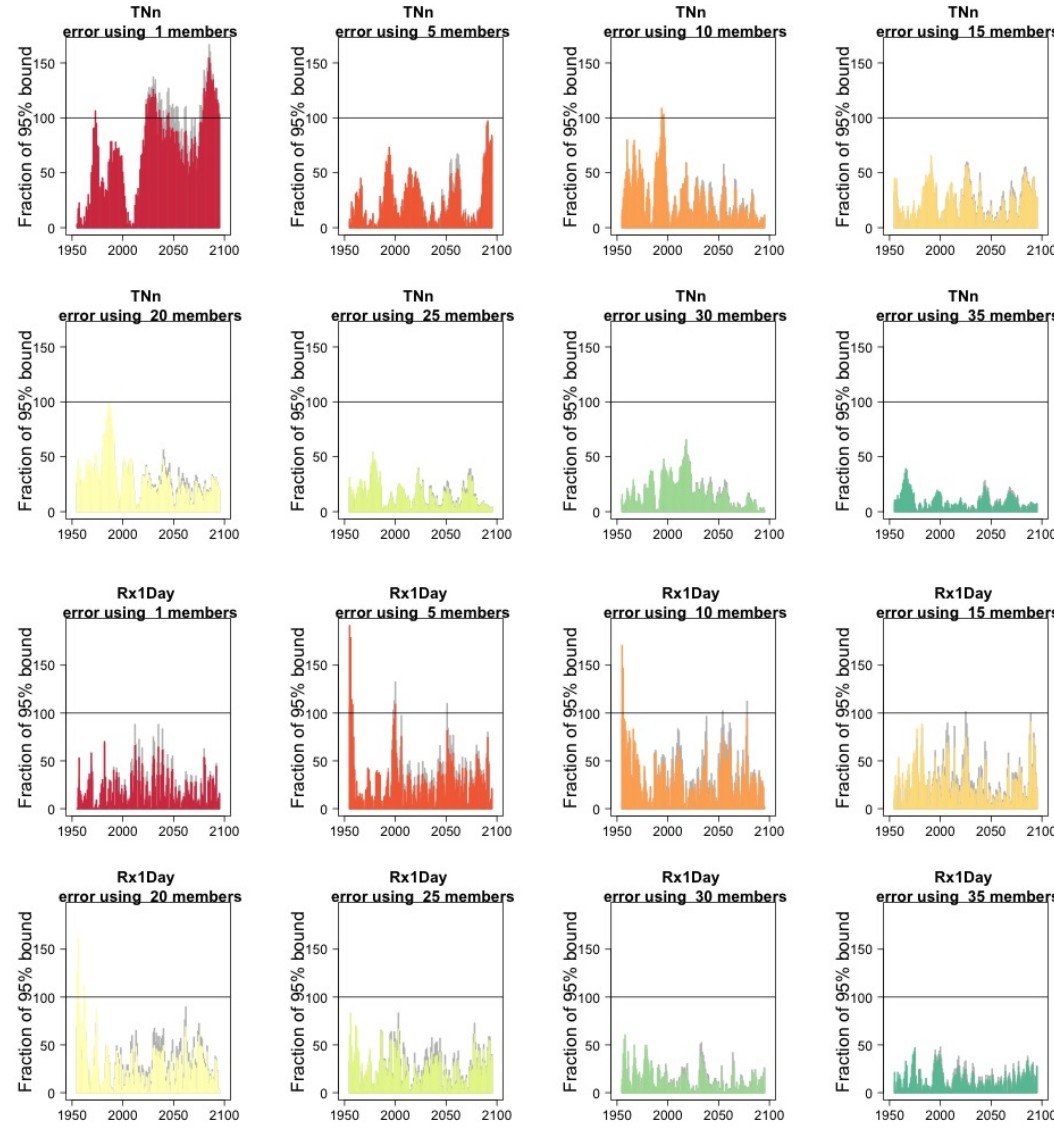

**Figure C4.** As Figure 2 for TNn and Rx1Day derived from the CESM ensemble.

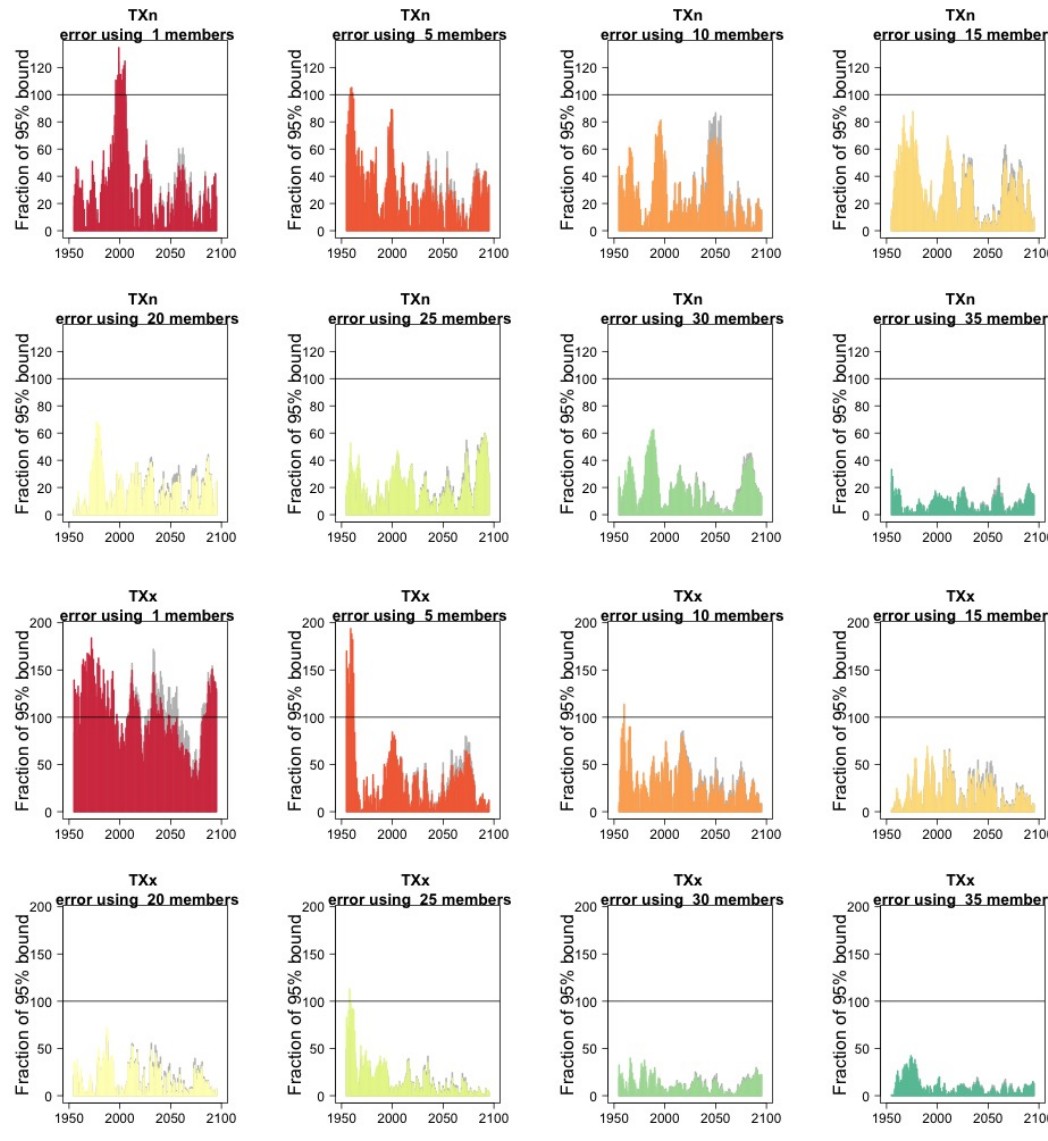

**Figure C5.** As Figure 2 for TXn and TXx derived from the CESM ensemble.

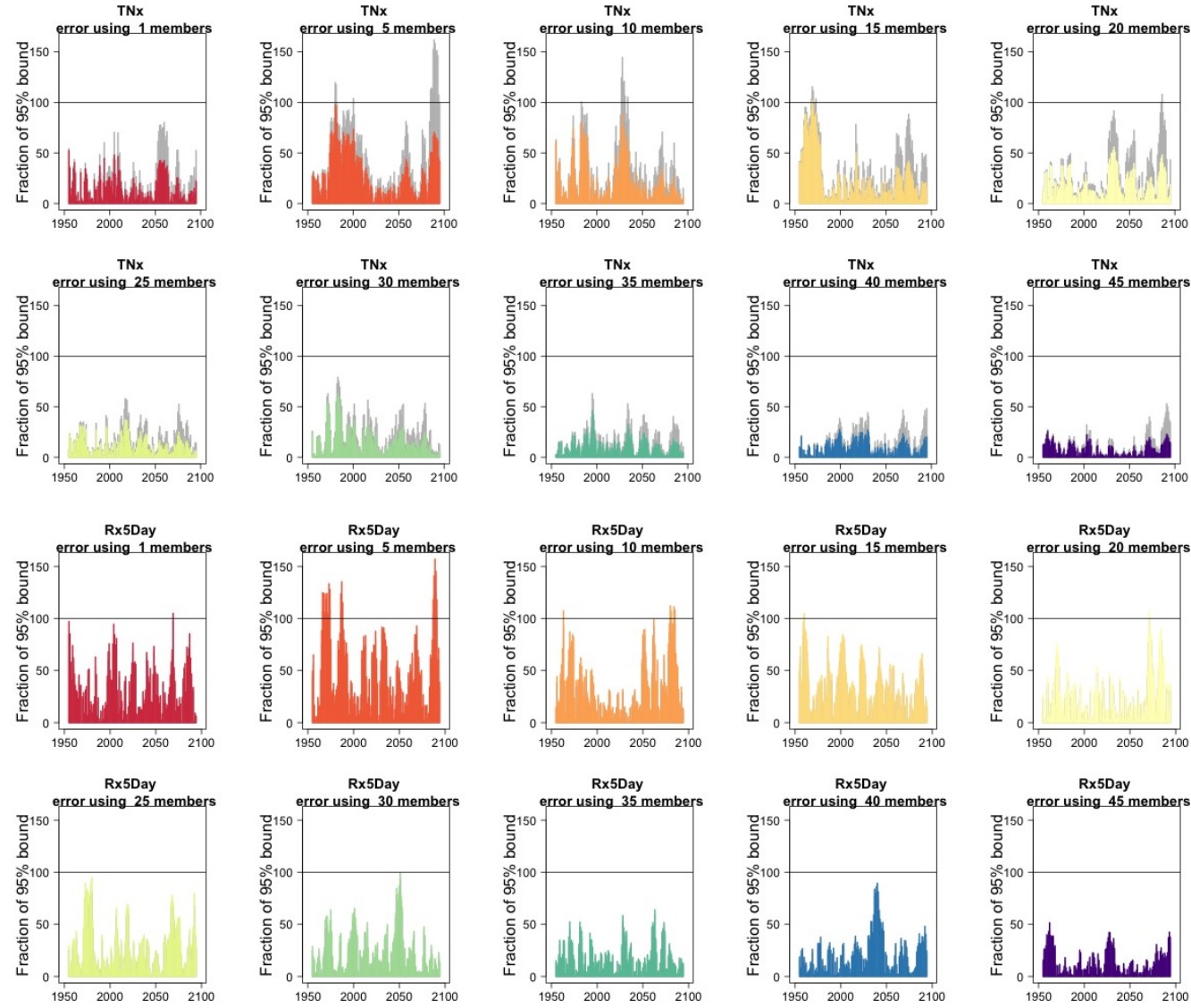

**Figure C6.** As Figure 2 but using the CanESM ensemble.

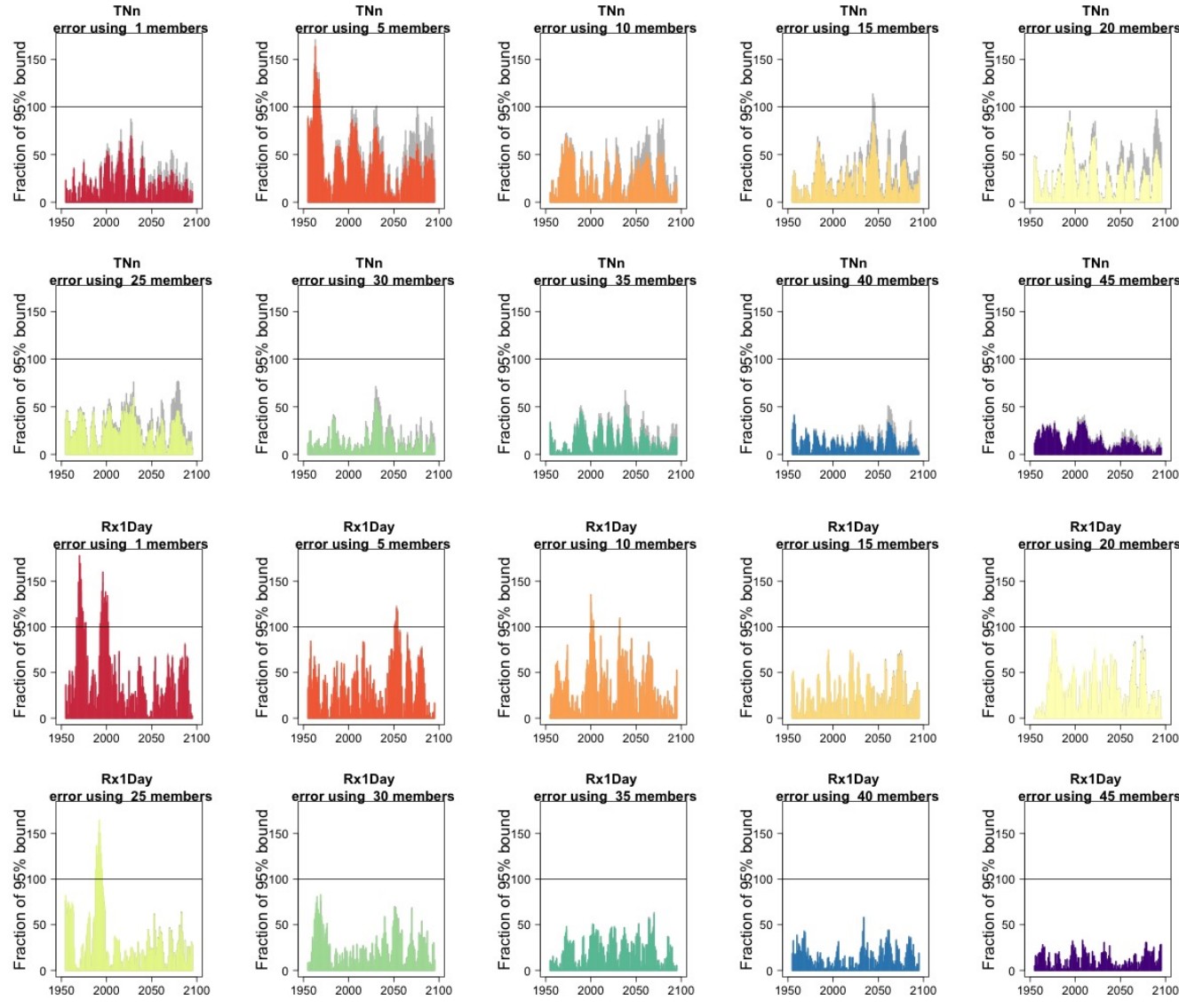

**Figure C7.** As Figure C6 for TNn and Rx1Day and using the CanESM ensemble.

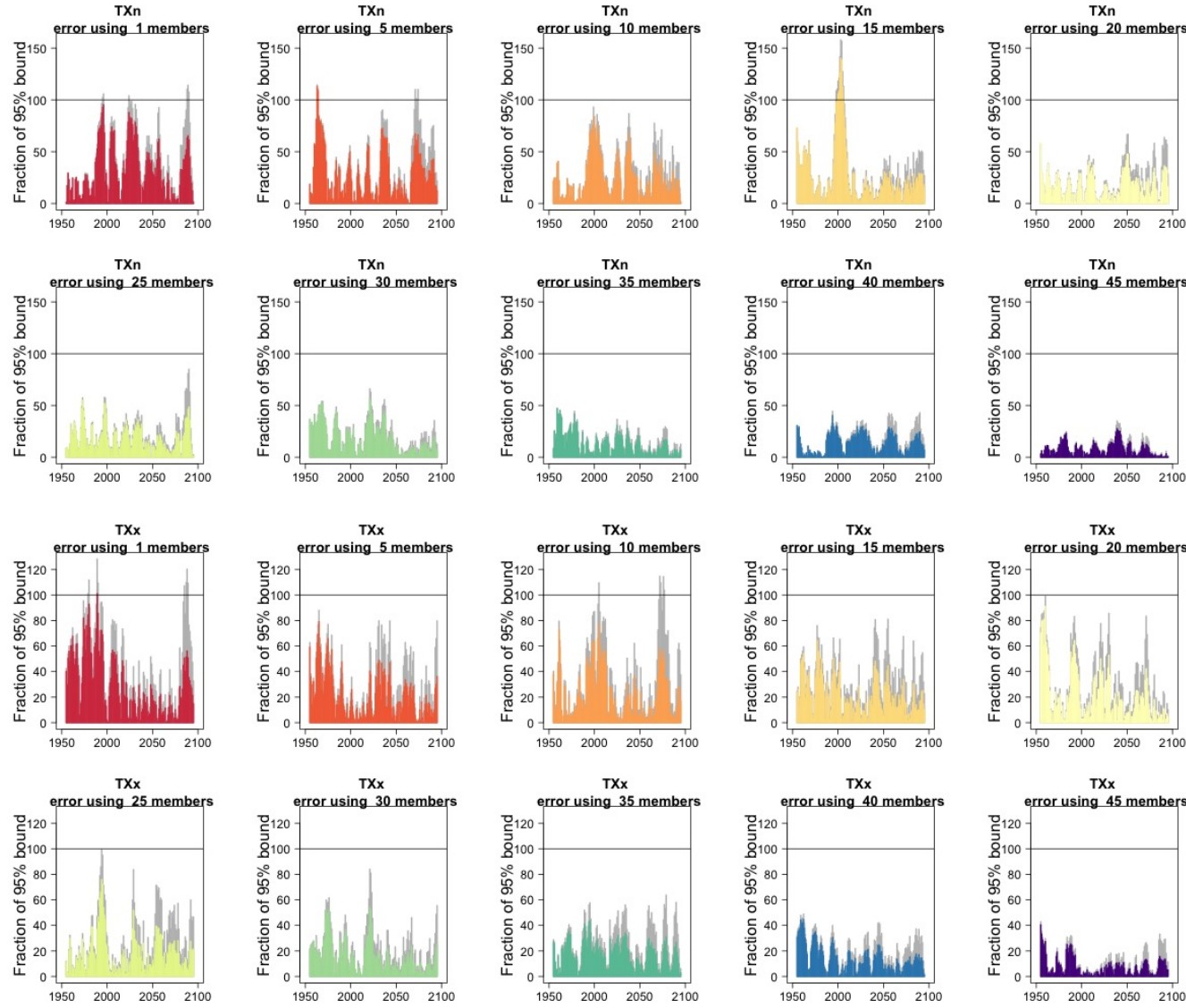

**Figure C8.** As Figure C6 for TXn and TXx and using the CanESM ensemble.

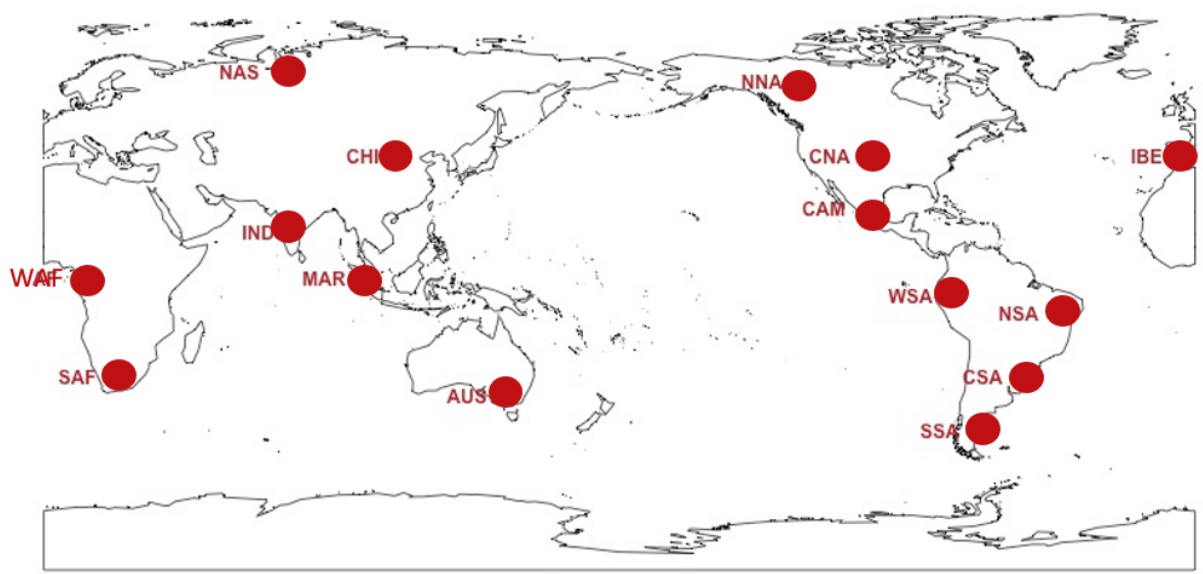

**Figure C9.** The fifteen locations at which we fit GEV distributions to the various quantities.

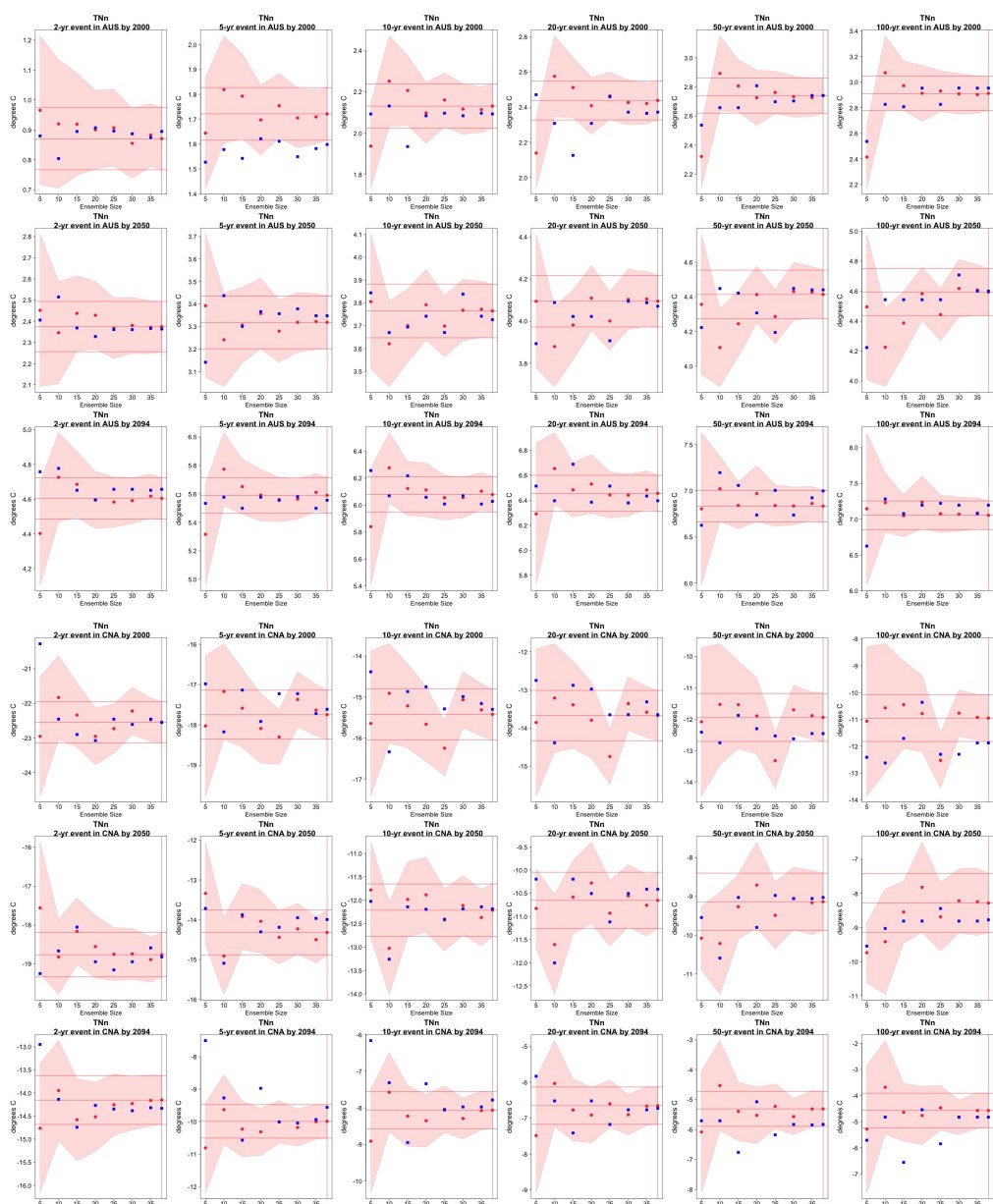

**Figure C10.** Return Levels for TNn from the CESM ensemble at (row-wise) 2000, 2050 and 2100 for (column-wise) 2-, 5-, 10-, 20-, 50-, 100-year return periods, based on estimating a GEV by using 11-yr windows of data around each date. In each plot, for increasing ensemble sizes along the x-axis (from 5 to the full ensemble, 40), the red dots indicate the central estimate, and the pink envelope represents the 95% confidence interval. The estimates based on the full ensemble, which we consider the truth, are also drawn across the plot for reference, as horizontal lines. The blue dots in each plot show the same quantities estimated by counting, i.e., computing the empirical cumulative distribution function of TNn on the basis of the same sample used for the estimation of the corresponding GEV parameters. The first three rows show results for a location in Australia while the following three rows show results for a location in Central North America (see Figure C9).

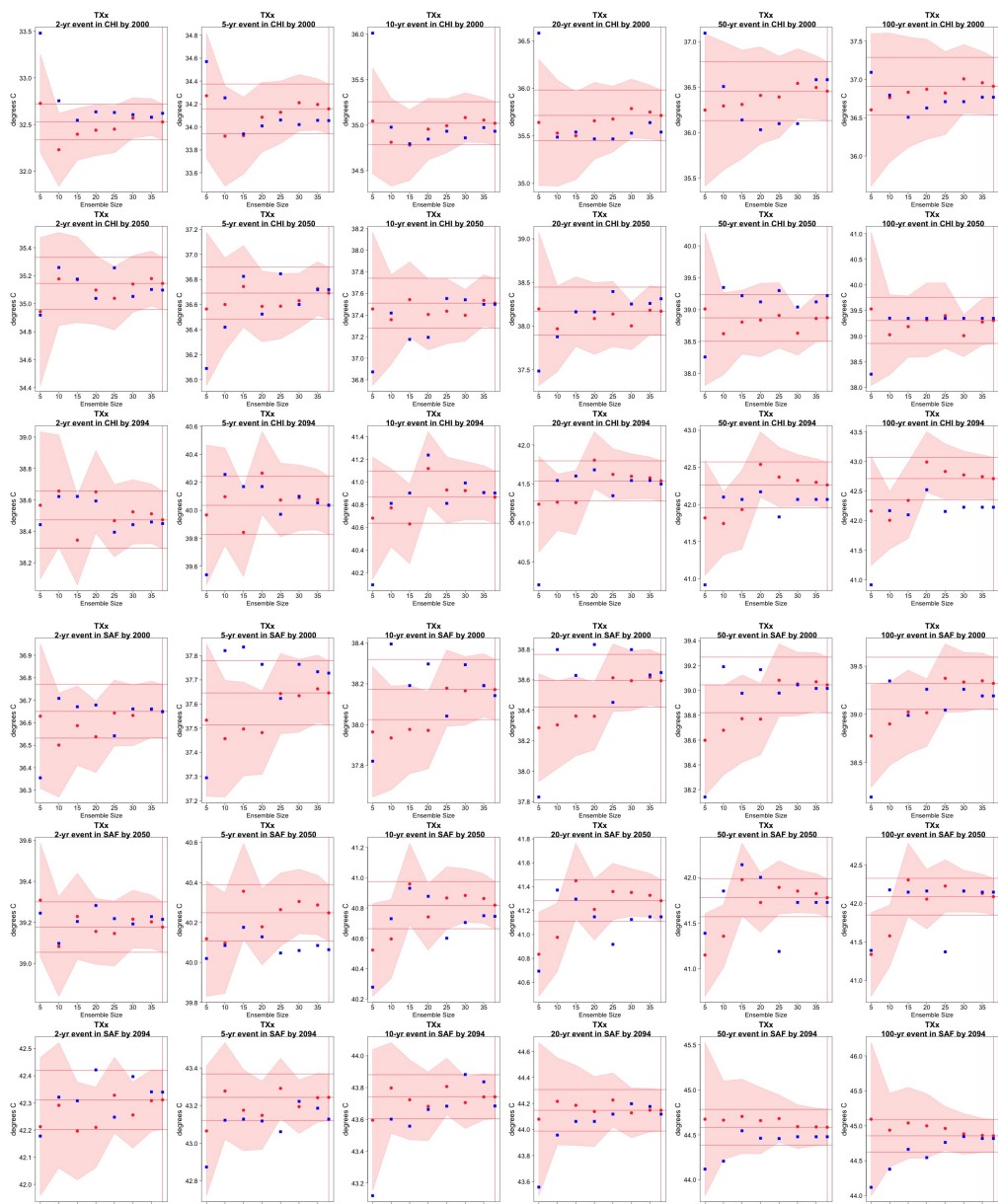

**Figure C11.** Return Levels for TXx from the CESM ensemble at (row-wise) 2000, 2050 and 2100 for (column-wise) 2-, 5-, 10-, 20-, 50-, 100-year return periods, based on estimating a GEV by using 11-yr windows of data around each date. In each plot, for increasing ensemble sizes along the x-axis (from 5 to the full ensemble, 40), the red dots indicate the central estimate, and the pink envelope represents the 95% confidence interval. The estimates based on the full ensemble, which we consider the truth, are also drawn across the plot for reference, as horizontal lines. The blue dots in each plot show the same quantities estimated by counting, i.e., computing the empirical cumulative distribution function of TXx on the basis of the same sample used for the estimation of the corresponding GEV parameters. The first three rows show results for a location in China while the following three rows show results for a location in Southern Africa (see Figure C9).

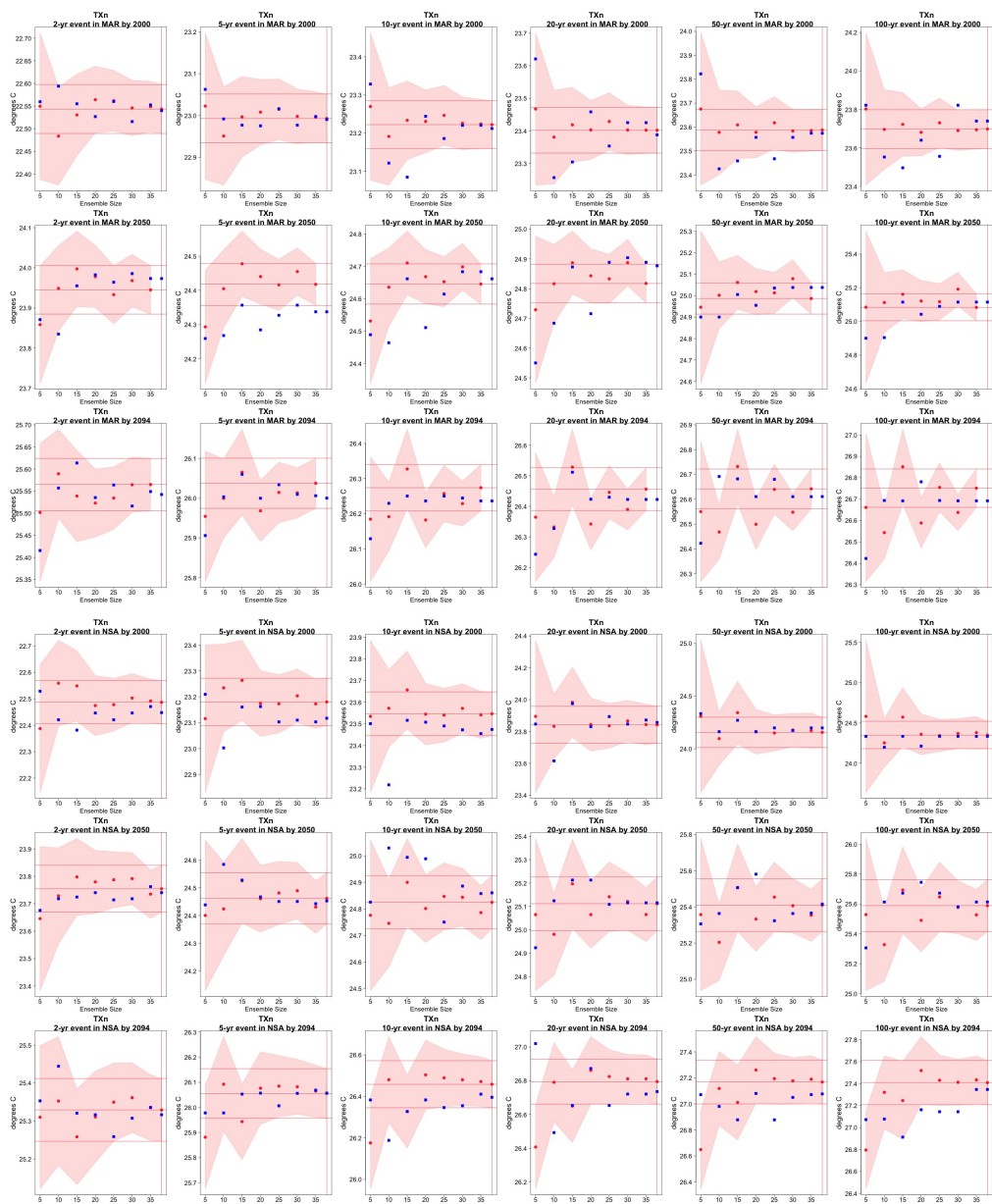

**Figure C12.** Return Levels for TXn from the CESM ensemble at (row-wise) 2000, 2050 and 2100 for (column-wise) 2-, 5-, 10-, 20-, 50-, 100-year return periods, based on estimating a GEV by using 11-yr windows of data around each date. In each plot, for increasing ensemble sizes along the x-axis (from 5 to the full ensemble, 40), the red dots indicate the central estimate, and the pink envelope represents the 95% confidence interval. The estimates based on the full ensemble, which we consider the truth, are also drawn across the plot for reference, as horizontal lines. The blue dots in each plot show the same quantities estimated by counting, i.e., computing the empirical cumulative distribution function of TXn on the basis of the same sample used for the estimation of the corresponding GEV parameters. The first three rows show results for a location on the Maritime Continent while the following three rows show results for a location in Northern South America (see Figure C9).

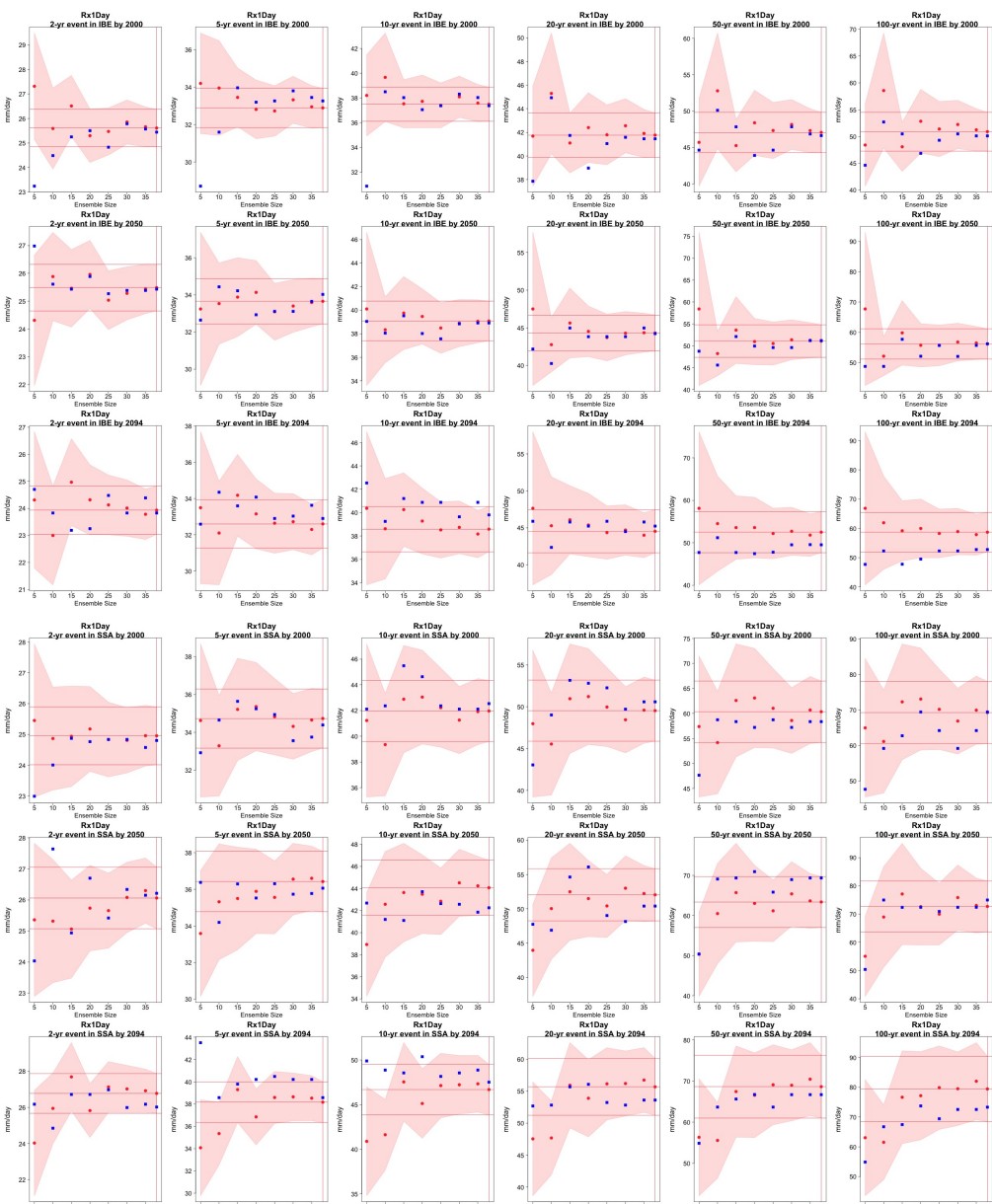

**Figure C13.** Return Levels for Rx1Day from the CESM ensemble at (row-wise) 2000, 2050 and 2100 for (column-wise) 2-, 5-, 10-, 20-, 50-, 100-year return periods, based on estimating a GEV by using 11-yr windows of data around each date. In each plot, for increasing ensemble sizes along the x-axis (from 5 to the full ensemble, 40), the red dots indicate the central estimate, and the pink envelope represents the 95% confidence interval. The estimates based on the full ensemble, which we consider the truth, are also drawn across the plot for reference, as horizontal lines. The blue dots in each plot show the same quantities estimated by counting, i.e., computing the empirical cumulative distribution function of Rx1Day on the basis of the same sample used for the estimation of the corresponding GEV parameters. The first three rows show results for a location on the Iberian peninsula while the following three rows show results for a location in Southern South America (see Figure C9).

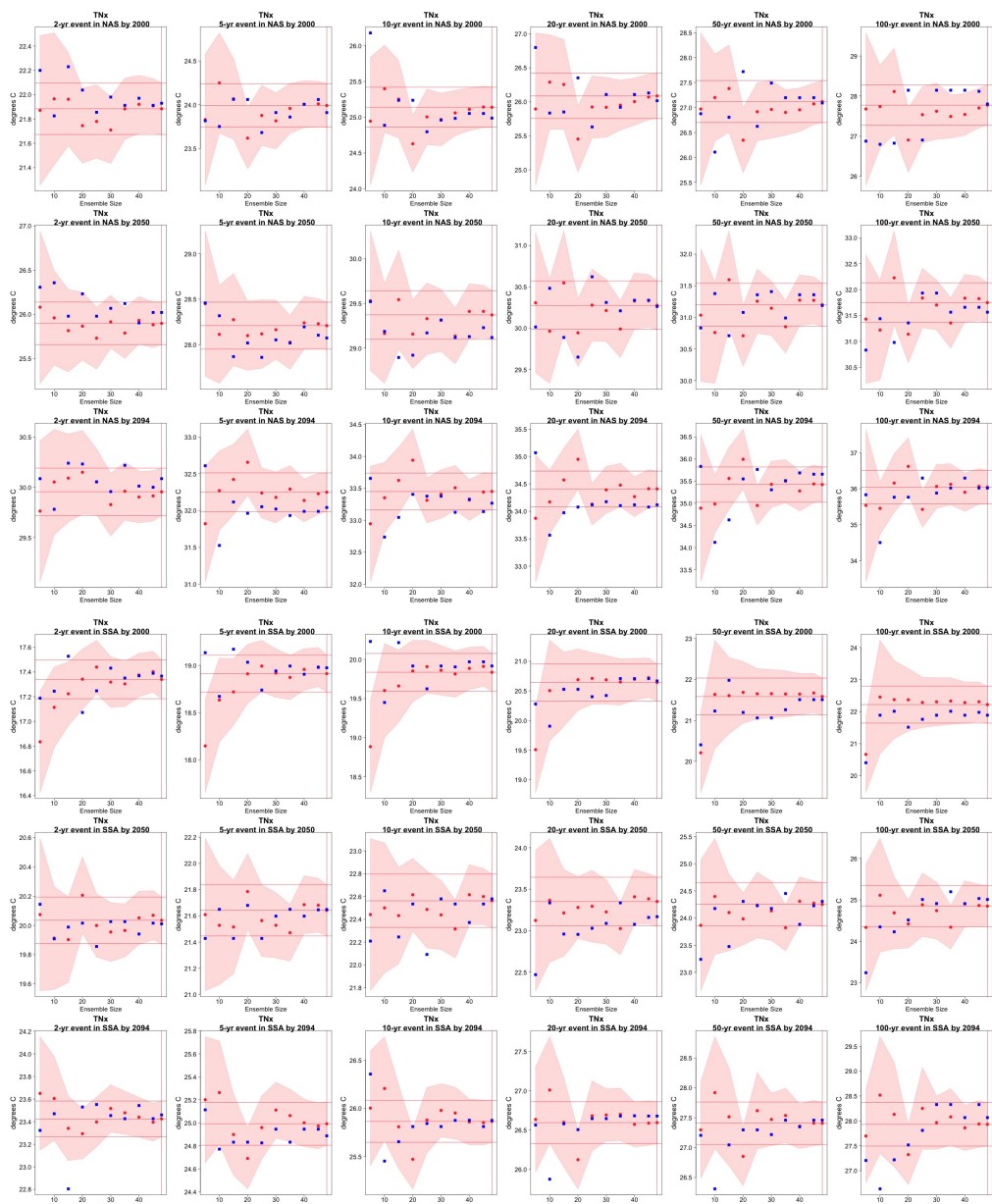

**Figure C14.** Return Levels for TNx from the CanESM ensemble at (row-wise) 2000, 2050 and 2100 for (column-wise) 2-, 5-, 10-, 20-, 50-, 100-year return periods, based on estimating a GEV by using 11-yr windows of data around each date. In each plot, for increasing ensemble sizes along the x-axis (from 5 to the full ensemble, 50), the red dots indicate the central estimate, and the pink envelope represents the 95% confidence interval. The estimates based on the full ensemble, which we consider the truth, are also drawn across the plot for reference, as horizontal lines. The blue dots in each plot show the same quantities estimated by counting, i.e., computing the empirical cumulative distribution function of TNx on the basis of the same sample used for the estimation of the corresponding GEV parameters. The first three rows show results for a location in Northern Asia while the following three rows show results for a location in Southern South America (see Figure C9).

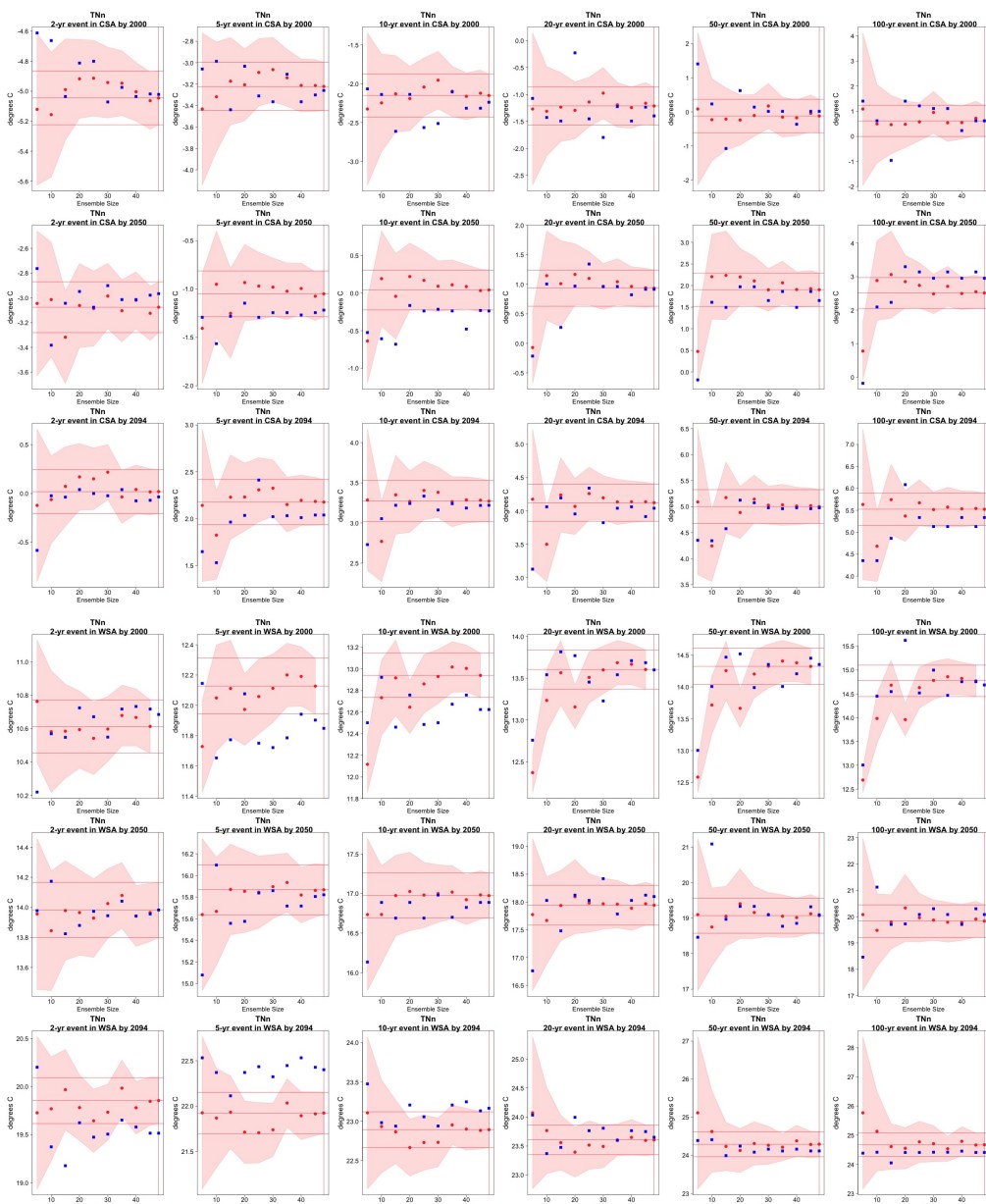

**Figure C15.** Return Levels for TNn from the CanESM ensemble at (row-wise) 2000, 2050 and 2100 for (column-wise) 2-, 5-, 10-, 20-, 50-, 100-year return periods, based on estimating a GEV by using 11-yr windows of data around each date. In each plot, for increasing ensemble sizes along the x-axis (from 5 to the full ensemble, 50), the red dots indicate the central estimate, and the pink envelope represents the 95% confidence interval. The estimates based on the full ensemble, which we consider the truth, are also drawn across the plot for reference, as horizontal lines. The blue dots in each plot show the same quantities estimated by counting, i.e., computing the empirical cumulative distribution function of TNn on the basis of the same sample used for the estimation of the corresponding GEV parameters. The first three rows show results for a location in Central South America while the following three rows show results for a location in Western South America (see Figure C9).

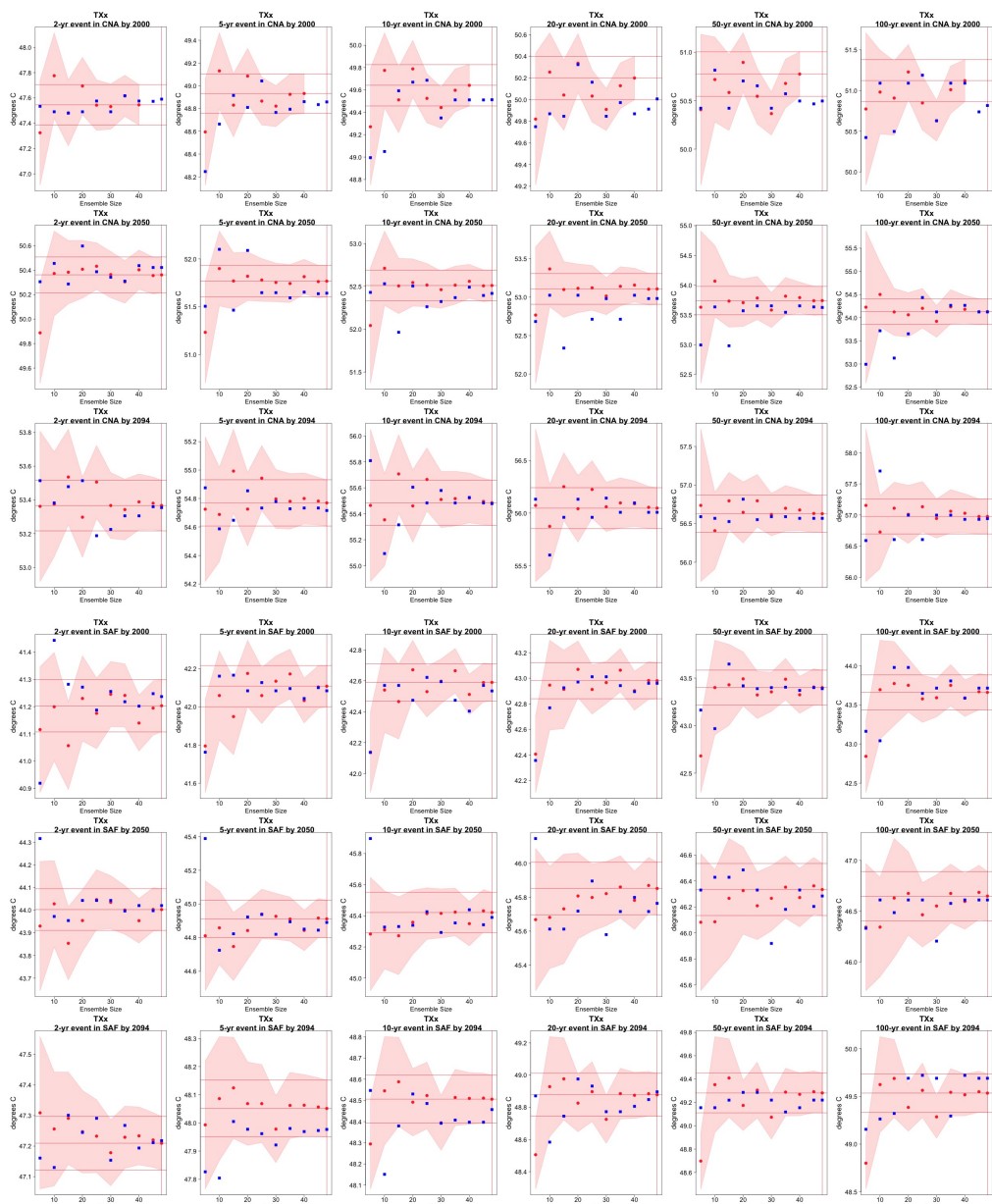

**Figure C16.** Return Levels for TXx from the CanESM ensemble at (row-wise) 2000, 2050 and 2100 for (column-wise) 2-, 5-, 10-, 20-, 50-, 100-year return periods, based on estimating a GEV by using 11-yr windows of data around each date. In each plot, for increasing ensemble sizes along the x-axis (from 5 to the full ensemble, 50), the red dots indicate the central estimate, and the pink envelope represents the 95% confidence interval. The estimates based on the full ensemble, which we consider the truth, are also drawn across the plot for reference, as horizontal lines. The blue dots in each plot show the same quantities estimated by counting, i.e., computing the empirical cumulative distribution function of TXx on the basis of the same sample used for the estimation of the corresponding GEV parameters. The first three rows show results for a location in Central North America while the following three rows show results for a location in Southern Africa (see Figure C9).

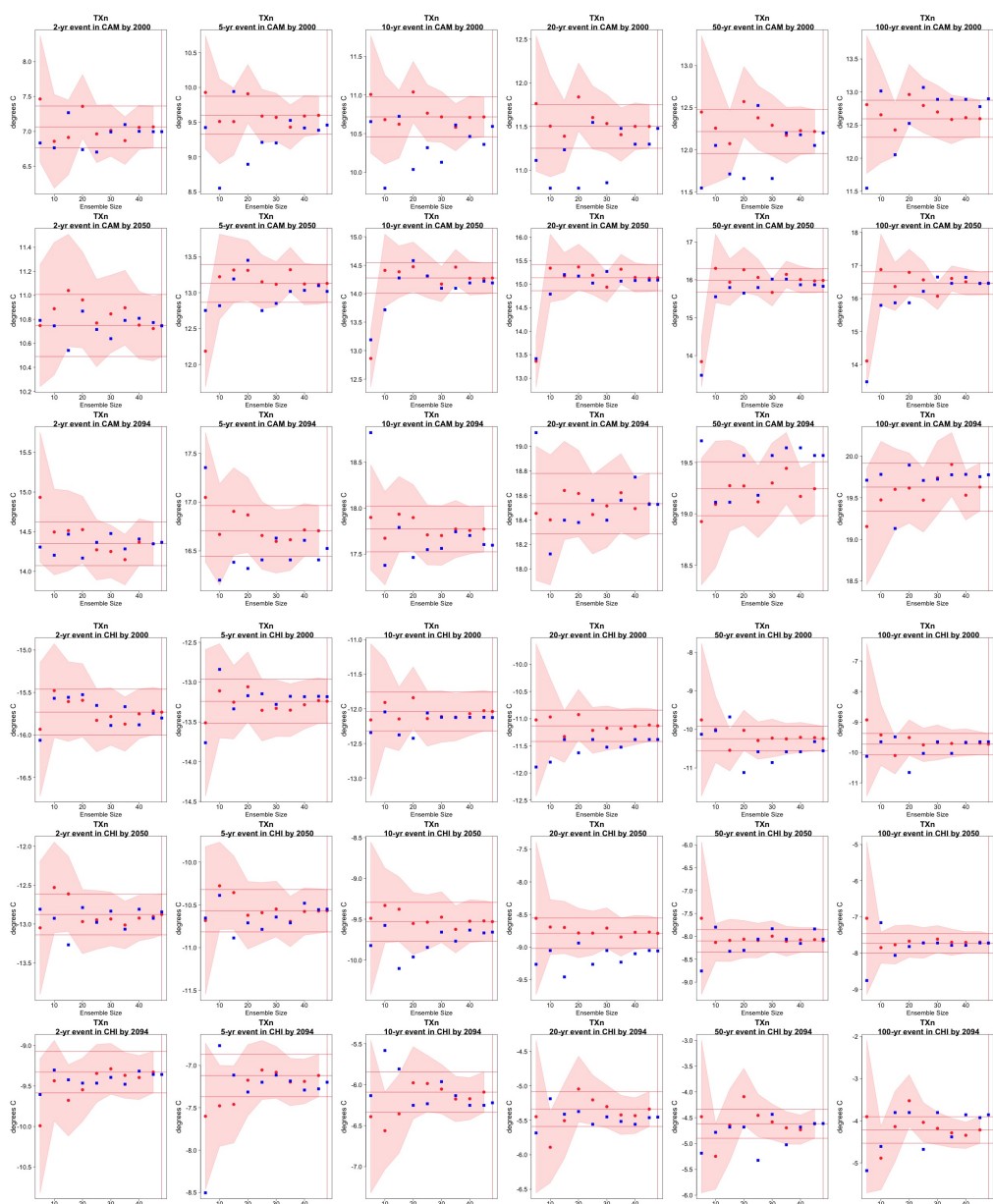

**Figure C17.** Return Levels for TXn from the CanESM ensemble at (row-wise) 2000, 2050 and 2100 for (column-wise) 2-, 5-, 10-, 20-, 50-, 100-year return periods, based on estimating a GEV by using 11-yr windows of data around each date. In each plot, for increasing ensemble sizes along the x-axis (from 5 to the full ensemble, 50), the red dots indicate the central estimate, and the pink envelope represents the 95% confidence interval. The estimates based on the full ensemble, which we consider the truth, are also drawn across the plot for reference, as horizontal lines. The blue dots in each plot show the same quantities estimated by counting, i.e., computing the empirical cumulative distribution function of TXn on the basis of the same sample used for the estimation of the corresponding GEV parameters. The first three rows show results for a location on the Central America while the following three rows show results for a location in China (see Figure C9).

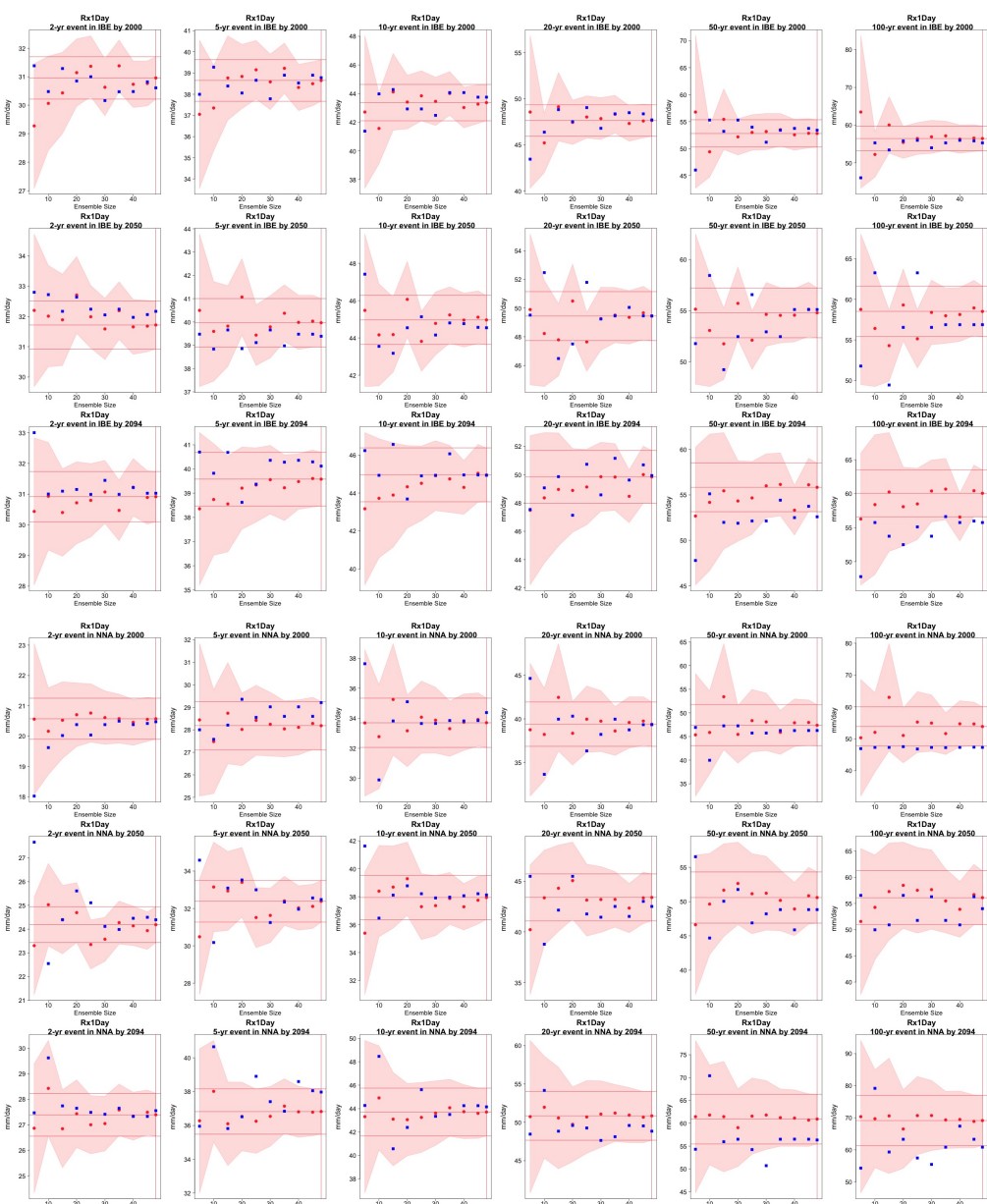

**Figure C18.** Return Levels for Rx1Day from the CanESM ensemble at (row-wise) 2000, 2050 and 2100 for (column-wise) 2-, 5-, 10-, 20-, 50-, 100-year return periods, based on estimating a GEV by using 11-yr windows of data around each date. In each plot, for increasing ensemble sizes along the x-axis (from 5 to the full ensemble, 50), the red dots indicate the central estimate, and the pink envelope represents the 95% confidence interval. The estimates based on the full ensemble, which we consider the truth, are also drawn across the plot for reference, as horizontal lines. The blue dots in each plot show the same quantities estimated by counting, i.e., computing the empirical cumulative distribution function of Rx1Day on the basis of the same sample used for the estimation of the corresponding GEV parameters. The first three rows show results for a location on the Iberian peninsula while the following three rows show results for a location in Northern North America (see Figure C9).

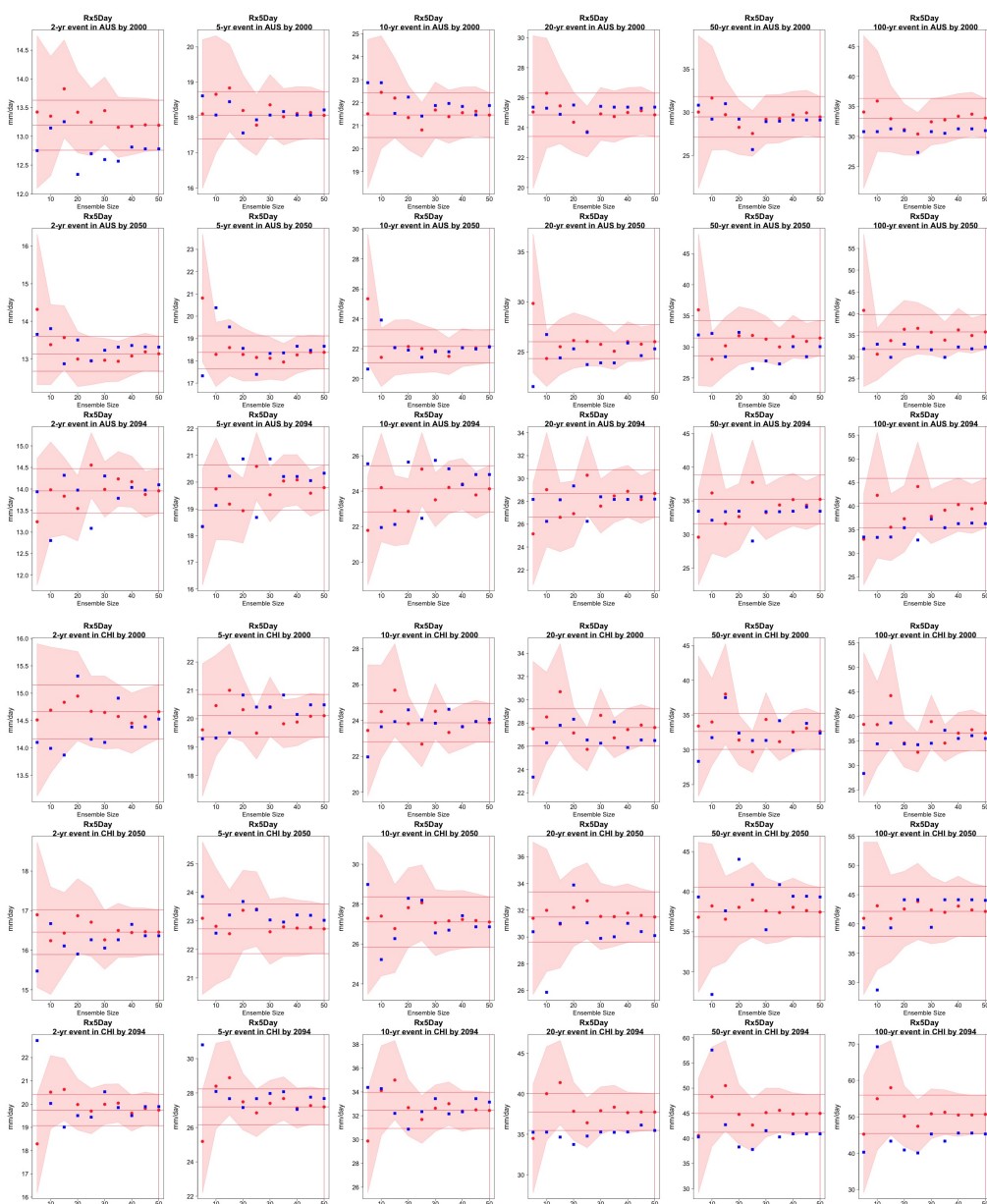

**Figure C19.** Return Levels for Rx5Day from the CanESM ensemble at (row-wise) 2000, 2050 and 2100 for (column-wise) 2-, 5-, 10-, 20-, 50-, 100-year return periods, based on estimating a GEV by using 11-yr windows of data around each date. In each plot, for increasing ensemble sizes along the x-axis (from 5 to the full ensemble, 50), the red dots indicate the central estimate, and the pink envelope represents the 95% confidence interval. The estimates based on the full ensemble, which we consider the truth, are also drawn across the plot for reference, as horizontal lines. The blue dots in each plot show the same quantities estimated by counting, i.e., computing the empirical cumulative distribution function of Rx5Day on the basis of the same sample used for the estimation of the corresponding GEV parameters. The first three rows show results for a location in Australia while the following three rows show results for a location in China (see Figure C9).

## C2 Variability

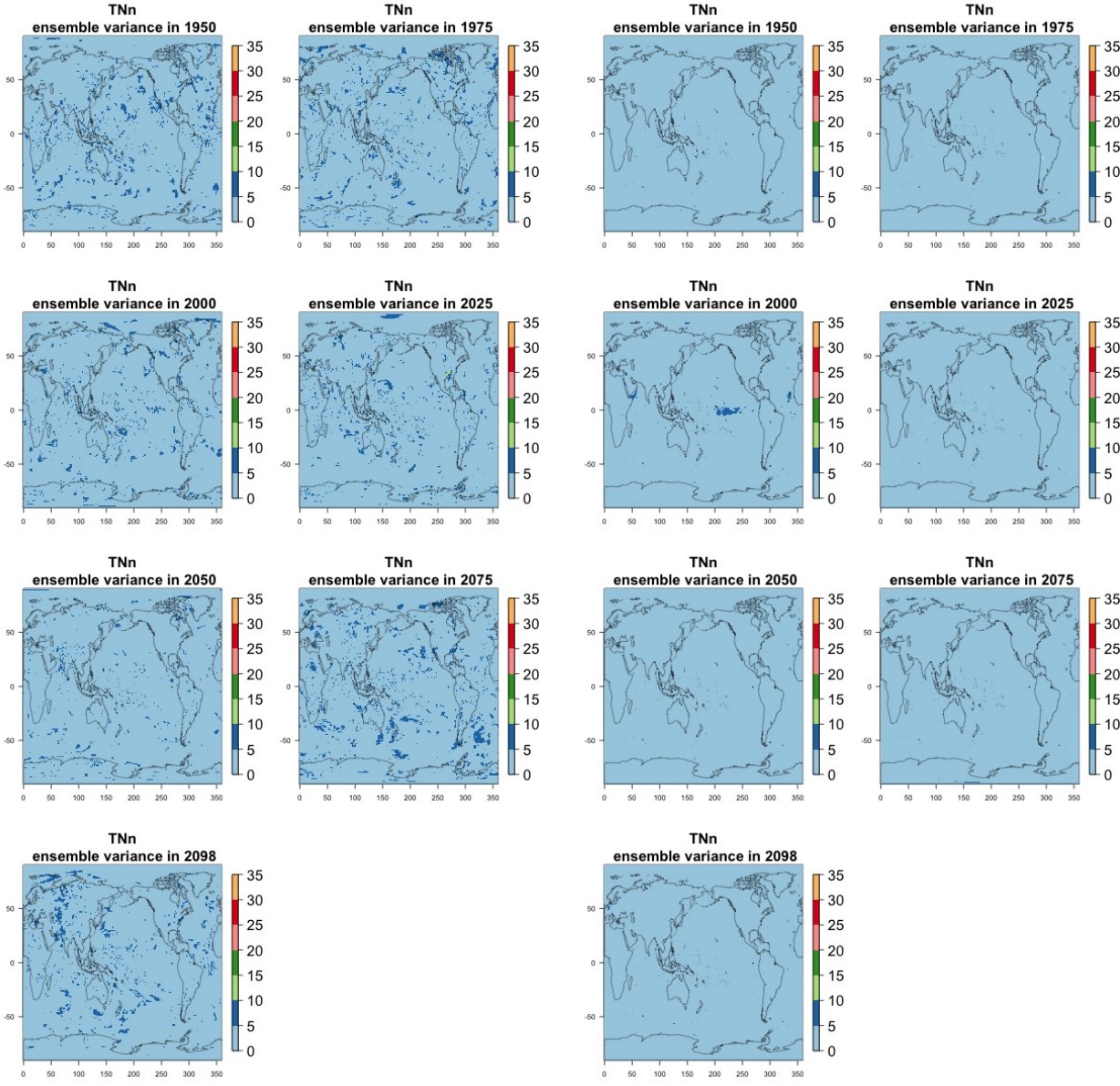

**Figure C20.** Estimating the ensemble variance for TNn in the CESM ensemble: Each plot corresponds to a year along the simulation length (1950, 1975, 2000, 2025, 2050, 2075, 2100). The color indicates the number of ensemble members needed to estimate a variance at that location that is statistically indistinguishable from that computed on the basis of the full 40-member ensemble. The results of the first two columns use only the specific year for each ensemble member. The results of the third and fourth columns enrich the samples by using 5 years around the specific date.

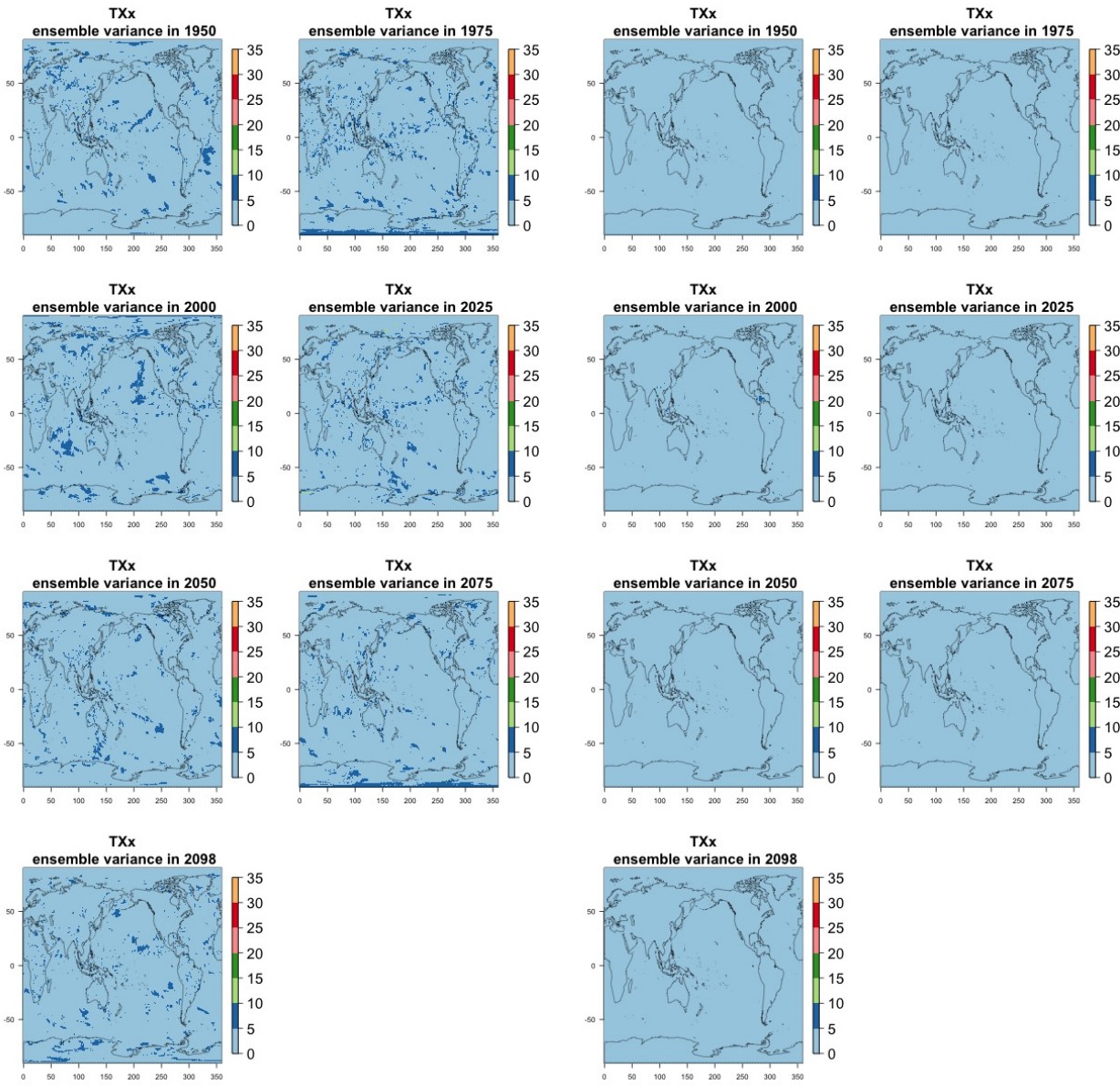

**Figure C21.** Estimating the ensemble variance for TXx in the CESM ensemble: Each plot corresponds to a year along the simulation length (1950, 1975, 2000, 2025, 2050, 2075, 2100). The color indicates the number of ensemble members needed to estimate a variance at that location that is statistically indistinguishable from that computed on the basis of the full 40-member ensemble. The results of the first two columns use only the specific year for each ensemble member. The results of the third and fourth columns enrich the samples by using 5 years around the specific date.

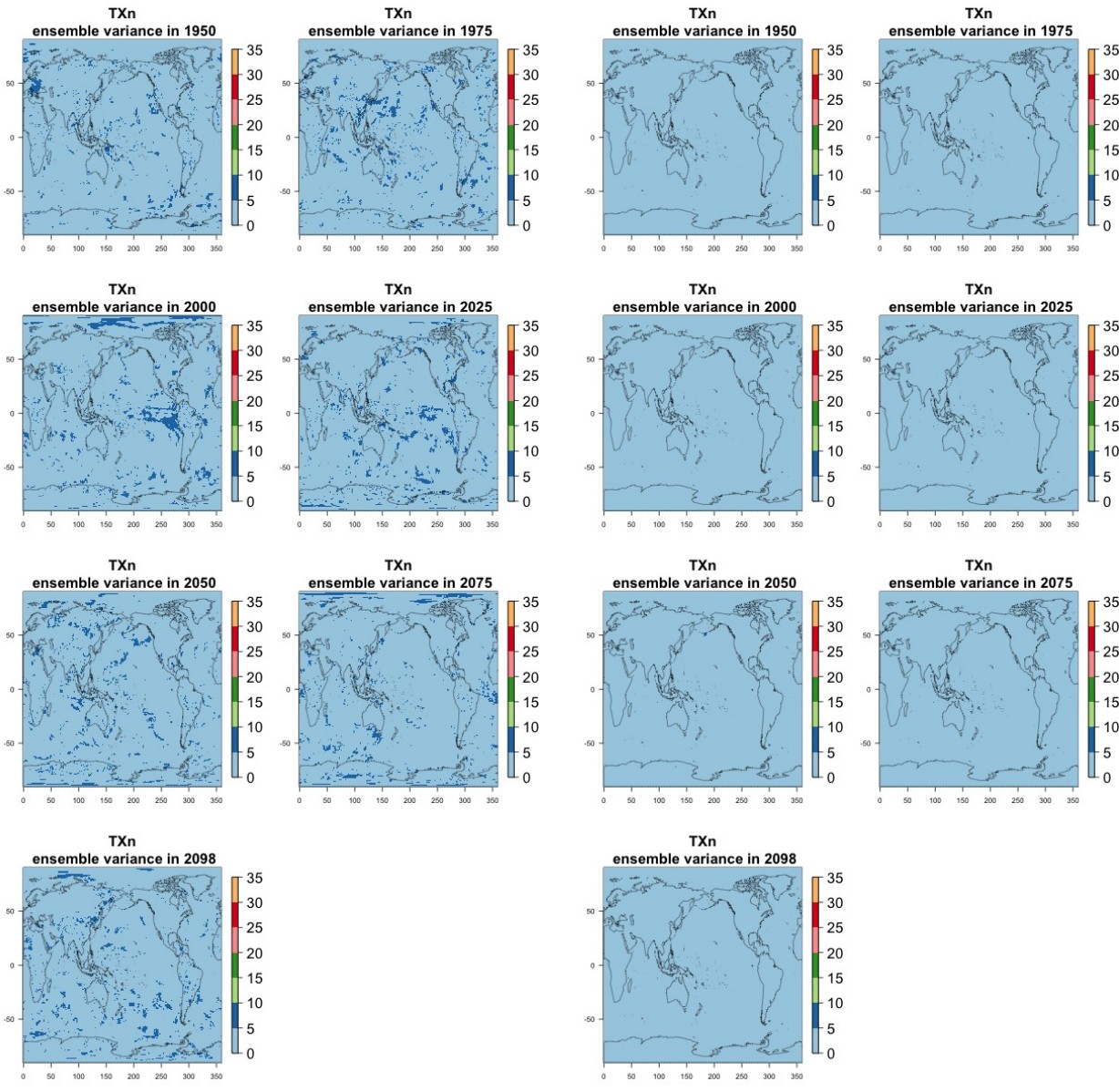

**Figure C22.** Estimating the ensemble variance for TXn in the CESM ensemble: Each plot corresponds to a year along the simulation length (1950, 1975, 2000, 2025, 2050, 2075, 2100). The color indicates the number of ensemble members needed to estimate a variance at that location that is statistically indistinguishable from that computed on the basis of the full 40-member ensemble. The results of the first two columns use only the specific year for each ensemble member. The results of the third and fourth columns enrich the samples by using 5 years around the specific date.

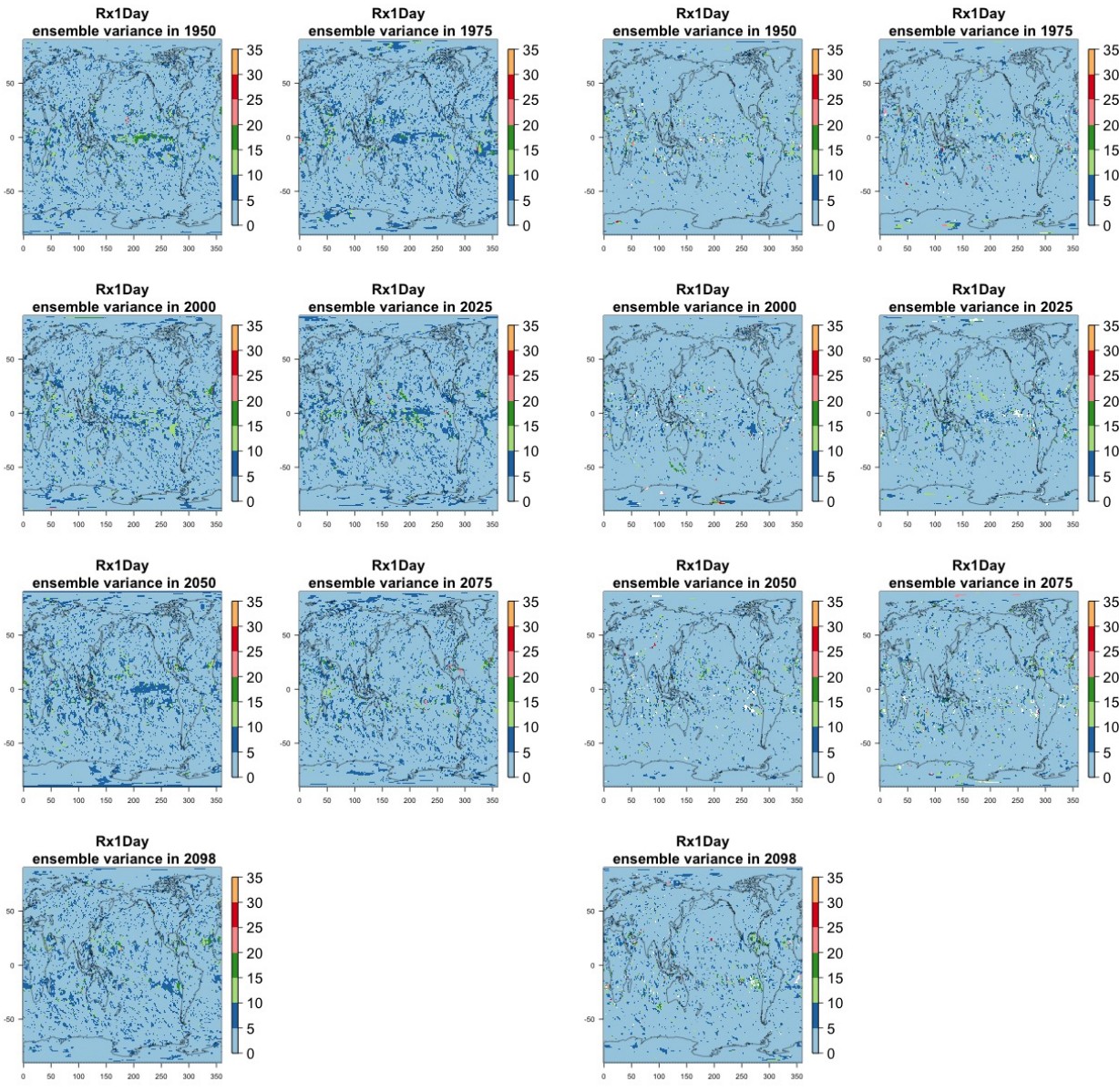

**Figure C23.** Estimating the ensemble variance for Rx1Day in the CESM ensemble: Each plot corresponds to a year along the simulation length (1950, 1975, 2000, 2025, 2050, 2075, 2100). The color indicates the number of ensemble members needed to estimate a variance at that location that is statistically indistinguishable from that computed on the basis of the full 40-member ensemble. The results of the first two columns use only the specific year for each ensemble member. The results of the third and fourth columns enrich the samples by using 5 years around the specific date.

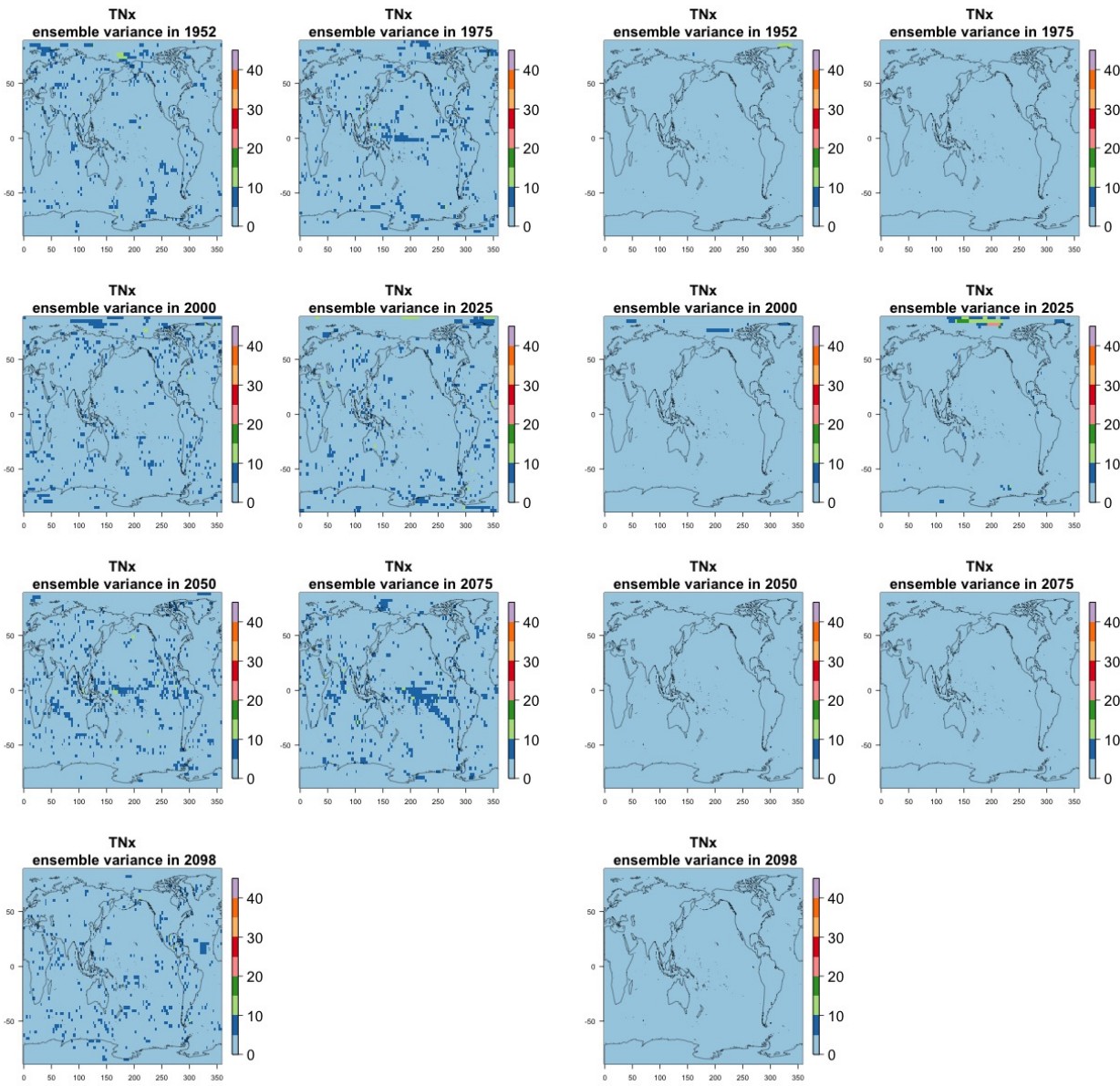

**Figure C24.** Estimating the ensemble variance for TNx in the CanESM ensemble: Each plot corresponds to a year along the simulation length (1950, 1975, 2000, 2025, 2050, 2075, 2100). The color indicates the number of ensemble members needed to estimate a variance at that location that is statistically indistinguishable from that computed on the basis of the full 50-member ensemble. The results of the first two columns use only the specific year for each ensemble member. The results of the third and fourth columns enrich the samples by using 5 years around the specific date.

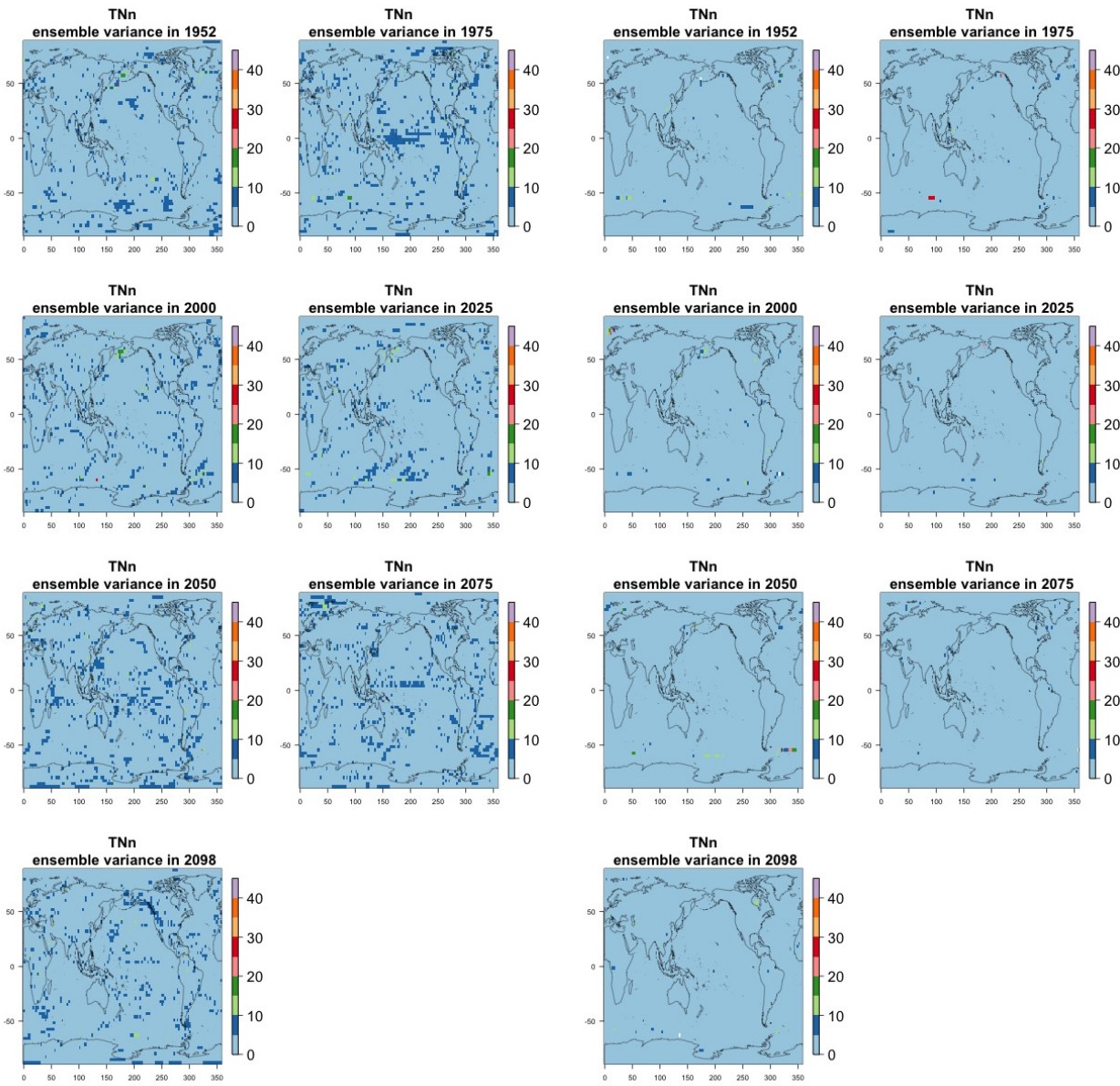

**Figure C25.** Estimating the ensemble variance for TNn in the CanESM ensemble: Each plot corresponds to a year along the simulation length (1950, 1975, 2000, 2025, 2050, 2075, 2100). The color indicates the number of ensemble members needed to estimate a variance at that location that is statistically indistinguishable from that computed on the basis of the full 50-member ensemble. The results of the first two columns use only the specific year for each ensemble member. The results of the third and fourth columns enrich the samples by using 5 years around the specific date.

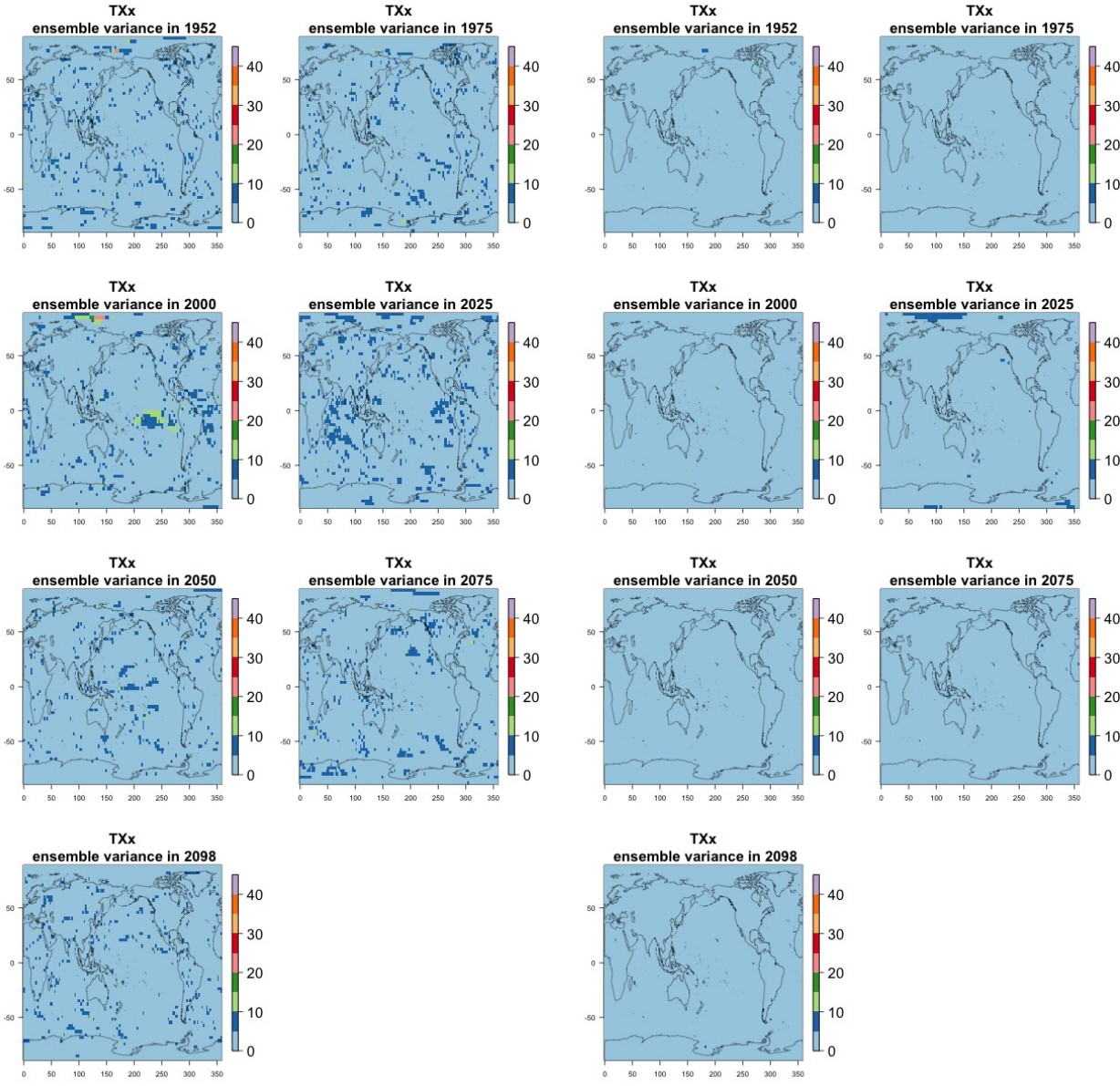

**Figure C26.** Estimating the ensemble variance for TXx in the CanESM ensemble: Each plot corresponds to a year along the simulation length (1950, 1975, 2000, 2025, 2050, 2075, 2100). The color indicates the number of ensemble members needed to estimate a variance at that location that is statistically indistinguishable from that computed on the basis of the full 50-member ensemble. The results of the first two columns use only the specific year for each ensemble member. The results of the third and fourth columns enrich the samples by using 5 years around the specific date.

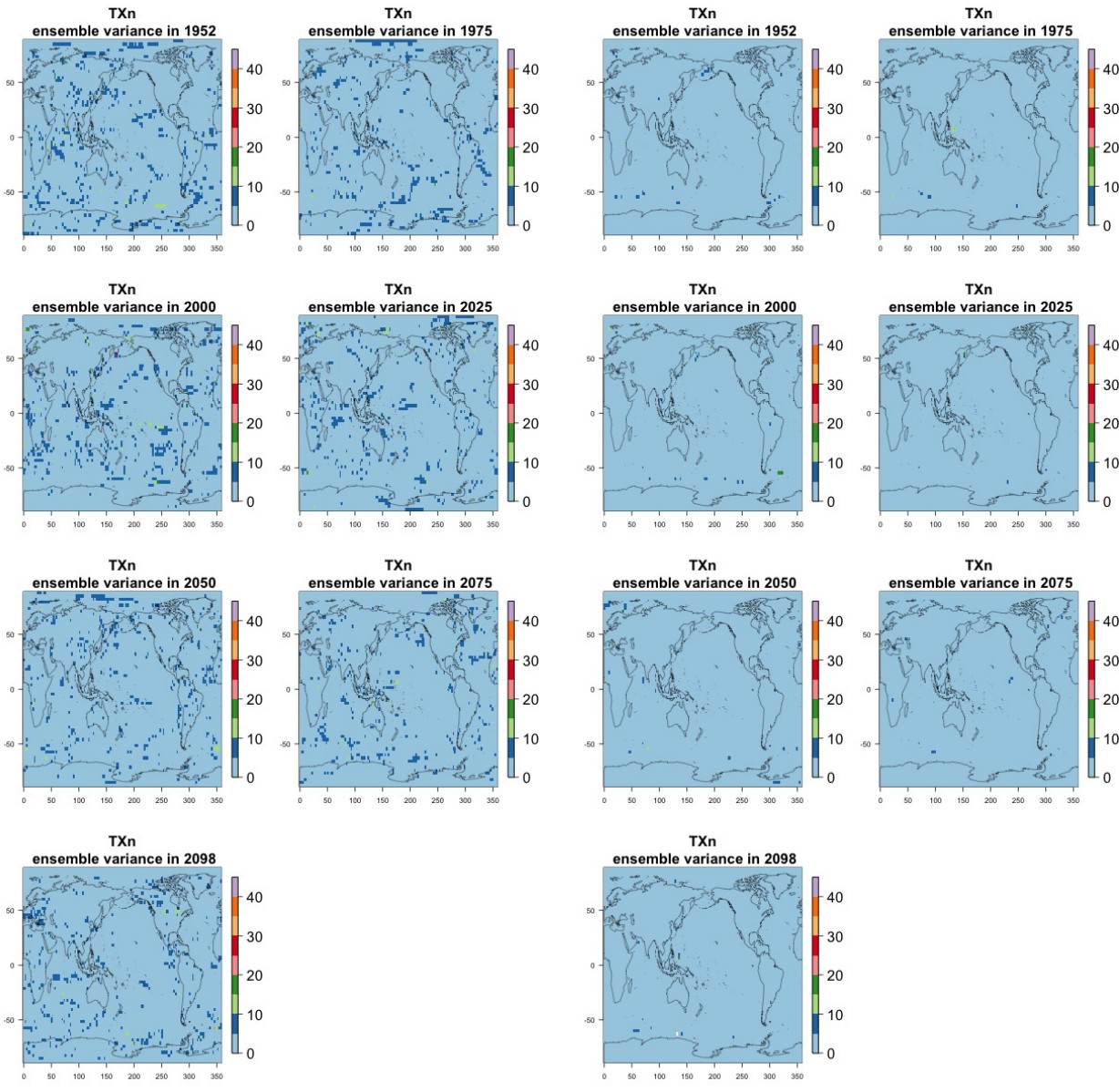

**Figure C27.** Estimating the ensemble variance for TXn in the CanESM ensemble: Each plot corresponds to a year along the simulation length (1950, 1975, 2000, 2025, 2050, 2075, 2100). The color indicates the number of ensemble members needed to estimate a variance at that location that is statistically indistinguishable from that computed on the basis of the full 50-member ensemble. The results of the first two columns use only the specific year for each ensemble member. The results of the third and fourth columns enrich the samples by using 5 years around the specific date.

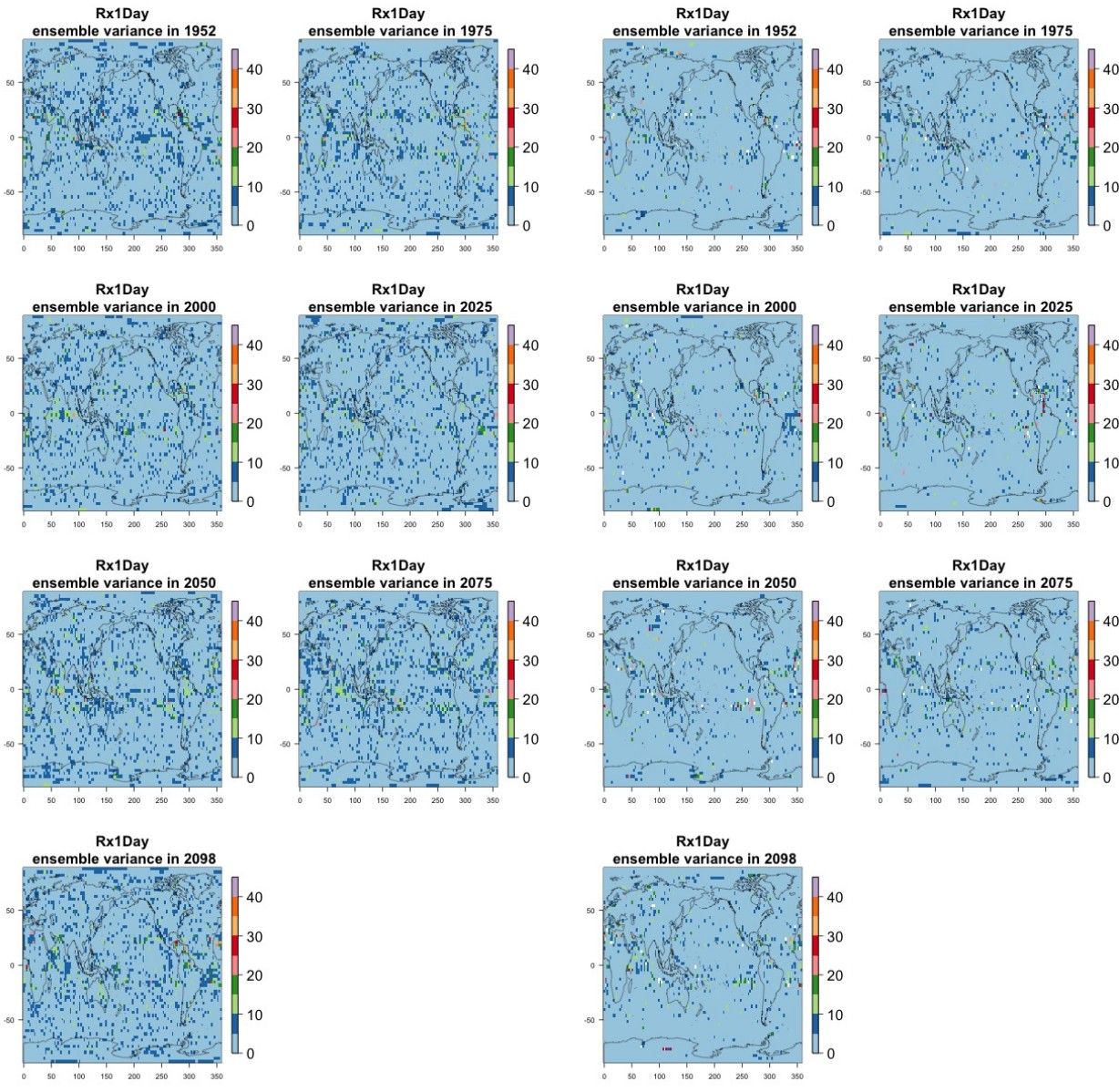

**Figure C28.** Estimating the ensemble variance for Rx1Day in the CanESM ensemble: Each plot corresponds to a year along the simulation length (1950, 1975, 2000, 2025, 2050, 2075, 2100). The color indicates the number of ensemble members needed to estimate a variance at that location that is statistically indistinguishable from that computed on the basis of the full 50-member ensemble. The results of the first two columns use only the specific year for each ensemble member. The results of the third and fourth columns enrich the samples by using 5 years around the specific date.

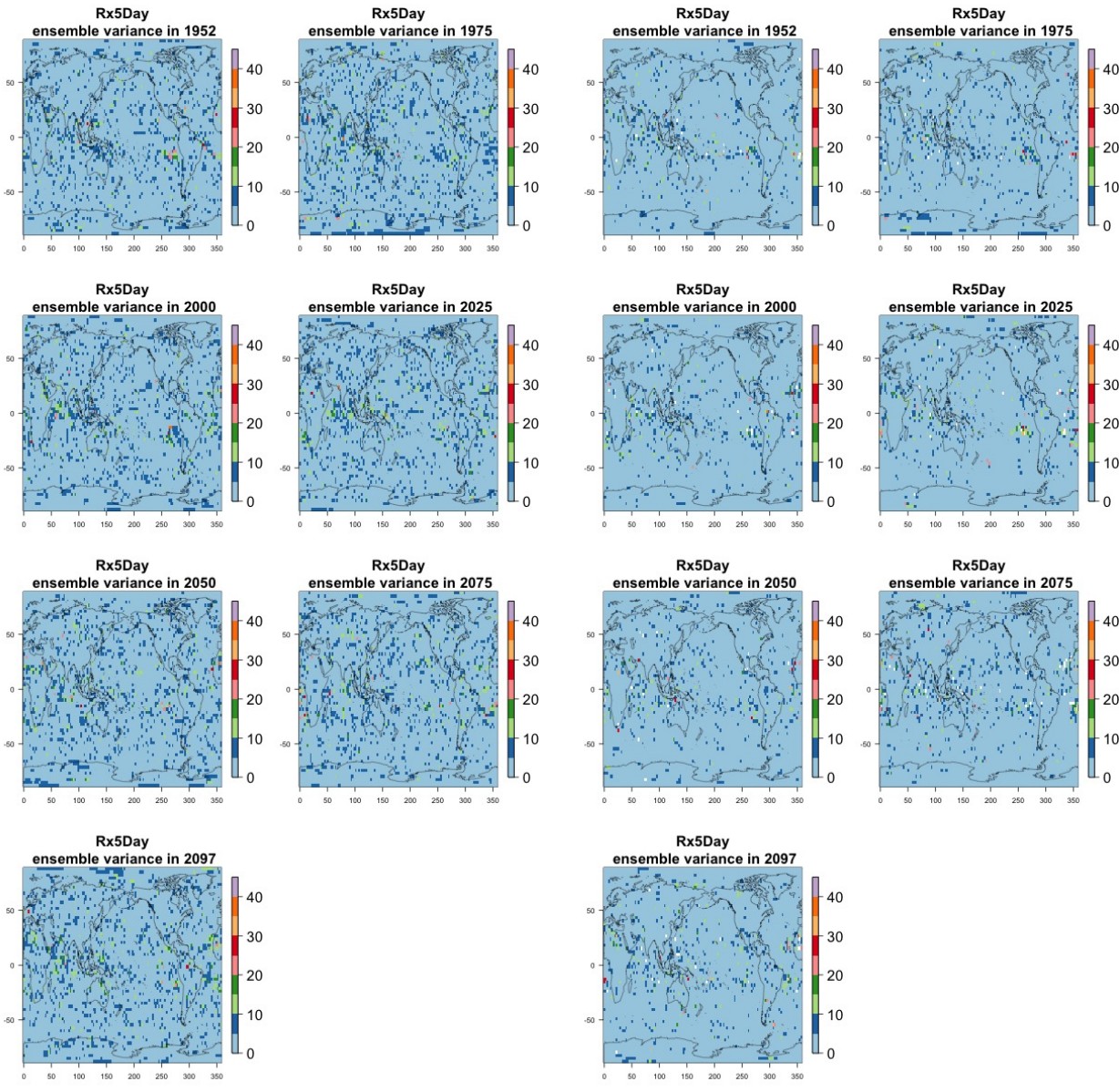

**Figure C29.** Estimating the ensemble variance for Rx5Day in the CanESM ensemble: Each plot corresponds to a year along the simulation length (1950, 1975, 2000, 2025, 2050, 2075, 2100). The color indicates the number of ensemble members needed to estimate a variance at that location that is statistically indistinguishable from that computed on the basis of the full 50-member ensemble. The results of the first two columns use only the specific year for each ensemble member. The results of the third and fourth columns enrich the samples by using 5 years around the specific date.

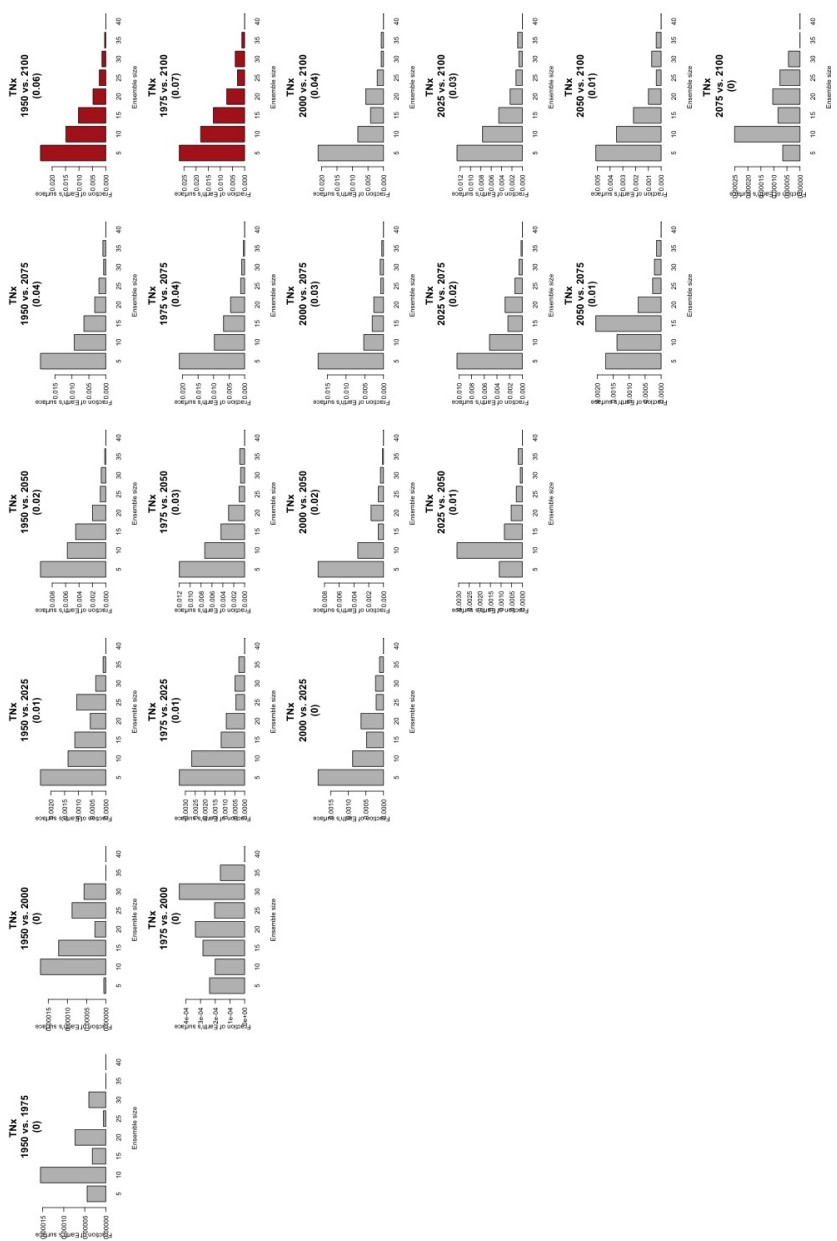

**Figure C30.** Histograms of required ensemble sizes at grid-boxes where significant change in variability is detected: each plot corresponds to a map in Figure 9 and shows a histogram of the values at each grid-box that is colored in that map. Histograms are weighted according to the cosine of the latitude of the grid-box, so that the values along the y-axes can be interpreted as fractions of the Earth's surface. We color red histograms that represent a total fraction of the Earth's surface larger than 5%.

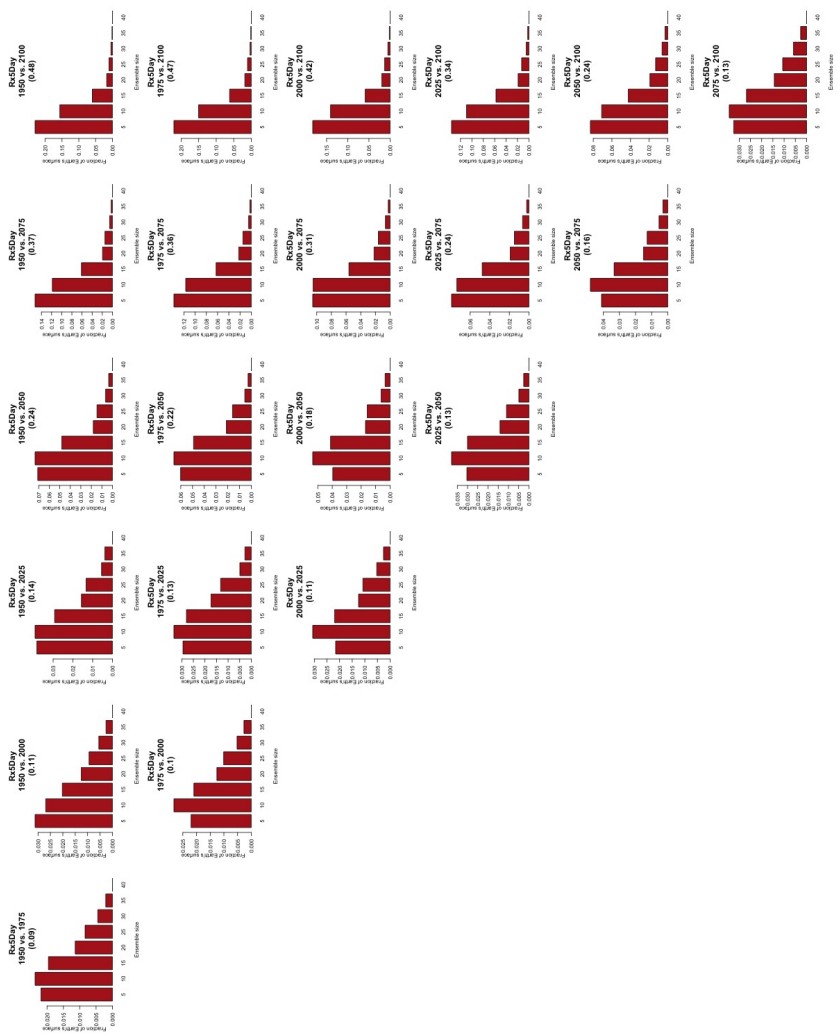

**Figure C31.** Histograms of required ensemble sizes at grid-boxes where significant change in variability is detected: each plot corresponds to a map in Figure 10 and shows a histogram of the values at each grid-box that is colored in that map. Histograms are weighted according to the cosine of the latitude of the grid-box, so that the values along the y-axes can be interpreted as fractions of the Earth's surface. We color red histograms that represent a total fraction of the Earth's surface larger than 5%.

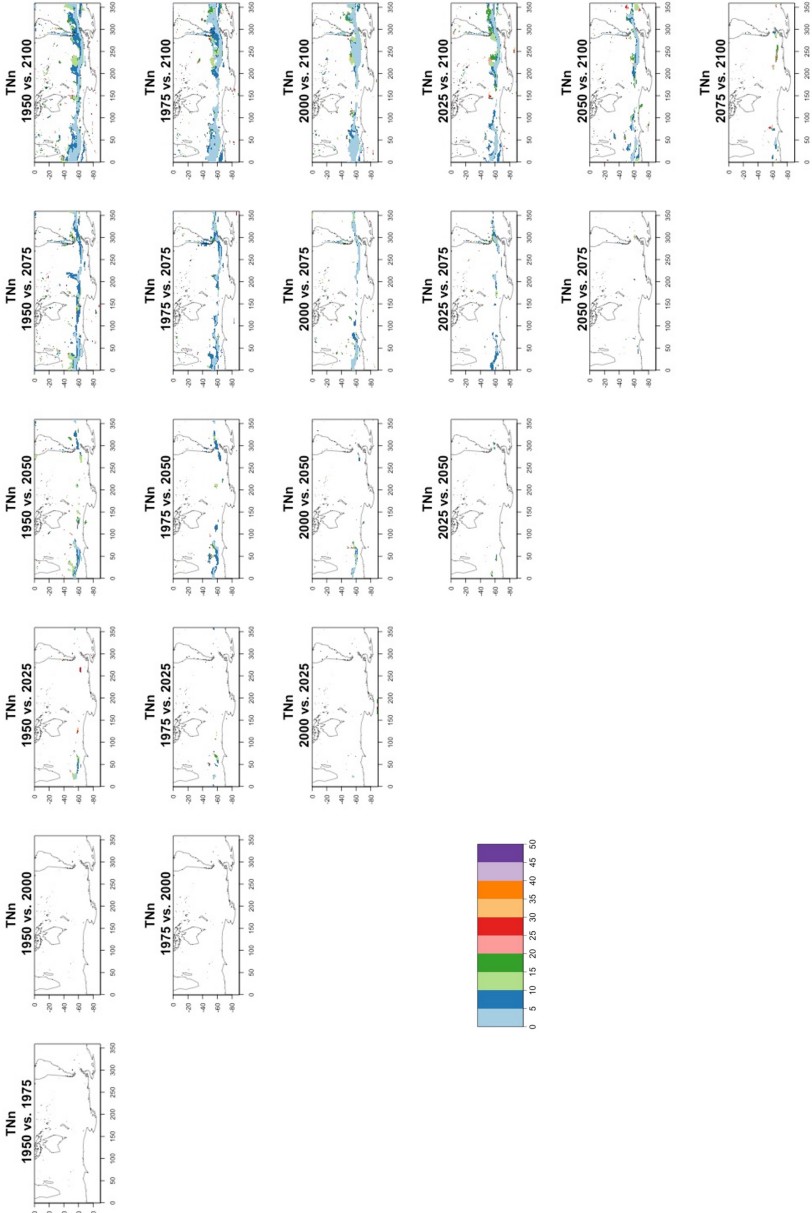

**Figure C32.** Estimating changes in ensemble variance for TNn in the CESM ensemble: each plot corresponds to a pair of years along the simulation. Colored areas are regions where on the basis of the full 40-member ensemble a significant change in variance was detected. The colors indicate the size of the smaller ensemble needed to detect the same change. Here too the sample size is increased by using 5 years around each date.

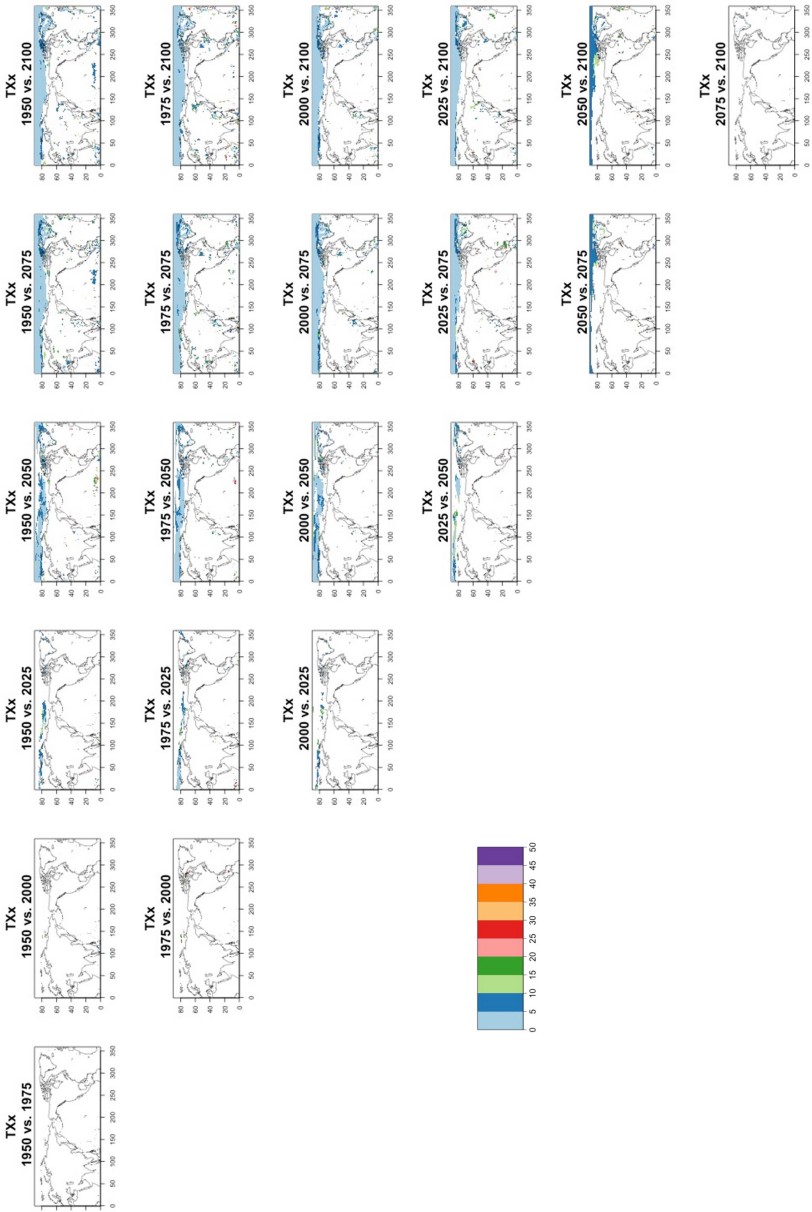

**Figure C33.** Estimating changes in ensemble variance for TXx in the CESM ensemble: each plot corresponds to a pair of years along the simulation. Colored areas are regions where on the basis of the full 40-member ensemble a significant change in variance was detected. The colors indicate the size of the smaller ensemble needed to detect the same change. Here too the sample size is increased by using 5 years around each date.

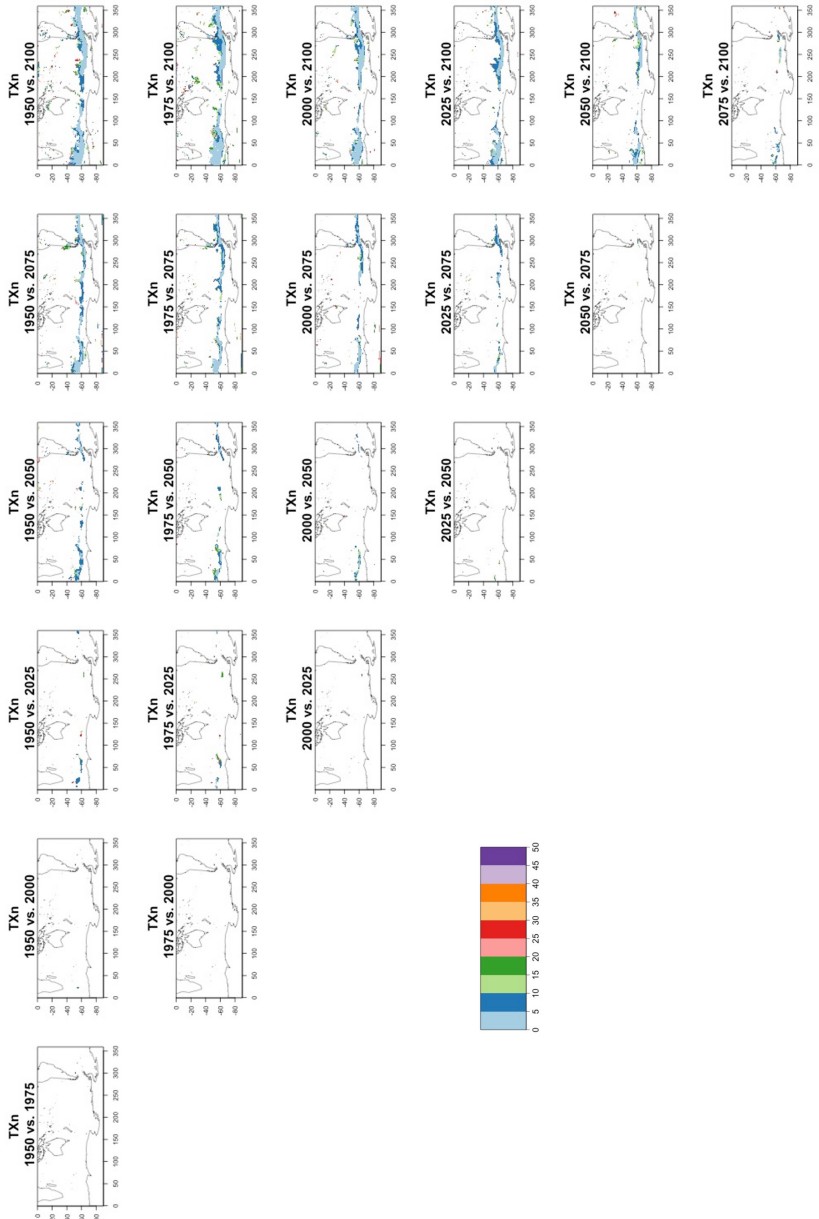

**Figure C34.** Estimating changes in ensemble variance for TXn in the CESM ensemble: each plot corresponds to a pair of years along the simulation. Colored areas are regions where on the basis of the full 40-member ensemble a significant change in variance was detected. The colors indicate the size of the smaller ensemble needed to detect the same change. Here too the sample size is increased by using 5 years around each date.

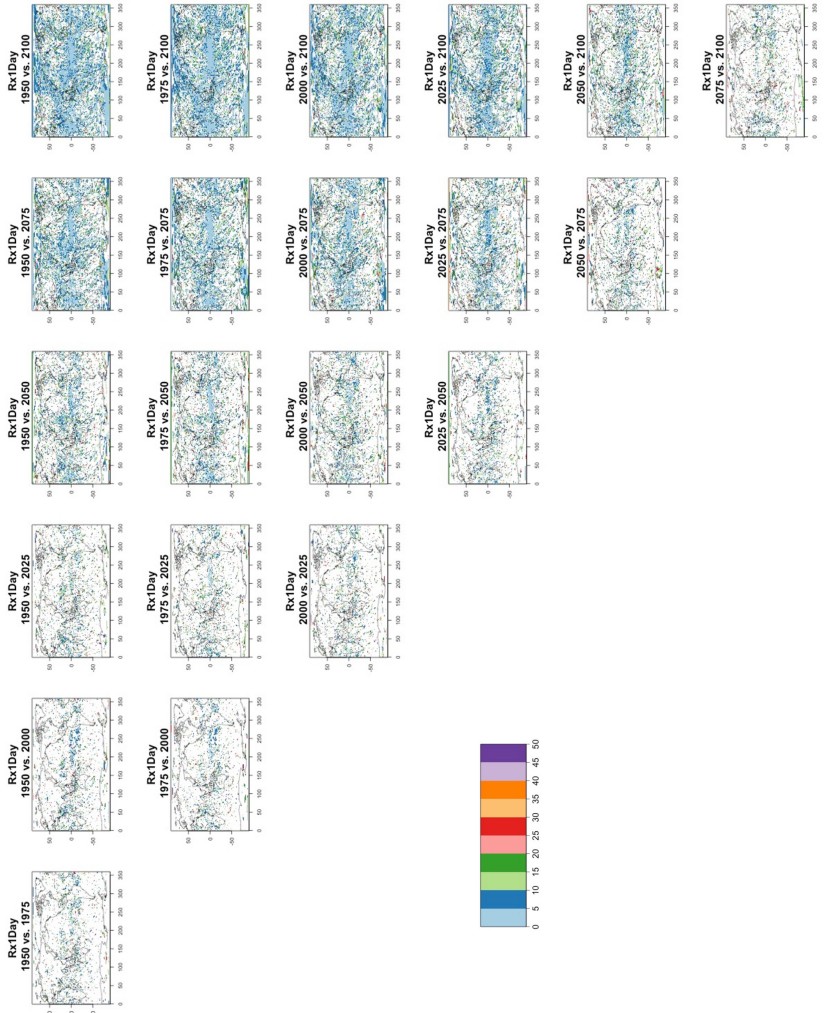

**Figure C35.** Estimating changes in ensemble variance for Rx1Day in the CESM ensemble: each plot corresponds to a pair of years along the simulation. Colored areas are regions where on the basis of the full 40-member ensemble a significant change in variance was detected. The colors indicate the size of the smaller ensemble needed to detect the same change. Here too the sample size is increased by using 5 years around each date.

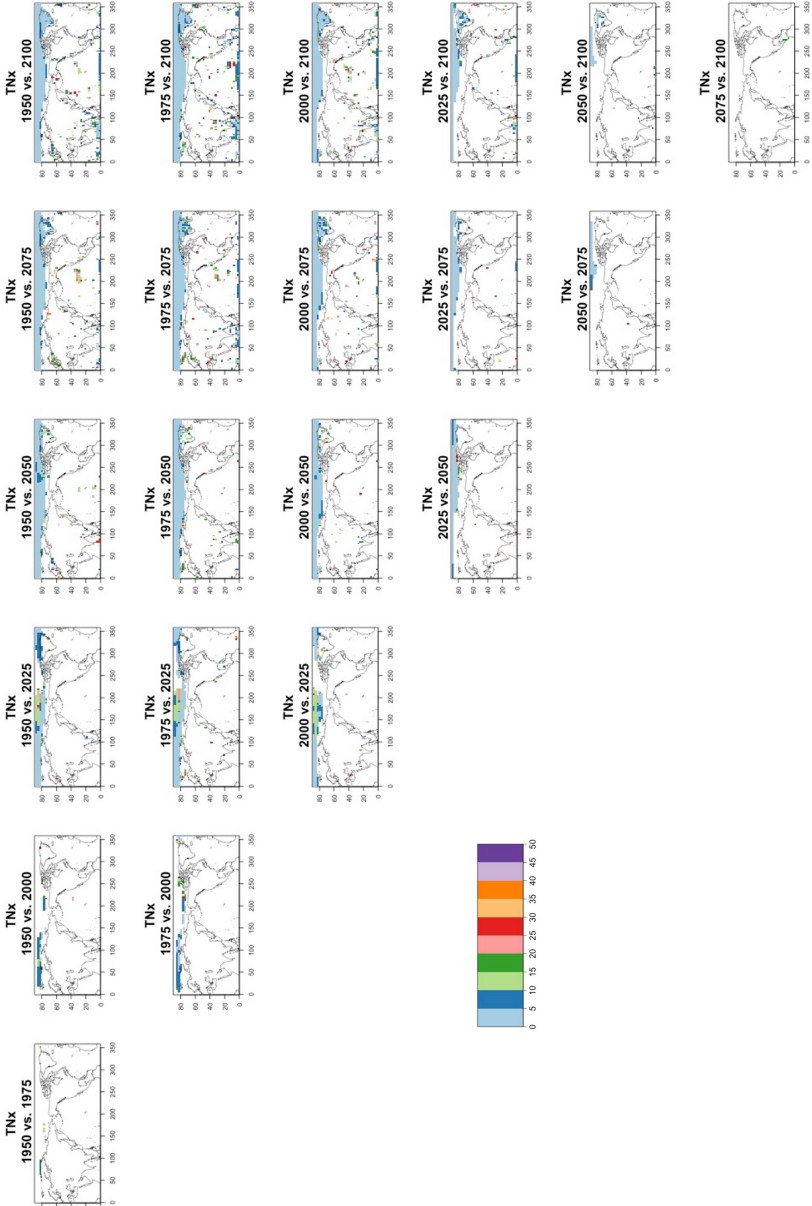

**Figure C36.** Estimating changes in ensemble variance for TNx in the CanESM ensemble: each plot corresponds to a pair of years along the simulation. Colored areas are regions where on the basis of the full 50-member ensemble a significant change in variance was detected. The colors indicate the size of the smaller ensemble needed to detect the same change. Here too the sample size is increased by using 5 years around each date.

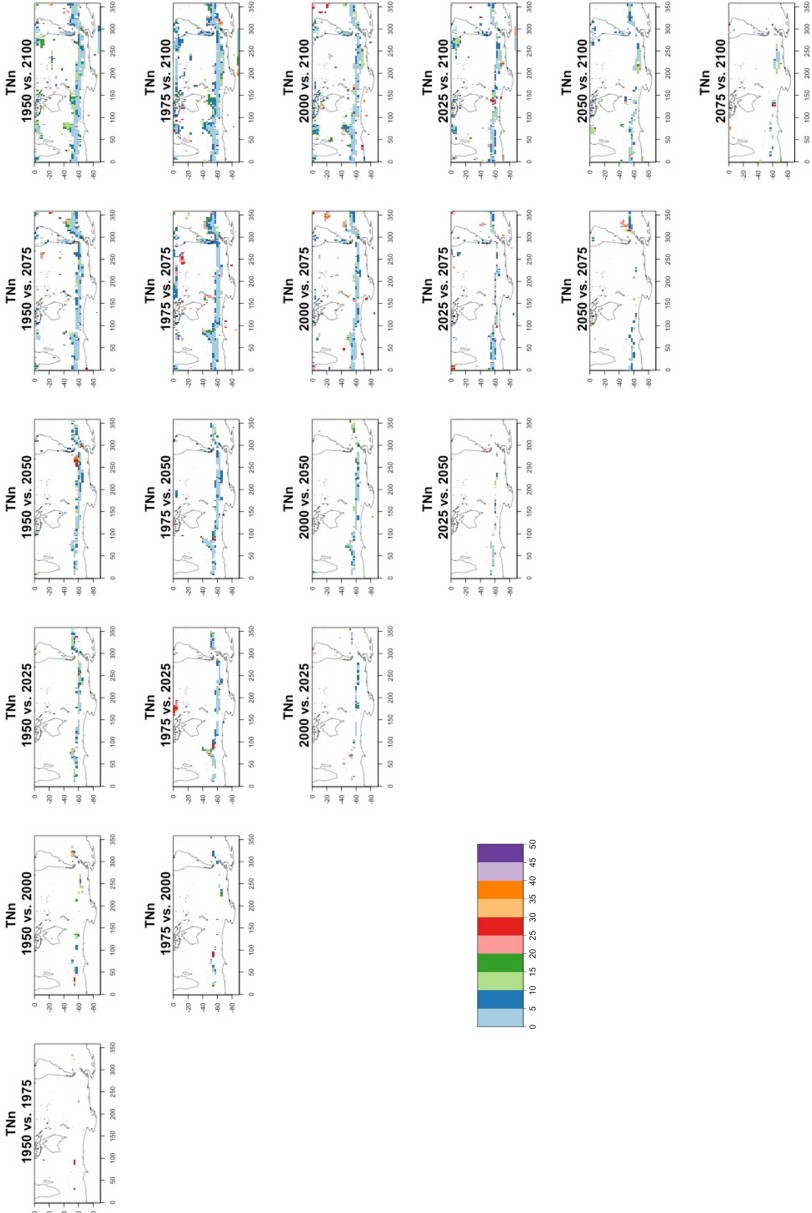

**Figure C37.** Estimating changes in ensemble variance for TNn in the CanESM ensemble: each plot corresponds to a pair of years along the simulation. Colored areas are regions where on the basis of the full 50-member ensemble a significant change in variance was detected. The colors indicate the size of the smaller ensemble needed to detect the same change. Here too the sample size is increased by using 5 years around each date.

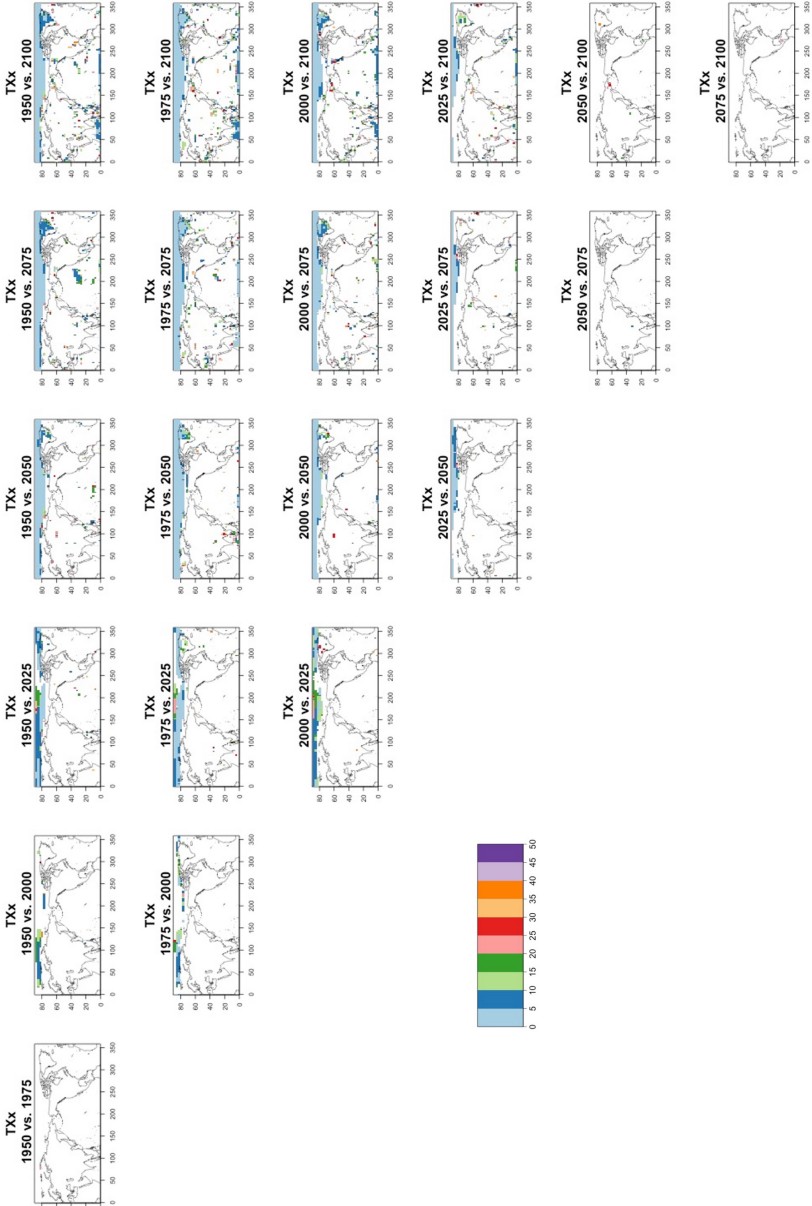

**Figure C38.** Estimating changes in ensemble variance for TXx in the CanESM ensemble: each plot corresponds to a pair of years along the simulation. Colored areas are regions where on the basis of the full 50-member ensemble a significant change in variance was detected. The colors indicate the size of the smaller ensemble needed to detect the same change. Here too the sample size is increased by using 5 years around each date.

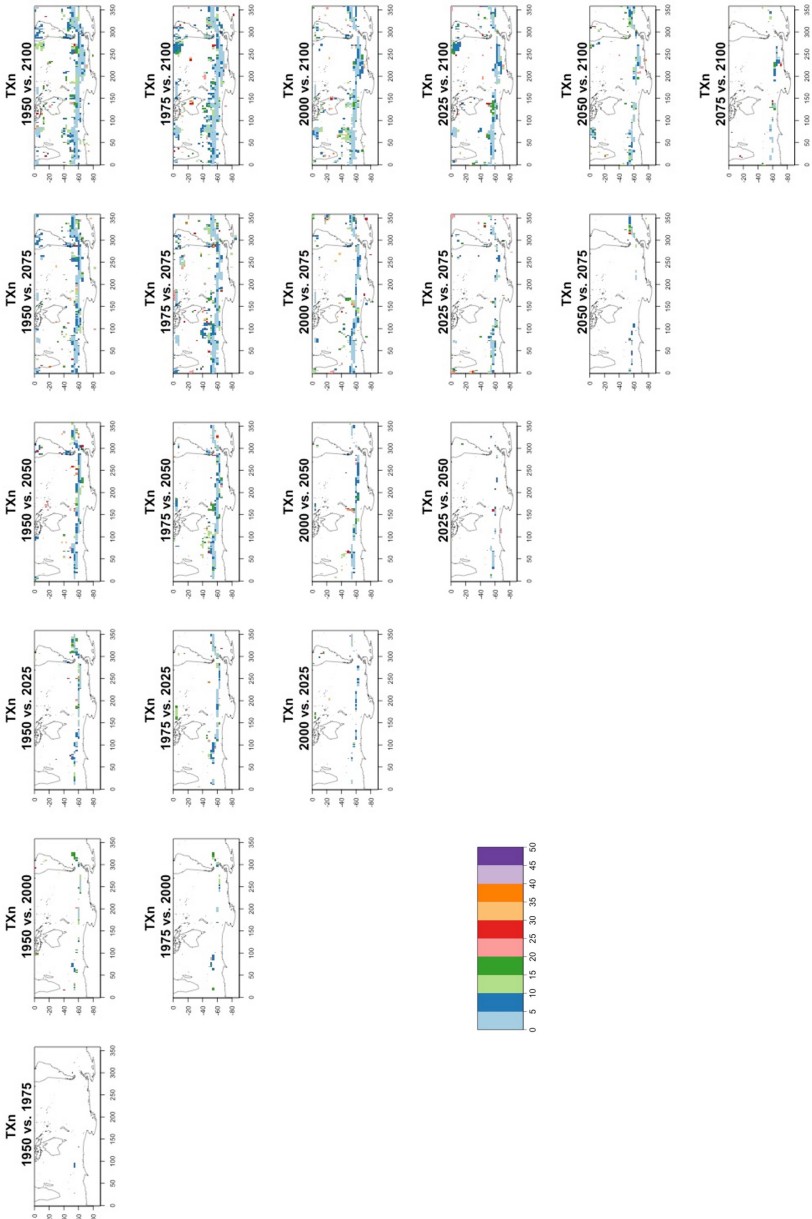

**Figure C39.** Estimating changes in ensemble variance for TXn in the CanESM ensemble: each plot corresponds to a pair of years along the simulation. Colored areas are regions where on the basis of the full 50-member ensemble a significant change in variance was detected. The colors indicate the size of the smaller ensemble needed to detect the same change. Here too the sample size is increased by using 5 years around each date.

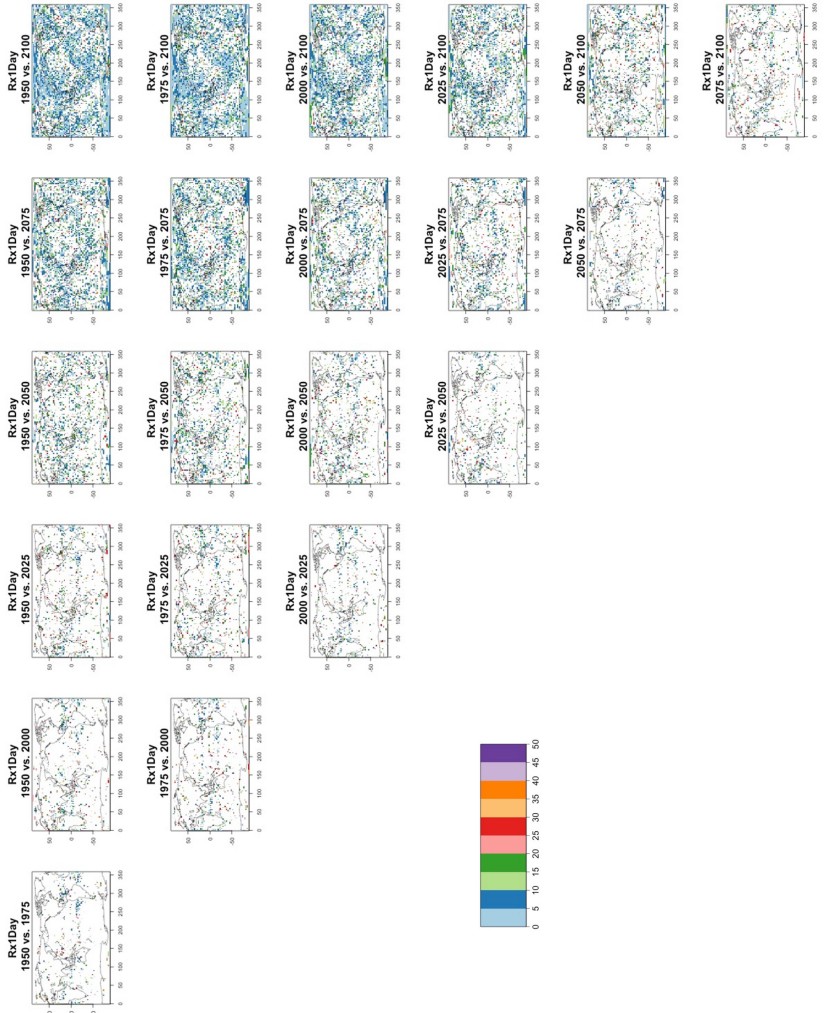

**Figure C40.** Estimating changes in ensemble variance for Rx1Day in the CanESM ensemble: each plot corresponds to a pair of years along the simulation. Colored areas are regions where on the basis of the full 50-member ensemble a significant change in variance was detected. The colors indicate the size of the smaller ensemble needed to detect the same change. Here too the sample size is increased by using 5 years around each date.

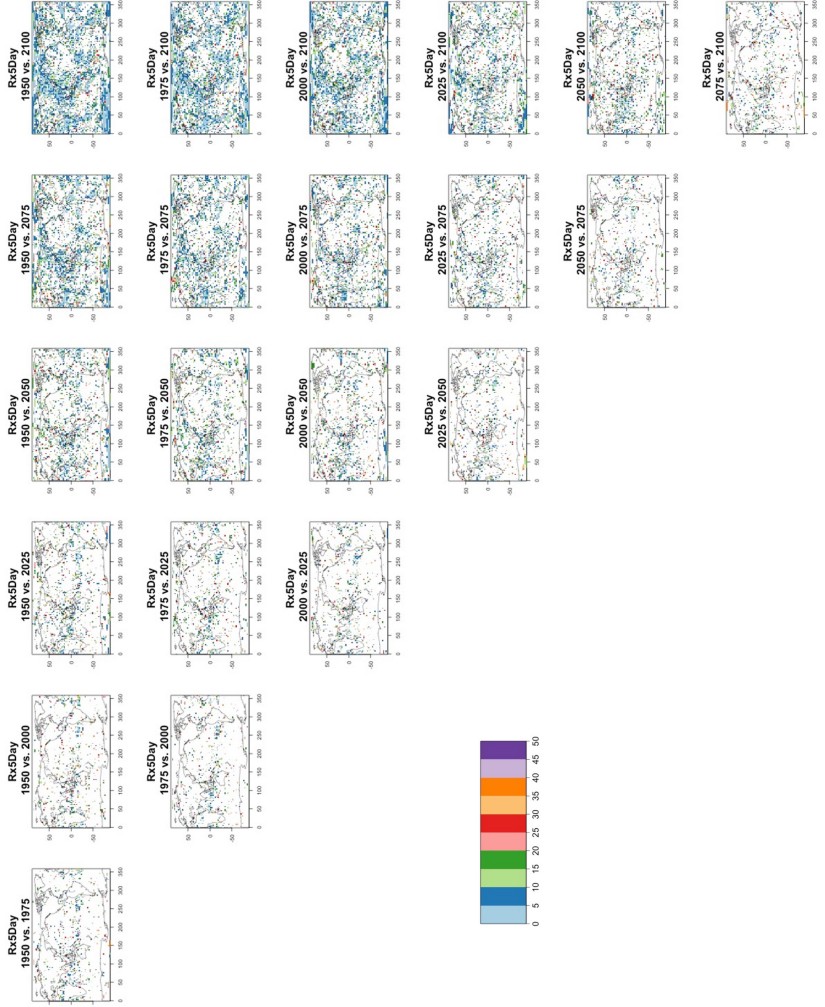

**Figure C41.** Estimating changes in ensemble variance for Rx5Day in the CanESM ensemble: each plot corresponds to a pair of years along the simulation. Colored areas are regions where on the basis of the full 50-member ensemble a significant change in variance was detected. The colors indicate the size of the smaller ensemble needed to detect the same change. Here too the sample size is increased by using 5 years around each date.

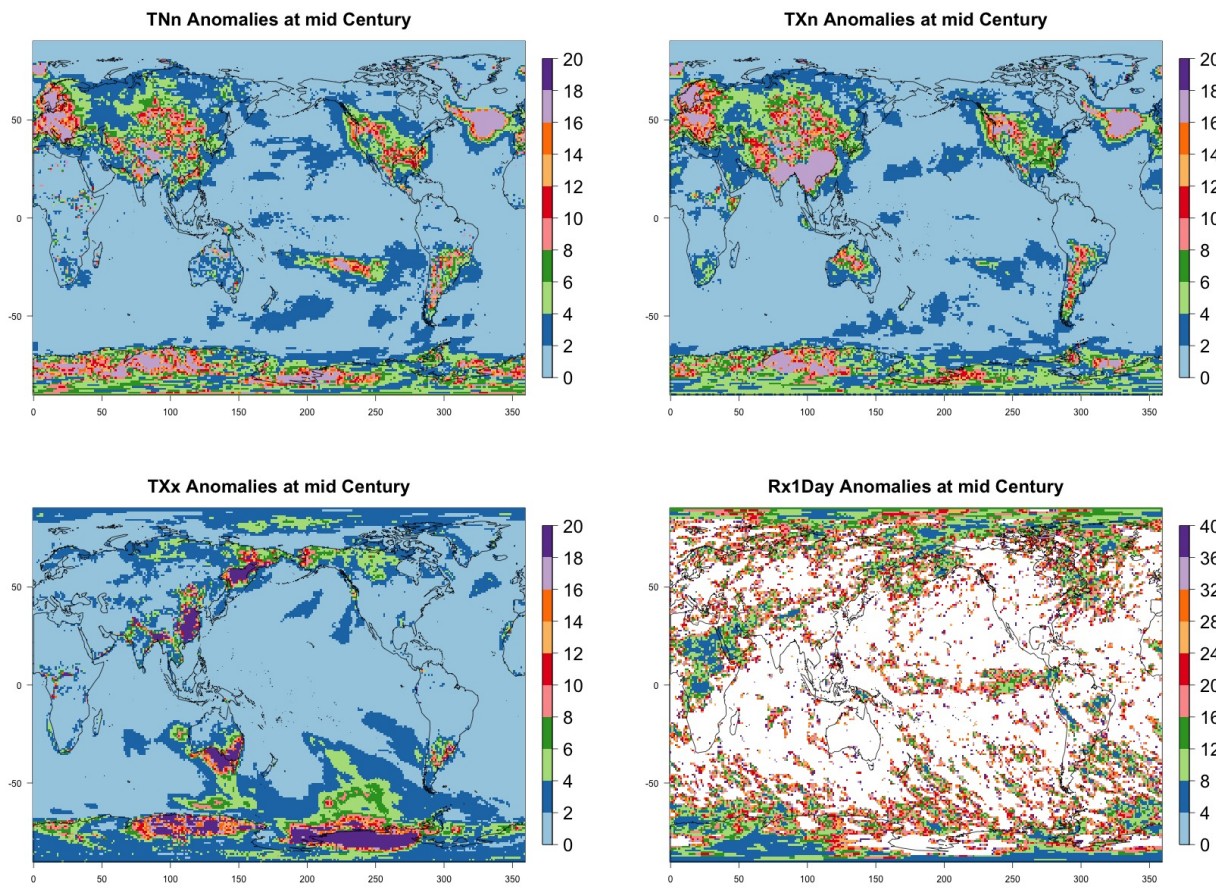

**Figure C42.** Ensemble size $n$ required for the signal to noise ratio of the grid-point scale anomalies to exceed 2 (anomalies defined as the mean of 2048-2052 minus the historical baseline of 2000-2005). Results for CESM, all remaining metrics not shown in the main text: Coldest night (TNn) and coldest day (TXn) of the year along the top row; Hottest day (TXx) and wettest day (Rx1Day) of the year along the bottom row.

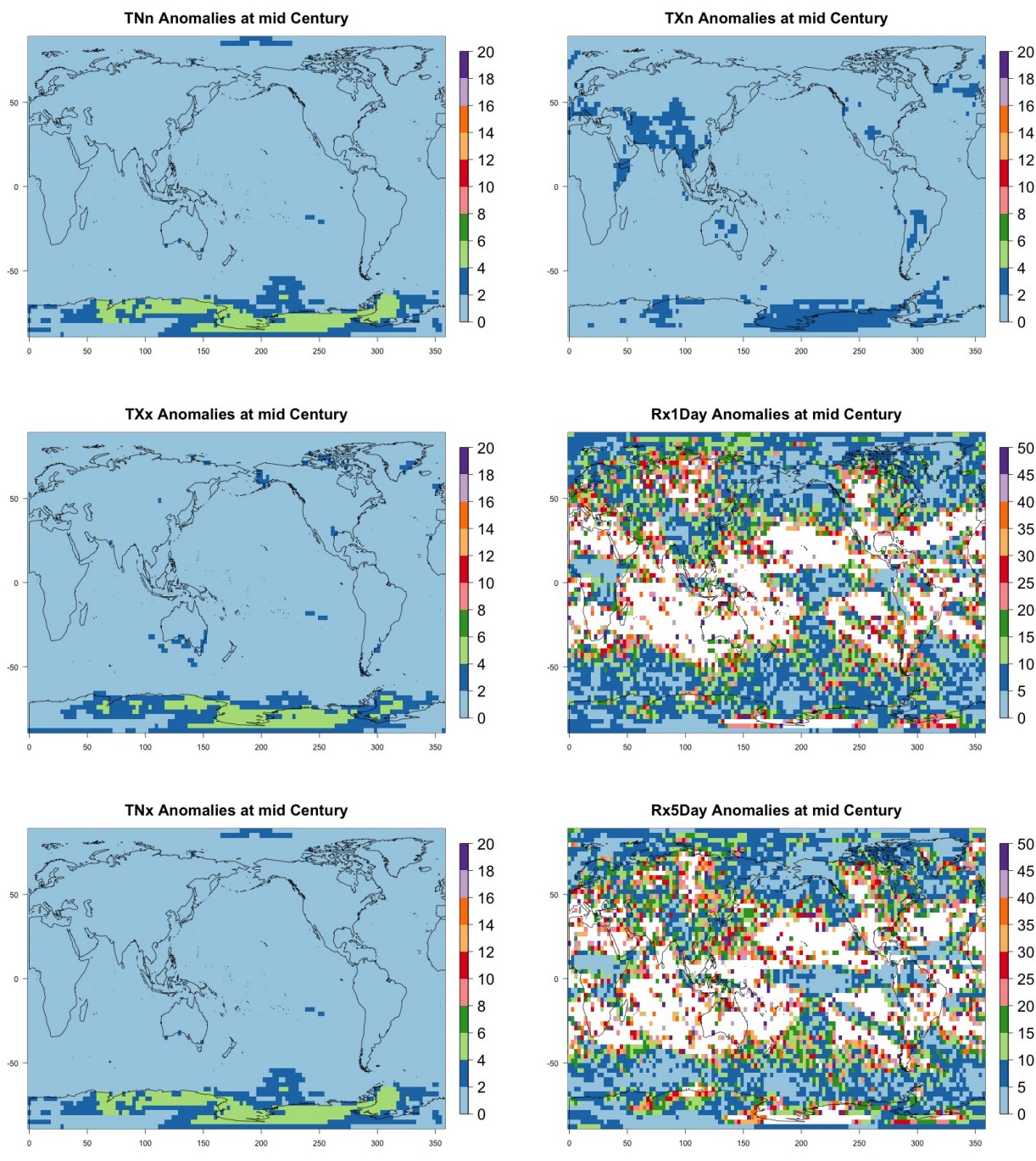

**Figure C43.** Ensemble size $n$ required for the signal to noise ratio of the grid-point scale anomalies to exceed 2 (anomalies defined as the mean of 2048-2052 minus the historical baseline of 2000-2005). Results for CanESM and all metrics: coldest night (TNn) and coldest day (TXn) of the year on the top row; hottest day (TXx) and wettest day (Rx1Day) of the year on the middle row; hottest night (TNx) and wettest 5-day (Rx5Day) of the year on the bottom row.

*Author contributions.* CT conceived the study, analyzed the data and wrote the paper. KD and MW provided data pre-processing and co-wrote the paper. RL advised and co-wrote the paper.

*Competing interests.* The authors declare no competing interests.

*Acknowledgements.* This study was supported by the Energy Exascale Earth System Model (E3SM) project funded by U.S. Department of Energy, Office of Science, Office of Biological and Environmental Research. MW was supported by the CASCADE project also funded by U.S. Department of Energy, Office of Science, Office of Biological and Environmental Research. The Pacific Northwest National Laboratory is operated by Battelle for the US Department of Energy under Contract DE-AC05-76RLO1830. Lawrence Berkeley National Laboratory is operated by U.S. Department of Energy under Contract No. DE340AC02-05CH11231.

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
