# Peer review of "Extreme Metrics from Large Ensembles: Investigating the Effects of Ensemble Size on their Estimates"

_Earth System Dynamics, 2021_

## Author Comment (AC1)

We thank both referees for a careful reading of the paper and many suggestions that have improved it substantially.

In the following we provide responses to all points raised by Referee 1.

Please also note that the paper has been edited throughout for clarity and rigor, and we will provide a version with tracked changes.

In the following, we highlight our responses in blue, with italics used when we cite word by word from the paper (apologies for the occasional latex syntax left throughout these verbatim sections).

Thank you,

Claudia Tebaldi and co-authors.

**Referee 1**

The authors address the problem of choosing the size of a climate model ensemble by investigating the number of members needed to obtain robust statistics for the warmest night (TNx) and wettest pentad (Rx5d) of the year. The main text presents results based on the CESM1-CAM5 38/40 member ensemble (CanESM2 48/50 member results are shown in the supplement). They investigate the accuracy, compared to the full ensemble, of variety of statistics obtained based on 1,5,10,...,35 ensemble members, namely: 1) the forced component (ensemble mean) and its uncertainty, 2) generalized extreme value (GEV) distribution fits, and 3) the internal variability (ensemble variance) at various time points, as well as detecting changes over time. They also briefly touch on the number of members needed to detect a signal to noise ratio greater than 1 for TNx and Rx5d by mid and end century. They argue that often, an ensemble of 20-25 members provides reliable statistical estimates. They also show that the full ensemble standard deviation (and error estimates that rely on it) can be accurately estimated using 5 ensemble members, supplemented by using 5 years along the time dimension of each member.

The study is of practical interest for optimizing computing resources and for potentially allowing broader investigations into structural/parameter uncertainty. While the general question of determining ensemble size has been addressed in previous studies, the analysis using yearly extremes is a new contribution as far as I'm aware. However, there are major issues that need to be addressed before the manuscript could be considered suitable to publish.

Thank you for the overall positive assessment.

Major comments:

1. There are several places where details of the methods are not provided (see specific comments below), particularly in Section 4.3 where the statistical tests used to determine the accuracy of the ensemble estimates of variability and for detecting the change in variance over time are not discussed. As such, I cannot yet comment on the validity of the methods for this section.

   We apologize for this specific oversight, and the more general lack of details, which we have now attempted to fix throughout. This will be visible in the track-changes document we provide, but please also see answers to specific comments for details.

2. The bootstrap estimates are all dubious for n > 20 due to oversampling. This is mentioned in the manuscript and was addressed in some detail in Milinski et al., 2020. They are therefore not suitable for comparison in the Tables, and misleading in Figures. I strongly suggest removing any bootstrap results for n>20. Alternatively, you can visually highlight the dubious numbers in the Tables and provide a caveat in the captions.

As a side note, I also have some doubts about number 20-25 being sufficient to provide reliable estimates, as that number constitutes a majority of the ensemble members and therefore may be biased low. I cannot point to any theory to confirm or deny my suspicion however. Repeating the analysis with an ensemble that has more members would increase the confidence in them, perhaps this can be done in another study.

Thank you for this comment and specific suggestion. We have now made clearer that the main approach in this paper is relying on the formula estimating the sample size n to bound the expected standard error, sigma/sqrt(n), where sigma, we demonstrate, can be effectively estimated by using only 5 members. We presented the first results also using the bootstrap method with the aim to quantify the method's shortcoming when the size of the ensemble "closes in" towards the full size of the ensemble, and justify our alternative approach. That was really the only reason to use the bootstrap in the analysis of the global time series, providing continuity with what we saw as the most natural predecessor paper, Milinski et al., and substantiating their argument further with specific numbers. We have tried to make all this clearer. We have tried to follow the reviewer's suggestions to include caveats for this and other issues throughout, starting from the caption and design of the tables. For the latter we now color in grey all the cells where the estimates of the bootstrap are not consistent with the confidence intervals derived from the full ensemble.

3) When supplementing the estimate using 5 ensemble members with information along the time dimension, one runs into the problem of serial correlation, which can reducing the effective sample size. So 5 members x 5 years results is a sample size of

25 at most. Depending on the variable of interest it can be substantially less. I am fairly confident that this is the reason that more ensembles are needed to estimate TNx variance over the ocean for example (Fig. 3). A caveat regarding this issue should be included in the text.

We have found that at a year-to-year scale, for the global quantities considered the presence of autocorrelation is not significant and therefore does not affect the validity of our estimates enough to make them faulty. We agree with the reviewer that that autocorrelation likely affects regions of the ocean and perhaps explain some of the areas that show large errors in Figure 3 and 4. However, the analysis of the frequency of these exceedances over space still supports the general approach by showing that the 2*sigma/sqrt(n) limit, with sigma estimated by 5x5 observations (5 runs, 5 years each run), works accurately. We have nonetheless included specific mentions of the problem with autocorrelation, and caveats in the text when needed.

For example, while discussing the tables of RMSE estimates and mentioning the use of the 5-year windows,

*"We are aware that this could introduce auto-correlation within the sample values, but the comparison of these results to the truth shows that the estimated values based on the smaller ensemble are an accurate approximation of it, always being consistent with the 95\% confidence intervals (shown in parentheses)."*

Or when describing the barplots of actual errors against the 95% bound:

*"Here is where our approximation, and the use of possibly autocorrelated samples in the estimates of $\sigma$ could possibly reveal shortcomings."*

We added the specific point from the reviewer when describing patterns of error:

*"The prevalence of red areas over the oceans could be due to an underestimation of $\sigma_i^c$ linked to the use of the 5-year windows and the autocorrelation possibly introduced, consistently with ocean quantities having more memory than land quantities, but we do not explore that further here."*

Specific comments

**All the figures are too small.**

We have redrawn all figures improving their readability and correcting some color legend inaccuracies throughout. We also added several figures to the appendix, documenting more extensively the additional metrics and the alternative model.

**Line 43**: Surface temperature is one of the strongest climate signals generally and with regard to extreme metrics. So an ensemble number based on analyzing TS is functionally a minimum value. Precipitation will also be a strong signal, especially averaging over larger regions where the dynamic changes might cancel out, leaving only the increase due to thermodynamics. So changes in precipitation extremes are also not difficult to detect. You later address this by looking a specific times when the changes are small, which I think is worth mentioning here.

Thank you for highlighting this point about the temporal range considered when identifying change, to which we would also add the fact that we are looking at statistics of precipitation extremes all the way to the grid-point scale. So we modified the sentence as:

*"The consideration of two models, two atmospheric quantities and several extreme metrics, each analyzed at a range of spatial and temporal scales help our conclusions to be robust and -- we hope -- applicable beyond the specifics of our study "*

**Line 65-66:** This is an overly strong statement. Model variability on increasingly long time scales (and hence increasing ocean depths) is indeed not included by design. But how do you justify saying these GCMs do not represent ocean variability?

This is a misunderstanding. It was not our intention to claim that GCMs do not represent ocean variability, and we have rephrased this sentence to make clear that with "experiments" we meant the ensemble design, not the ESM experiment quality per se. The sentence now reads:

*"We note that sources of variability from different ocean states, particularly at depth, cannot be effectively explored by this type of initial condition ensemble design, which only perturbs atmospheric conditions."*

**Line 68, elsewhere:** Generally, you should avoid contractions in scientific papers

Thank you, we have corrected them throughout.

**Line 93:** Some brief info on the location, shape, and scale parameters would be helpful. If nothing else just name them here, which would help the reader's intuition.

Also in response to a similar request by the other reviewer, we have added the following after listing the three parameters' domains:

*"[...] represent the location, scale and shape parameter respectively, responsible for the mean, variability and tail behavior of the random quantity z."*

**Line 124:** A symmetric sample around 1953, 2097 would result in 7 years, not 11.

We corrected those years to read 1957 and 2094 respectively (this was a mix-up with another series of window centers that we use in the calculation of the sigma parameter).

**Fig 1: The full ensemble line looks white, not blue as in the legend. Only plot up to n=25, to avoid issues with oversampling.**

We have corrected these plots' color legends and redrawn the full ensemble estimate in black. Now that we think we have made clearer the shortcomings of the bootstrap and our motivation for using it we thought we could leave the whole range of estimates in the plots.

**Line 168:** Since the full ensemble statistics are being treated as the true statistics, the F values are the true RMSE values. So, the results show that the bootstrap method is not "optimistic" but inaccurate, as mentioned in line 109 and Milinski, 2020.

Agreed, and we reworded using "inaccurate".

**Line 167-168:** Need to consider autocorrelation. Are the F-5 values bootstrapped, or is only a single 5-member subset used?

We use only the first 5 members of the ensemble, again adopting the point of view of a modeling center that only has 5 runs to estimate sigma and drawing the comparison to the estimates computed on the basis of the full ensemble available. We have specified now "*on the basis of a small ensemble (the first 5 members)*" for clarity. We have discussed the problem of autocorrelation as detailed in an earlier response to the general comment about this.

**Table 1:** Show percentages for more precision. Remove or visual highlight the spurious bootstrap numbers. Comparing the n=5 results to the bootstrap result that are known to be spurious is not useful. Consider putting the "true" full ensemble number on the left.

We are keen to show actual values to also bring home the accuracy of the estimates in the unit of the quantity considered. We are also giving the general rule for the percentage change in RMSE according to ensemble size in a paragraph of the text:

*"Thus, compared to a single model run's RMSE, we expect the RMSE of mean estimates derived by ensemble sizes of $n=5, 10, 20, 35$ or $45$ to be 45%, 32%, 22%, 17% or 15% of that, respectively."*

However, we have improved the tables by switching the order of the columns as the Reviewer suggests, and by coloring in grey the cells of the tables corresponding to the inaccurate bootstrap estimates.

**Lines 180-184:** It seems the more general result that 5 ensemble members are sufficient to obtain an accurate measure the true instantaneous (i.e. changing over time) model internal variability at global scales.

Agreed, and we have changed the lessons learned exposition in the following way:

*The lessons learned here are that*

> *1 For both metrics, an accurate estimate of $\sigma_t$, i.e., the instantaneous model internal variability at global scale, is possible using 5 ensemble members (and a window of 5 years around the year t of interest);*
>
> *2 if the formula for computing the RMSE on the basis of a given sample size is adopted, and that estimate for $\sigma_t$ is plugged in, it is possible, on the basis of an existing 5-member ensemble, to accurately estimate the required ensemble size to identify the forced component within a given tolerance for error. Of course, the size of this tolerance will change depending on the specific application.*

**Line 186:** It's not clear what you mean by "RMSE affecting estimates" or "our estimate of σ"; What about starting with: "Thus, compared to a single model run, we would expect …."

Thank you for the suggestion, we have rephrased as

*"Thus, compared to a single model run's RMSE, we expect the RMSE of mean estimates derived by ensemble sizes of n=5, 10, 20, 35 or 45 to be 45%, 32%, 22%, 17% or 15% of that, respectively."*

**Line 191:** Why twice the expected RMSE? The 2 sigma level is roughly the 95% confidence interval of the expected error, so you would expect to exceed that level 5% of the time. You cannot conclude from these results that "the actual error is in most cases much smaller than the expected…"

You made us realize that our wording was imprecise all through this exposition, and we apologize for that. We now consistently refer to the 2sigma/sqrt(n) as the 95% bound

for the error, not as the expected error and we describe more rigorously the expectation of the actual error behavior compared to this bound throughout.

"*Figure 2 for global averages of the two same quantities, shows the ratios of actual error vs. the 95% probability bound, indicating the 100% level by a horizontal line for reference. As can be assessed, the actual error is in most cases much smaller than the 95% bound (as it is not reaching the 100% line in the great majority of cases), and we see that only occasionally the actual error spikes above the 95% bound for individual years, consistent with what would be expected of a normally distributed error. This behavior is consistently true for ensemble sizes larger than n=5* "

**Line 205:** "We use the full ensemble or only 5 members to estimate the ensemble standard deviation…" You specify two definitions here and then never clarify which one you use.

We have eliminated the mention of "the full ensemble" which was a residue of doing things step by step to compare results. In the results shown the estimate is based on 5 members * 5 years.

**Figure 2:** Why is 2-sigma the cutoff when the standard error is sigma/sqrt(n)?

Again, we believe this question comes from our less than rigorous description of the 95% bound.

The caption now (partially) reads:

"*In each plot, for each year, the height of the bar gives the error in the estimate of the forced component (defined as the mean of the entire ensemble) as a percentage of the expected 95% probability bound, estimated by the formula $2\sigma_t/\sqrt{n}$ with n the ensemble size*."

The diagonal lines are spurious; they show a large widening of the variability that is absent in Fig 1 (no change in the width of the lines in a-d.) Table 1 also indicates that TNx variability increase by maybe 50%, not a factor of 3 as shown here. Also the variability may not change monotonically with time.

This comment was likely a product of the axis labels being illegible (which we corrected throughout). The change is – as the caption says – stylized, and for TNx the right axis labels show a change over time consistent with the reviewer's estimate of about 50%. The larger change is for Rx5Day which is to be expected in precipitation-based metrics since precipitation variability follows precipitation mean increase with warming. We

decided however to eliminate this piece of information from the plots given that also Reviewer 2 found these diagonal lines confusing.

As for the reconciliation of Figure 1 with the behavior of variability over time, we note that in fact the width of the red envelope appears appreciably greater at the end of the period than at its beginning for Rx5Day (panel (b)), while the x-y aspect ratio for panel (a) makes it difficult to appreciate a relatively smaller change. As for panels (c) and (d), please note that these have a limited time range, covering only the historical period, while the change appears most pronounced when comparing the latest years of the 21$^{st}$ century to the earliest period in the historical years.

**Line 210:** "Small and sparse" over land perhaps, but certainly not over the ocean due to the longer timescales of variability which necessitate more samples.

We have added this point, specifying that these indices have been usually adopted in impact analysis over land.

*"As can be gauged even by eye, only small and sparse areas appear where the actual error exceeds the expected error, especially if land regions are considered (incidentally, these indices have been mostly used over land areas, as input to impact analyses). The prevalence of red areas over the oceans could be due to an underestimation of $\sigma_i^c$ linked to the use of the 5-year windows and the autocorrelation possibly introduced, consistently with ocean quantities having more memory than land quantities, but we do not explore that further here. Over the majority of the Earth's surface, particularly when errors are estimated for ensemble sizes of 20 or more, the bound is a good measure providing an accurate estimate of the error behavior according to normal distribution theory."*

We specifically note here that when we compute the percentage of points affected by errors above the 2*sigma/sqrt(n) bound, it is still consistent with the 5% expectation (as the tables in appendix describe).

**Line 212:** "consistently providing a conservative estimate of it according to normal distribution theory": 1) I don't think that conservative is the goal, but accuracy, and 2) this is dependent on using the 2-sigma threshold, which was not justified.

**Fig 3:** The 2 sigma level is not an "upper bound", but an estimate of the 95% level.

For both these comments we believe that our improved discussion of the use of 2*sigma/sqrt(n) as the 95% bound will help. The sentence that used to be on line 212 now reads:

*"Over the majority of the Earth's surface, particularly when errors are estimated for ensemble sizes of 20 or more, the bound is a good measure providing an accurate estimate of the error behavior according to normal distribution theory."*

We have also corrected the caption of Figure 2, always referring to 2*sigma/sqrt(n) as the 95% confidence bound.

**Fig 3&4:** From Fig. 1, it seems safe to assume a normal distribution for the ensemble range of the global average of the block maxima (TNx, Rx5Day). But is that true at each grid point. Don't we expect them to be follow the GEV distribution?

These plots show averages (across ensemble members, except for the first panels in row 1 and 3, which show 1 ensemble member only) of the difference between two 5-year means. We agree that the individual quantities (for a given year and model run at the grid-point level) follow the GEV, but for these average quantities we invoke the central limit theorem, and our calculations of the exceedances seem to confirm that the behavior is indeed consistent with the normal approximation.

**Line 219:** The goal is to find the number of ensembles, n, that can be used to accurately "decide how large an ensemble we need in order to approximate the forced component to a given degree of accuracy." As I read it, that amounts to finding the a threshold for n, such that the ensemble standard deviation sigma_n accurate measure the true sigma. So rather that comparing to a threshold of 2 sigma (the rationale for which is never given) you should compare the error with respect to the true sigma, including the confidence intervals of both the true value and the estimate. The two aspects being how accurate is the estimate, and how narrow is the range of estimate.

We have given our analysis a slightly different angle, as we have shown that *sigma estimated on the basis of 5 ensemble members* (which we consider a standard ensemble size traditionally run by modeling centers) produces results (in terms of estimating the RMSE around the mean signal within a given tolerance as a function of n) that are not significantly different from what the full ensemble size would produce.

When we show a "bound" defined as 2sigma/sqrt(n) in our various analyses it is because we want to compare actual realizations of the error, and therefore we use the fact that 2*sigma/sqrt(n) represents the size of the deviation from the mean that we expect to be exceeded 5% of the times.

What the reviewer argues could be considered another way to look at the issue where the "5 member ensemble" is not a given and finding the size to estimate sigma accurately is the first order of business. That however would not be enough to ensure that the mean component estimate would be within a desired tolerance for error, as

that error is a function of sigma/sqrt(n*) where n* may be different from the n found to be accurate in estimating sigma.

We do approach the problem of accurately estimating sigma per-se in the sections dedicated to internal variability. We have now pointed out that the results in this section were in fact implicit in our finding 5-members sufficient to estimate sigma, and therefore the appropriate ensemble size for the estimate of the forced component, in the previous sections.

**Line 220:** You do not actual address regional means.

It is true, and apologies for this mention that we wrote thinking of including results that at the time of organizing the paper we could not show without making the paper an overly long, overly tedious, list of results. We have rephrased as in

*"This holds true across the range of spatial scales afforded by these models, from global means all the way to grid-point values."*

**Line 232:** Provide a bit more information or reference about temporal covariates for the reader. You can safely assume that the forced climate change signal is not causing temporal autocorrelation, but interannual to decadal internal variability can still induce serial correlation.

We have elaborated as in

*"Further statistical precision could be attained by relaxing the quasi-stationarity assumption and extending the analysis period to contain a longer window of years. Exchanging time for ensemble members however, when beyond a decade's worth, necessitates in most cases the inclusion of temporal covariates: for example, indicators of the phase and magnitude of major modes of variability known to affect the behavior of the atmospheric variables in question over multi-decadal scales. The inclusion of covariates of course adds another source of fitting uncertainty."*

**Fig 5:** Is this based on a bootstrap, or a single selection of N ensemble members? From the NNA plots, I assume the latter, because the N=10 return levels are way off. If that is the case, were the same ensemble members used in all the plots? How are the confidence intervals for the full ensemble (the upper and lower horizontal lines) calculated?

Your assumption is right, and in all cases the CI are computed using the default MLE approach to their estimation from the extRemes package, which applies to any number

of ensemble members, also the full ensemble. We have specified both aspects at the opening of the section describing the GEV results:

*"We estimate return levels (event sizes, $z_p$ in our notation) for a number of return periods, i.e., 2-,5-,10-,20-,50- and 100-years by concatenating the 11 year segments across the first $n$ ensemble members, with $n$ varying between 5 and the full ensemble size by 5 units. For each value of $n$ the same subset of members is used across all metrics, locations, times and return periods."*

**Line 243:** You're not bootstrapping here so oversampling is not an issue, but 20-25 is still a majority of the ensemble members, which may artificially improve the results. The exercise would need to be repeated with a larger ensemble to increase confidence in the result. A caveat along these lines should be included.

This comment caused us to reevaluate the way we describe this part of the results, after considering more closely the patterns emerging from the same type of exercise with the CanESM ensemble, which sports 50 members (please note additional figures included in the supplementary material). As a consequence we revised slightly our argument, by distinguishing what size of an ensemble ensures the central estimate to be within the "true" confidence interval, and separately, how the size relates to the behavior of the C.I. width.

We now write:

> *"We first observe that in the great majority of cases the central estimate settles within the "true" confidence interval as soon as the ensemble comprises 15 or 20 members. This is true for both model ensembles, i.e., both when the truth is identified through 40 and through 50 ensemble members, as the corresponding plots in the appendix confirm. Therefore, if all that concerns us is the central estimate, an ensemble of 20 members, from which we sample 11-yr windows to enrich the sample size, delivers an estimate of the "truth" within its confidence interval. When an estimate of the uncertainty is concerned, however, the truth remains, by definition, an unattainable target, as the size of the confidence intervals is always bound to decrease for larger sample sizes. The behavior of the confidence intervals for the return level estimates in the plots, however, suggests that there might be only marginal gains for ensemble sizes beyond 30, for both models. The value of this general result will benefit from an analysis of larger ensembles. In addition, the value of increasing the sample size should always be judged on the basis of the actual size of the 95\% confidence intervals in the units of the quantity of interest, and what that size means for managing risks associated with these extremes. This is an aspect that, however, goes beyond the scope of our work."*

**Line 271:** What test is used to determine if they are distinguishable?

We use an F-test in all cases, and determine the threshold for significance by applying the False Discovery Rate method. The lack of mention was an oversight that we have now fixed. We have also added that we fix the FDR at 5%, which we also failed to specify before.

**Line 283:** The Ventura et al, 2004 method addressing spatial covariability, not temporal, correct?

Correct. The Ventura et al method is shown not to be sensitive to spatial correlation, and that is what we are concerned here (besides the general problem of multiple testing.) We added this point to the text:

*"We note here that the patterns shown in some of these figures have indeed the characteristics of noise. To minimize that possibility we have applied a threshold for the significance of the p-values from the F-test obtained through the method that controls the False Discovery Rate (Ventura et al,2004). The method has been shown to control for the false identification of significant differences "by chance" due to repeating statistical tests hundreds or thousands of times, as in our situation. The same method has been proved effective, in particular for multiple testing over spatial fields, despite the presence of spatial correlation (Ventura et al., 2004; Wilks, 206). We fix the false discovery rate to 5%."*

**Line 286:** What test is used to detect variance changes?

As before, we are still comparing variances  (in this case, estimates at different times during the simulation) by using the F-test. We have made that explicit here as well.

**Fig 9,10:** Several color bars are mislabelled

We have redrawn and corrected the layout of the plots not to cut off the legends.

**[TO DO]**

**Line 294:** "minimum temperature" might be confusing; consider using "warmest night" as you do elsewhere

We rephrased as in *"In the case TNx, a metric based on daily minimum temperature behavior,"* as we wanted to keep the reference to the basic atmospheric quantity on which the metric is based.

**Line 315:** Is sigma held constant for this calculation?

We have slightly changed this section to only show results for mid-century (but results for more metrics) so we believe this question is now mute. We were however using different sigmas for mid-century and end-of-century signal to noise calculations.

**Line 326:** Only grid points and global averages were shown, not a range of aggregation.

Apologies for the misleading sentence. We repeat here the answer to a previous point: We have considered intermediate aggregations in the analysis phase, but we could not make it work to show them without making the paper too tedious a list of results. We have mentioned testing the forced component error analysis at different spatial scales in the corresponding section of the paper:

"*In the appendix we report the results of applying the same analysis to the rest of the indices. We cannot show all results, but we tested country averages, zonal averages, land- and ocean-areas averages separately, confirming that the qualitative behavior we assess here is common to all these other scales of aggregation.*"

Otherwise the point is redundant, given that most our results are for the grid-point scale and therefore provide an upper bound for n, so we do not mention additional scales any longer.

**Line 336:** Are the variables of interest here normally distributed? Shouldn't they obey an GEV distribution

As we argued a little earlier, aside from the moment when we start considering grid-point level, yearly-resolved quantities for our GEV analysis, everything else undergoes multi-year averages (difference of five year means in the cases of grid-point level patterns of change). We do not believe the GEV would be a good choice for quantities that result from averaging multiple years.

**Line 365:** Missing parentheses

Thank you, corrected.

---

## Author Comment (AC2)

We thank both referees for a careful reading of the paper and many suggestions that have improved it substantially.
In the following we provide responses to all points raised by Referee 2.
Please, also note that the paper has been edited throughout for clarity and rigor, and we will provide a version with tracked changes.

In the following, we highlight our responses in blue, with italics used when we cite word by word from the paper (apologies for the occasional latex syntax left throughout these verbatim sections).

Thank you,

Claudia Tebaldi and co-authors.

**Referee 2**

The paper 'Extreme Metrics and Large Ensembles' considers the problem of ensemble size applied to 6 widely used extreme metrics. It asks whether we can estimate the ensemble size needed using a smaller 5-member ensemble, then validates the answer using two large ensembles. This manuscript provides additional information to the field and presents novel results. However, there are a few issues that need to be addressed prior to publication.

Thank you for an overall positive and constructive review and the many concrete suggestions.

Major comments:

1. Milinski et al 2020 discuss the problem of bootstrapping and how the errors increase as you approach the ensemble size. This is discussed in the manuscript on lines 108 onwards, however it is then largely ignored for the rest of the manuscript. This is a major issue as some of the results approaching the size of the ensemble may be buased due to this problem. Additionally this manuscript determines that the ensemble size needed is 20-25members, which is about hald the ensemble size and where this problem starts affecting the results. This needs to be addressed before this manuscript is published.

We agree wholeheartedly, and that is why we use the bootstrap approach only in the opening of our work, exactly to show how it may be biased, and to quantify such effect. The remaining of our analysis is based on the use of the formula for the determination of the ensemble size given a certain tolerance for error, so we never use the bootstrap

again after showing and quantifying in our tables exactly what the reviewer, and the Milinski et al. analysis, pointed out.

2.All Figures are too small, on a printout it is impossible to see what the Figures are. On the screen one needs to magnify the pdf to be able to see anything.

We have redrawn all figures (also added several in the appendix) and we have improved legibility throughout.

3. Details of the statistics used and results are missing, sometimes the text is vague or non-speciifc. See comments below for details.

We apologize for the oversight when failing to describe our statistics. We hope to have addressed all shortcomings, please see below.

In the following we answer to specific comments, please note however that we have extensively edited the paper, in order to make the discussion clearer and more detailed and add caveats throughout according to Reviewer 1's and 2's suggestions.

Minor issues:

**The title** could be more descriptive as currently one has no idea the paper is about ensemble size from the title.

Thank you for this  good observations. We have changed the title to: Extreme metrics from large ensembles: investigating the effects of ensemble size on their estimates.

**line 20** 'like' should be replaced with 'such as'

Corrected

**line 40** I think your citation of CanESM2 large ensemble is wrong: https://open.canada.ca/data/en/dataset/aa7b6823-fd1e-49ff-a6fb-68076a4a477c

Thank you, we now use Kirchmeier-Young et al. 2017 and Kushner et al. 2018 as the webpage suggests.

**lines 59-62** there are only citations for CESM not CanESM here

We added Arora et al., 2011 for CanESM2 which is used as a reference for CanESM2 in Kushner et al. 2018 and Kirchmeier-Young et al. 2017

**line 63** - I don't think this is the correct initialisation procedure for CanESM2 see https://open.canada.ca/data/en/dataset/aa7b6823-fd1e-49ff-a6fb-68076a4a477c

Thank you, we have added that 5 different historical simulations were used for CanESM2 ensemble initialization.

**line 65** - you could cite Marotzke, 2019 here: https://wires.onlinelibrary.wiley.com/doi/full/10.1002/wcc.563

Thank you, we do now.

**Line 70** - would it make sense to show the results for the larger ensemble in the main paper?

We apologize for pushing back on this, which would mean a complete overhaul of the paper. In addition, by using the CESM ensemble we are choosing to show the results for a model whose resolution is closer to state-of the art ESMs at this point in time. This we hope would make the results more relatable, also considering the nature of CESM as a community model.

**Equations on page 4** - you need to define what each term in the equation is

We have added mention of mean, scale and shape and their meaning, now, also in response to a similar note by Reviewer 1, as

"[...] *represent the location, scale and shape parameter respectively, responsible for the mean, variability and tail behavior of the random quantity z."*

**line 126** - please define X

We have rewritten this sentence as in:

"*On the basis of the GEV parameters estimated for a range of ensemble sizes n up to the full size available we compute return levels for several return periods X, X=2,5,10,20,50,100 and their uncertainty and assess when the estimates of the central value converge and what the trade-off is between sample size and width of the confidence interval."*

**Section 3.2** what tests do you use to detect changes in variance?

We use F-tests and we now have added that information which we forgot to specify because of an oversight.

**Figure 1 -** titles on the subplots could be more descriptive

We have changed them accordingly.

**Figure 2** – ofthe should be of the

Corrected, thank you

**Figure 2 -** I don't fully understand what the diagonal line is. Is it the actual time evolution of the expected error?

**I**t was the linear trend of the time evolution of sigma/sqrt(n). We have decided to erase this line, also according to Reviewer 1's criticism, since it seemed to create only confusion rather than add information.

**[TODO]**

**Line 211** - do you have a hypothesis why the error exceeds the expected error in these regions?

We show in our table that the fraction of the global (or ocean, or land) areas that show an error exceeding the 95% bound is in fact consistent with 5%, so we deem this behavior consistent with the distribution of the mean component that we estimate through our sigma/sqrt(n) computation. However we have added a short point that Reviewer 1 suggested about the possible effect of autocorrelation in the samples used to estimate sigma:

"The prevalence of red areas over the oceans could be due to an underestimation of $\sigma_i^c$ linked to the use of the 5-year windows and the autocorrelation possibly introduced, consistently with ocean quantities having more memory than land quantities, but we do not explore that further here."

**Line 215** - be specific, for which variabiles are you talking about?

We added "*for all metrics considered in our analysis*"

**Figure 5** - how do you calculate 95% confidence

We have now specified that we use the standard maximum likelihood approach to the computation of the CIs.

**Line 252** - more detail please

We have added an explicit reference to the confidence intervals that are computable by the GEV approach, that is what we meant here. The sentence now reads:

"*As for the results of the empirical counting approach, i.e., the blue point estimates, we can assess that in the majority of cases, but not across the board when we look closely to all the plots in the appendix, they do not deviate significantly from the central estimates based on fitting the GEV using the same sample size. However, while the latter can provide a measure of uncertainty through confidence intervals, the estimates based on counting events do not come with uncertainty bounds.*"

**Line 275** - this is really an odd sentence cna you rephrase?

We have rephrased as in

"*For all times considered, 5 members are sufficient to estimate an ensemble variance indistinguishable, statistically, from that which would be estimated using the full ensemble at most grid-points over the Earth's surface, as the light blue color indicates. For some sparse locations, however, 10 members are needed to achieve the same type of accuracy.*"

**Line 281** - more detail please

 The sentence now reads:

"*We note here that the patterns shown in some of these figures have indeed the characteristics of noise. To minimize that possibility we have applied a threshold for the significance of the p-values from the F-test obtained through the method that controls the False Discovery Rate (Ventura et al., 2004). The method has been shown to control for the false identification of significant differences "by chance" due to repeating statistical tests hundreds or thousands of times, as in our situation. The same method has been proved effective, in particular for multiple testing over spatial fields, despite the presence of spatial correlation (Ventura et al., 2004; Wilks, 2016). We fix the false discovery rate to 5%.*"

**Figure 9 -** could you zoom in on the Arctic as this is the only place there are colours, it is otherwise small, and seems to have limited information in the plot

We have changed the plots to show the Northern Hemisphere only for hot extremes and the Southern only for cold extremes (in the appendix too), thank you for the suggestion.

**Line 294** - how do you detect changes?

We use an F-test comparing variance computed at the different times during the simulation. The F-test is now mentioned explicitly, apologies for the oversight.

**Line 300** - missing specific details here

**Line 307** - it is not clear how you calculate significance

In response to both these comments we have added a sentence that should clarify our analysis of significant changes in variance:

"*Here as in the previous analysis a significant change is detected when the F-test for the ratio of the two variances that are being compared across time has a p-value smaller than the threshold determined by applying the false discovery rate method, and fixing the false discovery rate to 5%.*"

**Section 4.4 -** this is short and has limited detail in it. S/N really is dependent on the quantity due to the size of each term. I feel like very little is actually said in this section. Maybe think about what the point you want to make here is?

We have attempted to make this section more representative of what we consider some interesting differences in metrics which we had definitely overlooked in the previous version. We decided to focus on the mid-century results as the end-of century would be too boring, and we focused on the requirements for $S\_N=2$. We hope the reviewer will find the new section improved and worth keeping.

**5. Conclusions** - perhaps mention recent regional large ensembles here such as: https://www.climex-project.org/

Thank you, done

**Line 358** - Marotzke et al, 2019 and Hawkins et al, 2016
https://link.springer.com/article/10.1007/s00382-015-2806-8

Thank you, added.

---

## Referee Report (RR1)

esd-2021-53 version 2 Review

The authors have incorporated or responded satisfactorily to all my comments on the initial submission. There remain two essential issues to rectify before the paper is ready to publish (corresponding comments are marked with an asterisk):

1) The authors conclusions regarding the number of ensembles needed to detect variance changes are too strong with regard to precipitation.

2) The method for estimating the empirical CDF, from which the blue dots in figure 5&6 are obtained, should be discussed. (I missed this the first round, apologies).

In the interest of time, I think the editor can determine whether the authors sufficiently address the points above, so I do not need to see the second revision, though I am happy to review it should the editor request.

Detailed comments:

*Line 15: This implies variance change can be detected with 5-10 members. The results do not support this for the case of Rx5Day (see comments below re lines 278, 343.)

Line 127: I would change 'responsible for' to 'related to' since the location parameter is not exactly the mean of the tail and variability and tail behavior and not well defined.

Line 133: The URL runs off the page.

Line 232 consistently —> consistent

Line 276-278 While this is true, the actual size of the confidence intervals will not be known in your scenario of 5 initial runs, as a larger ensemble would be needed.

*Line 278 The 'counting' approach should be inaccurate for long return periods using a small number of ensembles. For N-yr events where N is larger than n*11, the CDF values need to be extrapolated. Consider removing the blue dots (and associated discussion in the text) or only using a single estimate from the full ensemble. If left in, provide a description of how the CDF was estimated.

Line 299 Consider rephrasing "bound to be an upper bound".

Line 341 Correct 'a spatially noisier pictures'.

*Line 343 The results do not support this conclusion, in my opinion. There seem to be quite a number of locations where n is larger (often much larger) than 5. To quantify this, show the histograms corresponding to the panels in Figures 9 & 10 in the supplementary material.

*Line 398 - 400. Related to the above comment, this statement is far too strong. There are a lot of point where n>5 in the top right panel of Figure 9, corresponding a change in Rx5Day between 1950 and 2100 (the largest time difference). My interpretation is that for rain metrics, you need a minimum of ten ensemble members to be reasonably confident you can detect variance changes in most locations where they occur (around 90%) or so. But this is hard to estimate from the maps, which is why I suggest showing the histograms in the supplement.

Figure 3, 4 9 & 10 are sideways.

---

## Author Response (AR2)

Dear Christian,

We thank the reviewers for the care they took in assessing our first revision and suggesting further improvements to the solidity of our results and discussion, which we have tried to implement. Here below please find a point-by-point response to the new round of reviews. We are also submitting a version of the manuscript with track changes.

Please note however that we are not sure how to interpret Reviewer 2's discomfort with our results pointing at 20-25 members as a sufficient size for most applications we have considered. His/her point would be justified, we think, only if we were using the bootstrap as the basis for our results. As we also explained in the first round of review, since we had the impression that the reviewer thought our study was based mainly on bootstrapping, we instead only use the bootstrap as a preliminary and limited comparison to our proposed approach, which does not rely on resampling. We have made this argument in our response, but if we are mistaken in interpreting what the reviewer meant we are ready to work on it further. In that case however we need help in understanding the criticism.

Meanwhile, we hope what we have done for this new version is satisfactory. Thank you and the reviewers for your fair handling of our work.

Claudia and co-authors.

PS We left the table colored but they do not need to be, since we were told that cannot be done. We have underlined the values that we want to highlight, and refer to them in the caption as underlined, not colored differently.

**Reviewer #1**

The authors have incorporated or responded satisfactorily to all my comments on the initial submission. There remain two essential issues to rectify before the paper is ready to publish (corresponding comments are marked with an asterisk):

1) The authors conclusions regarding the number of ensembles needed to detect variance changes are too strong with regard to precipitation.

2) The method for estimating the empirical CDF, from which the blue dots in figure 5&6 are obtained, should be discussed. (I missed this the first round, apologies).

In the interest of time, I think the editor can determine whether the authors sufficiently address the points above, so I do not need to see the second revision, though I am happy to review it should the editor request.
*We thank the reviewer for pointing out these additional needs, we have tried to satisfy them carefully. Please see below for details.*

Detailed comments:
*Line 15: This implies variance change can be detected with 5-10 members. The results do not support this for the case of Rx5Day (see comments below re lines 278, 343.)
*Thank you for pointing this out, we do agree and apologize for this oversight. Also, we agree that histograms could bring out these results more clearly. Therefore, we have reworded the discussion of the results regarding the detection of the variance changes, starting from the abstract, and added histogram plots mirroring the layout of Figure 9 and 10 to the supplementary material.*
*The abstract now has this additional sentence: "However, the detection of changes in the variance when comparing different times along the simulation requires larger sizes, up to 15 or 20 especially for the precipitation-based metrics." Please also see below for additional modifications regarding this issue.*

Line 127: I would change 'responsible for' to 'related to' since the location parameter is not exactly the mean of the tail and variability and tail behavior and not well defined.
*Agreed, and changed accordingly.*

Line 133: The URL runs off the page.
*We rely on the typesetting to fix this type of issue, but we have for now moved the url in a footnote.*

Line 232 consistently —> consistent
*Corrected, thank you.*

Line 276-278 While this is true, the actual size of the confidence intervals will not be known in your scenario of 5 initial runs, as a larger ensemble would be needed.
*For this set of results we are using increasing sample sizes, not relying on any estimate derived from the first 5, so the assessment that we are talking about would be achieved by considering the size of the confidence intervals as the ensemble size is increased gradually. Also, in theory, even the width of the confidence interval for the estimate obtained by 5 ensemble members could be considered narrow enough for some type of application not requiring highly precise estimates.*

*Line 278 The 'counting' approach should be inaccurate for long return periods using a small number of ensembles. For N-yr events where N is larger than n*11, the CDF

values need to be extrapolated. Consider removing the blue dots (and associated discussion in the text) or only using a single estimate from the full ensemble. If left in, provide a description of how the CDF was estimated.

*We consider more appropriate to compare the results obtained by the GEV analysis to that obtained by computing the empirical CDF on the basis of the same number of data points, but we agree with the reviewer that for the only case of the first blue point (n=5, based on 55 data points) in the 100-yr return level plots the empirical result is based on interpolating the value of the CDF between the 54/55~=98th percentile and the 100$^{th}$, rather than a straightforward count. We have specified that in the caption by adding the sentence* "Note that in the 100-yr return level plots the first such dot is obtained by interpolation of the last two values of the CDF, since the sample size is less than 100 (see text).". *We have also now added a description of the computation of the empirical CDF in the text of the Methods section. That section of the text now reads:* "Lastly, we can use a simple counting approach, based on computing the empirical cumulative distribution function from the same sample, to determine those same $X$-year events. I.e., after computing the empirical CDF we choose the value that leaves to its right no more than $p*n*11$ data points, where $p$ is the tail probability corresponding to the $1/p$-year return period as defined above."

Line 299 Consider rephrasing "bound to be an upper bound".
☺ *yes, we now say* "as that answer serves as an upper bound".

Line 341 Correct 'a spatially noisier pictures'.
*Done, thank you.*

*Line 343 The results do not support this conclusion, in my opinion. There seem to be quite a number of locations where n is larger (often much larger) than 5. To quantify this, show the histograms corresponding to the panels in Figures 9 & 10 in the supplementary material.
*Line 398 - 400. Related to the above comment, this statement is far too strong. There are a lot of point where n>5 in the top right panel of Figure 9, corresponding a change in Rx5Day between 1950 and 2100 (the largest time difference). My interpretation is that for rain metrics, you need a minimum of ten ensemble members to be reasonably confident you can detect variance changes in most locations where they occur (around 90%) or so. But this is hard to estimate from the maps, which is why I suggest showing the histograms in the supplement.

*We agree with these comments, apologize for the oversight, and we have added such figures in the supplementary material, together with rewording the discussion of these results.*
*The relevant section now reads:* "In the case of TNx, a metric based on daily minimum temperature, the changes are confined to the Arctic region and in most cases the ensemble size required is again 5, with only one instance where the changes between mid-century and end-of the century require consistently a larger ensemble size over an appreciable extent (as

*many as 15 members over the region). When we conduct the same analysis on the precipitation metric, shown in Figure~\ref{VarianceChanges_Rx5Day}, we are presented with a spatially noisier picture, with changes in variance scattered throughout the Earth's surface, especially over the oceans. In the case of this precipitation metric the ensemble size required is in many regions as large as 15 or 20 members. These results are made clearer by Figures~\ref{VarianceChanges_TNx_histograms}and~\ref{VarianceChanges_Rx5Day_histograms} in the Appendix, where the grid-boxes where significant changes are present are gathered into histograms (weighted according to the Earth's fraction that the grid-boxes represent) that show the ensemble size required along the x-axis.*

*We highlight in those figures the fact that for the temperature-based metric only three histograms, corresponding to three specific time-comparisons, gather grid-boxes covering more than 5\% of the Earth's surface, while the coverage is more extensive than 5\% for all time comparisons for the precipitation metric.*

*These results are representative of the remaining metrics and the alternative model as Figures~\ref{VarianceDiff_TNn} through~\ref{VarianceChanges_Rx5Day_CanESM} in the Appendix document.”*

Figure 3, 4 9 & 10 are sideways.
*We did this by purpose as they are wider than they are tall, to facilitate reading, but we hope to rely on the journal technical help for an optimal setting.*

**Reviewer #2**

Second review of Extreme Metrics from Large Ensembles: Investigating the Effects of Ensemble Size on their Estimates

The authors have largely responded to and made changes to my original comments, I recommend minor revisions.

My main concern is still the claim that 20-25 members is enough given that this is ½ the ensemble size. The argument on lines 10, 391 and 395 seems to be on contradiction with your argument on line 185 and Milinski et al.

*We wonder if the reviewer still thinks that we are basing our results on a bootstrap approach, which we are not. All our results are based on estimating quantities/variability/standard errors on the basis of formulas in which we plug in an estimate derived from a 5-member ensemble, or by showing how results change for increasing sample sizes, but without involving bootstrapping (aside from the initial exercise to set the stage). Therefore, none of our result should be undermined by the fact that 20/25 is half the ensemble size, while we agree with the reviewer that if we were using the bootstrap there would be a problem, also in line with Milinski et al.'s argument. If we are*

*misinterpreting the reviewer's concerns, then we apologize and remain eager to clarify what needs to be clarified.*

Minor comments:

46 – you should cite both Hawkins & Sutton papers: https://link.springer.com/article/10.1007/s00382-010-0810-6
*Done, thank you.*

69 – this is not completely true, this is partially addressed in the CanESM2 setup
*Noted. We have rephrased as in* "We note that sources of variability from different ocean states, particularly at depth, are not systematically sampled by these type of ensembles, albeit they are partially addressed by the design of the CanESM2 that uses de-facto different ocean states."

125 – I am still unclear on what each term in the equation is
*This was a standard way to introduce a GEV distribution, so we are not sure how to change it to make it more readable. We have tried to reword the definition slightly. We are not pasting the new wording here due to the mathematical notation that would go awry.*

133 – issue with formatting
*We have moved the url to a footnote.*

251 – should this paragraph be in the methods?
*Good point, we have moved it there and reworded a bit the entire section about the GEV methodology in the Methods section.*

369 – why do we expect this?
*Every quantity emerging at the grid-point of a GCM represents an average quantity, and the average is representative of the size of the grid-box. Therefore, we expect quantities that are representative of larger averages to be less affected by noise. The same way as if we were averaging across grid-boxes, the larger the region considered, the stronger the signal should be, all else being equal.*

391 – should read ' we compared;
*We changed "We use" to "we compared" hoping this is what the reviewer meant.*

417 – should be e.g. Knutti as there are many studies on this topic
*Good point, corrected.*

Papers that also look at ensemble size that might be worth citing in the introduction. The authors can decide if these are useful.
*Thank you for this list, and for taking the time to include the urls!*

Deser, C., Phillips, A., Bourdette, V., and Teng, H.: Uncertainty in climate change projections: the role of internal variability, Clim. Dynam., 38, 527–546, 2012
*We already cite this first one.*

Olonscheck, D. and Notz, D.: Consistently Estimating Internal Climate Variability from Climate Model Simulations, J. Climate, 30, 9555–9573, https://doi.org/10.1175/JCLI-D-16-0428.1, 2017.
*This paper does not use initial condition ensembles.*

Li, H. and Ilyina, T.: Current and Future Decadal Trends in the Oceanic Carbon Uptake Are Dominated by Internal Variability, Geophys. Res. Lett., 45, 916–925, 2018
*Included, thank you.*

Steinman, B. A., Frankcombe, L. M., Mann, M. E., Miller, S. K., and England, M. H.: Response to Comment on "Atlantic and Pacific multidecadal oscillations and Northern Hemisphere temperatures", Science, 350, 1326, https://doi.org/10.1126/science.aac5208, 2015
*Included, thank you.*

Pausata, F. S. R., Grini, A., Caballero, R., Hannachi, A., and Seland, Ø.: High-latitude volcanic eruptions in the Norwegian Earth System Model: the effect of different initial conditions and of the ensemble size, Tellus B, 11, 2050–2069, 2015
*Included, thank you.*

Bittner, M., Schmidt, H., Timmreck, C., and Sienz, F.: Using a large ensemble of simulations to assess the Northern Hemisphere stratospheric dynamical response to tropical volcanic eruptions and its uncertainty, Geophys. Res. Lett., 43, 9324–9332, 2016
*Included, thank you.*

Daron, J. D. and Stainforth, D. A.: On predicting climate under climate change, Environ. Res. Lett., 8, 034021, https://doi.org/10.1088/1748-9326/8/3/034021, 2013
*We left this out as it uses simpler models.*

Drótos, G., Bódai, T., and Tél, T.: On the importance of the convergence to climate attractors, Eur. Phys. J. Spec. Top., 226, 2031–2038, https://doi.org/10.1140/epjst/e2017-

70045-7, 2017
*We left this out due to the theoretical nature of the discussion.*

Maher, N., Matei, D., Milinski, S., and Marotzke, J.: ENSO change in climate projections: forced response or internal variability?, Geophys. Res. Lett., 45, 11390–11398, 2018
*Included, thank you.*

Table 1 – did you try bootstrapping with replacement? Is the colour or the underline important as you refer to both?
*We use the  bootstrap, without replacement, to confirm the Milinski et al. argument, and to position our alternative method with respect to it. We do not use the bootstrap for anything further, and not for any of our results.  We therefore did not try anything but the simple bootstrap approach without replacement, as we do not consider the bootstrap a focus of our study. Sampling with replacement would also introduce unrealistic replicates of the values, making the estimation of the error even more biased low.*
*We did not mean for "highlighting" to refer to the colors, but rather in the general sense. We changed that to "pointing out" to avoid confusion.  We expect the colors to disappear in the journal version, as we were told the journal does not print colored tables.*

Figure 3/4 – add % on colourbar
*Added.*

Figures 5/6 are still really small
*Apologies but we cannot figure out a way to expand the size without cutting out the legend. We will work with the journal to try and maximize the quality of the graphics.*

Figure 9 does not show much, did you consider zooming in on just the Arctic?
*Apologies but we cannot get a projection of the Arctic looking satisfactory with our R graphic package.*